# Pt-doped Ru nanoparticles loaded on 'black gold' plasmonic nanoreactors as air stable reduction catalysts

Gunjan Sharma[1], Rishi Verma [1], Shinya Masuda [2], Khaled Mohamed Badawy[3], Nirpendra Singh [3], Tatsuya Tsukuda [2] ✉ & Vivek Polshettiwar [1] ✉

This study introduces a plasmonic reduction catalyst, stable only in the presence of air, achieved by integrating Pt-doped Ru nanoparticles on black gold. This innovative black gold/RuPt catalyst showcases good efficiency in acetylene semi-hydrogenation, attaining over 90% selectivity with an ethene production rate of 320 mmol $g^{-1}$ $h^{-1}$. Its stability, evident in 100 h of operation with continuous air flow, is attributed to the synergy of co-existing metal oxide and metal phases. The catalyst's stability is further enhanced by plasmon-mediated concurrent reduction and oxidation of the active sites. Finite-difference time-domain simulations reveal a five-fold electric field intensification near the RuPt nanoparticles, crucial for activating acetylene and hydrogen. Kinetic isotope effect analysis indicates the contribution from the plasmonic non-thermal effects along with the photothermal. Spectroscopic and in-situ Fourier transform infrared studies, combined with quantum chemical calculations, elucidate the molecular reaction mechanism, emphasizing the cooperative interaction between Ru and Pt in optimizing ethene production and selectivity.

Plasmonic nanochemistry provides approaches to develop light-harvesting nanoreactors capable of exceeding the limitations of conventional catalysts[1–11]. Photoexcitation of plasmonic metal nanoparticles (NPs) can trigger direct photochemical reactions. However, the chemical reactivity of these metal surfaces limits their application in catalysis for only a few reactions[12,13]. Therefore, functional nanostructures with a plasmonic material that concentrates light energy and efficiently guides it to more active catalytic sites are desired[13–15]. These 'hybrid plasmonics' can pave new ways for various catalytic transformations with better activities and selectivities by providing alternative reaction pathways and the efficient use of solar light[16–20]. One such challenging reaction is acetylene semi-hydrogenation in excess ethene in terms of high selectivity and conversion[21–36]. Surprisingly, reports on photocatalytic acetylene semi-hydrogenation are scarce[31,35–37], even though it has been shown as a promising method for enhancing the activity and selectivity of numerous important reactions like $CO_2$

hydrogenation, reverse water gas shift reaction, and Fischer-Tropsch synthesis[38–41].

This work reports a hybrid plasmonic reduction catalyst, synthesized by loading Pt-doped Ru nanoparticles (RuPt NPs)[42] over dendritic plasmonic colloidosomes (DPC) of gold[43]. The catalyst design consists of two components- i) DPC, also known as black gold[43] (Au deposited on dendritic fibrous nano-silica (DFNS)[44] using cycle by cycle approach), which can harvest a broad region of visible light and generate hot-spots due to the plasmonic coupling between Au NPs[45] and ii) Pt-doped Ru NPs (Ru:Pt = 90:10) as the catalytic sites[42], rationally designed to control the extent of semi-hydrogenation. RuPt NPs loaded on DPC (DPC/RuPt) showed an ethene production rate of 320 mmol $g^{-1}$ $h^{-1}$ for competitive acetylene semihydrogenation (in excess ethene) with over 90% selectivity just by utilizing solar light. Moreover, DPC/RuPt was stable only in the air and showed on-stream stability for at least 100 h for acetylene semi-hydrogenation with a high

[1]Department of Chemical Sciences, Tata Institute of Fundamental Research, Mumbai 40005, India. [2]Department of Chemistry, Graduate School of Science, The University of Tokyo, Tokyo 113-0033, Japan. [3]Department of Physics, Khalifa University, Abu Dhabi 127788, United Arab Emirates. ✉e-mail: tsukuda@chem.s.u-tokyo.ac.jp; vivekpol@tifr.res.in

gas hourly space velocity (GHSV) of 1,320,000 mL g$^{-1}$ h$^{-1}$. Several mechanistic studies, like light intensity dependence, temperature dependence in the dark, and kinetic isotope effect (KIE) were conducted to understand the mechanism of plasmonic activation. Finite-difference time-domain (FDTD) simulations were carried out to investigate the electric field enhancement due to localized surface plasmon resonance (LSPR) and its spatial distribution. In-depth structural characterizations were performed using high-angle annular dark-field scanning transmission electron microscopy (HAADF-STEM), energy dispersive X-ray spectroscopy (EDS), powder X-ray diffraction (PXRD), X-ray photoelectron spectroscopy (XPS), X-ray absorption near edge structure (XANES) spectroscopy, extended X-ray absorption fine structure (EXAFS) analysis, temperature programmed reduction (TPR), thermogravimetric analysis (TGA), and Raman spectroscopy. In-situ Fourier transform infrared (FTIR) studies were conducted to investigate the dynamics of intermediate formation, and a molecular reaction mechanism was proposed and supported by density functional theory (DFT) calculations.

## Results and discussions

### Plasmonic acetylene semihydrogenation over DPC/RuPt

We impregnated polyvinylpyrrolidone (PVP)-stabilized RuPt clusters over DPC with various loadings (3, 5, 10, and 20 wt%) (Table S1-S2), naming catalysts as DPC/RuPt-wt%. The resultant as-prepared catalyst (named ASP) was then calcined to remove PVP in the air at 800 °C (named Calc) (Fig. S1). Non-competitive acetylene semi-hydrogenation was carried out in a flow reactor at 1 bar pressure with a total flow of 100 mL min$^{-1}$ (C$_2$H$_2$/H$_2$/Ar = 1/5/94 mL min$^{-1}$) under the illumination of visible light (Xenon lamp, 400–1100 nm, 2.7 W cm$^{-2}$) and the products were monitored using online micro-gas chromatography (GC) (Fig. S2). The best-performing catalyst DPC/RuPt-10-Calc was used for further studies (Fig. S3–S14). We then conducted plasmonic acetylene semi-hydrogenation in excess ethene using the flow reactor. First, the acetylene percentage (Fig. S15) and H$_2$:C$_2$H$_2$ ratio (Fig. 1a) in the reactant feed were optimized. We then optimized the GHSV using 5 mg of DPC/RuPt-10-Calc at 1 bar pressure under light (2.7 W cm$^{-2}$). Under the optimized conditions, the ethene production rate of 320 mmol g$^{-1}$ h$^{-1}$ was achieved with GHSV of 1,320,000 mL g$^{-1}$ h$^{-1}$ (total gas flow-110 mL min$^{-1}$) (Fig. 1b). With a further rise in the space velocity, the increment in production rate was much less to compensate for the loss in conversion, which dipped below 20% due to a decrease in the residence time of the reactant gases on the active sites.

Photocatalytic acetylene semi-hydrogenation was carried out under varying light intensities without external heating (Fig. 1c, S16). The catalytic activity showed superlinear dependence of ethene production on the light intensity with a power law exponent of 2.37 (rate ∝ I$^n$) and an increase in the quantum efficiency up to 3 W cm$^{-2}$ (Fig. 1c, Fig. S17), indicating a hot electron-mediated non-thermal pathway[7]. However, there have been reports by different groups[46–50], emphasizing that a similar superlinear dependence trend can be obtained by assuming a purely thermal mechanism and a temperature-shifted Arrhenius equation based on a linear dependence of temperature on light intensity model. We checked the applicability of this model[49] to our catalytic system. The catalyst bed temperature (Ts) was measured at every light intensity using a thin thermocouple inserted directly into the catalyst's powder bed to determine the contribution of plasmonic photothermal effects (Fig. S18, S19). The catalysis was then carried out in the dark at different temperatures to understand purely thermal effects (Fig. 1d). The best production rate of ~300 mmol g$^{-1}$ h$^{-1}$ was achieved at Ts = 200 °C (in the dark) by external heating, similar to what was achieved at 3 W cm$^{-2}$ (Ts = 137 °C). This indicated the role of both thermal and non-thermal effects and the possibility of lowering the activation energy barrier during plasmonic catalysis. We observed that the difference between the temperatures at which the same catalyst activity was obtained in dark T (I$_{inc}$) and the measured

temperature in light (T$_M$ =Ts) did not conform to the anticipated linear trend by the shifted Arrhenius equation at higher intensities (Fig. S19). Even in the linear regime, the non-thermal effects can still not be ruled out as this model[49] exclusively considers the temperature increase as the only effect of light excitation, neglecting the effect of light on the activation energy as discussed by Jain et al. [51], thus camouflaging the non-thermal effects under the guise of temperature increase. Also, the quantum efficiency of this reaction first increased with an increase in light intensity (up to 3 W cm$^{-2}$), consistent with the superlinear intensity dependence[7] and then decreased with a further increase in light intensity (Fig. S17b, S20), which could be due to more photo-induced heating of the catalyst at higher light intensities and consequently limited acetylene adsorption. Since the disparity in the catalyst's performance in light and dark conditions diminished at higher light intensities (higher catalyst bed temperatures) (Fig. S19a, S20c), we propose that photo-thermal effects dominate at higher light intensities. For purely thermal conditions in the dark, an Arrhenius-type relation is expected between the rate and temperature, but the production rate at reaction temperatures of 200, 220 and 250 °C showed a relatively small increase (Fig. 1d, S19a). This could be due to the sintering of the NPs at these temperatures. However, the HAADF-STEM image and EDS maps of the spent catalyst did not show any signs of sintering (Fig. S21). This indicated that at these higher temperatures, acetylene desorption was dominated over adsorption, which limited the overall rate of the reaction.

The catalytic reaction was then carried out at various wavelengths (Fig. 1e, S22-23) in the visible region. The production rate at different wavelengths was comparable owing to the broadband absorption of black gold. The Ts values also indicated a similar trend.

We then compared the activities of DPC/RuPt-10-Calc with their monometallic versions, DPC/Pt-1-Calc and DPC/Ru-9-Calc (Fig. 1f, S24). Notably, Ru was relatively inactive towards acetylene semihydrogenation, while Pt was less selective towards ethene in light and dark. DPC/RuPt-10-Calc outperformed both its monometallic counterparts, indicating the synergy between Ru and Pt to retain an excellent production rate and high selectivity towards ethene. The catalyst was then tested for long-term stability up to ~100 h. It was observed that flowing air along with the reactant feed could result in the long-term stability of the catalyst with negligible loss in its activity (Fig. 1g), which was not the case without flowing air (Fig. 1h, S25). A similar behaviour was observed during the non-competitive acetylene semi-hydrogenation, where the activity dropped to ~20% of its initial value in 100 h without airflow (Fig. S12). Interestingly, the activity could be recovered by high-temperature air treatment (Fig. S13). The in-depth characterizations of the spent catalysts were performed to understand the air-enhanced stability and are discussed in later sections.

### Structural characterization of the DPC/RuPt catalysts

Various characterization techniques were employed to correlate the catalytic performance of the catalysts with their physical and chemical properties. The high resolution transmission electron microscopy (HRTEM) images and EDS of DPC/RuPt-10-Calc showed successful loading of RuPt NPs on the DPC sphere (Fig. 2a–b, d–i). To test if the broadband absorption of DPC was altered during the loading and calcination procedure, UV-vis extinction spectra were recorded for DPC and DPC/RuPt-10-Calc. They showed a broadband absorption in the visible region (Fig. 2c). Au has been reported to show interband transitions in the wavelength region around 400 nm[52–54]. We believe that inter-band transitions, as well as plasmonic excitations, both occur in the wavelength region around 400 nm for black gold, but it is the plasmonic excitation that is dominant because of its higher absorption cross-section than the inter-band transitions. The plasmonic black gold employed in this work shows a broadband absorption profile in the visible range due to plasmonic coupling between different Au NPs at varying inter-particle distances (Fig. S26a), which

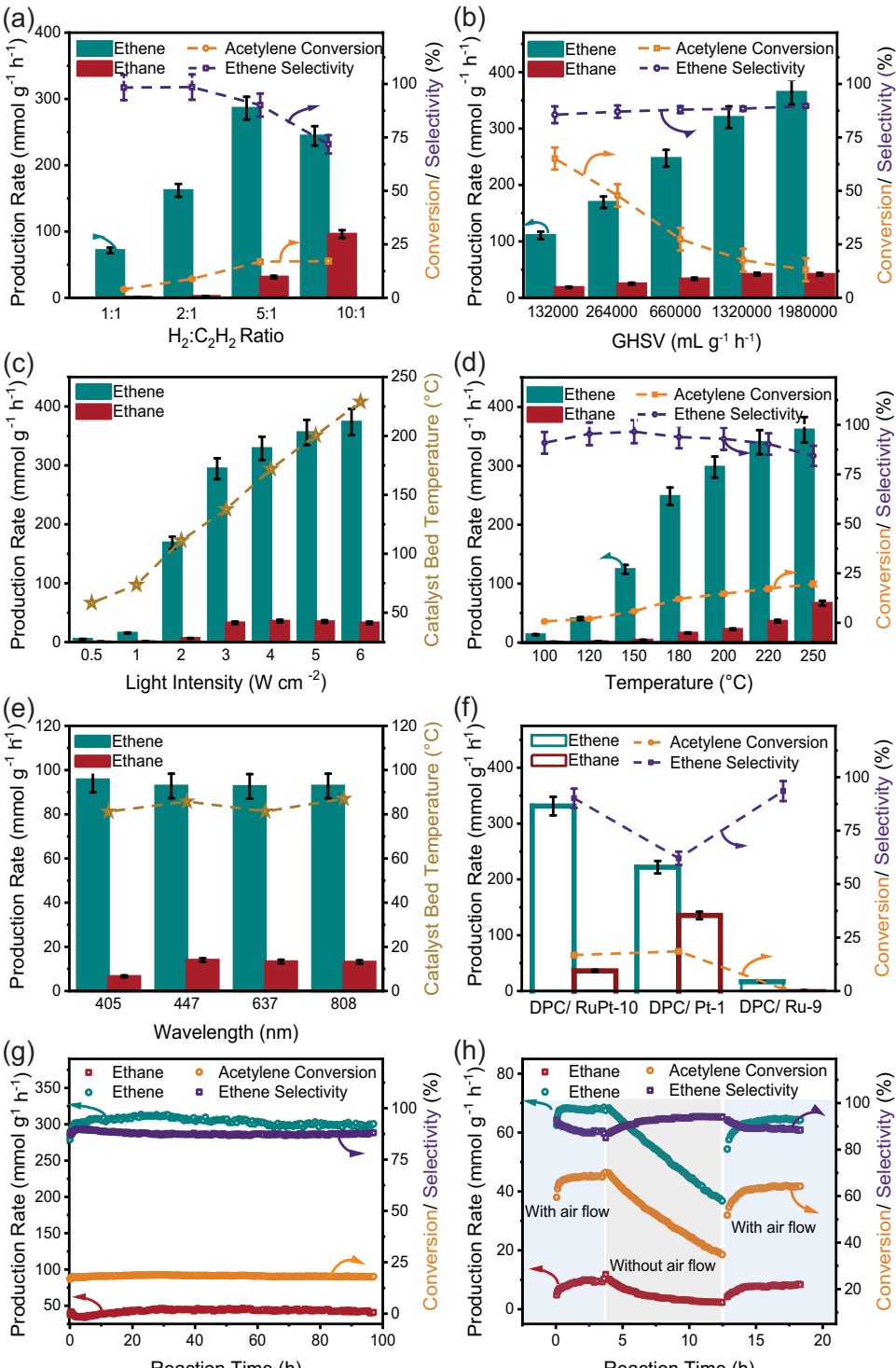

**Fig. 1 | Acetylene semi-hydrogenation in the presence of excess ethene using DPC/RuPt-10-Calc. a** at different $H_2:C_2H_2$ ratios with visible light illumination (2.7 W cm$^{-2}$ light intensity, 1 bar pressure, 3% acetylene in feed, $C_2H_4:C_2H_2$ = 20:1, air flow- 5 mL min$^{-1}$ and 1,320,000 mL g$^{-1}$ h$^{-1}$ GHSV); **b** Effect of GHSV (using 5:1 $H_2:C_2H_2$ ratio and 3% acetylene in feed, air flow- 5 mL min$^{-1}$, 2.7 W cm$^{-2}$); **c** Ethene and ethane production rate with the catalyst bed temperature reached at different light intensities (using optimized flow conditions-5:1 $H_2:C_2H_2$ ratio and 3% acetylene in feed, $C_2H_4:C_2H_2$ = 20:1, air flow- 5 mL min$^{-1}$ and 1,320,000 mL g$^{-1}$ h$^{-1}$ GHSV); **d** The effect of temperature on acetylene semi-hydrogenation over DPC/RuPt-10-Calc catalyst in dark at optimized flow conditions; **e** Effect of light wavelength on

production rate and corresponding catalyst bed temperatures (using optimized flow conditions at 2.7 W cm$^{-2}$); **f** Comparison of different air calcined catalysts at optimized reaction conditions and similar individual metal loadings; **g** Long-term stability of DPC/RuPt-10-Calc catalyst for competitive acetylene semi-hydrogenation, with 400–1100 nm illumination at 2.7 W cm$^{-2}$, 1 bar pressure, optimized flow conditions: $C_2H_2/C_2H_4/H_2/Ar/Air$-3/60/15/27/5 ml min$^{-1}$; **h** Effect of air flow on the stability of the catalyst in high conversion regime (Flow-$C_2H_2/C_2H_4/H_2/Ar/Air$-0.2/4/1/1.8/0.5 ml min$^{-1}$ at 2.7 W cm$^{-2}$, 400–1100 nm)). Error bars: Calculated from data of at least three repeated experiments (Source data are provided as a Source Data file).

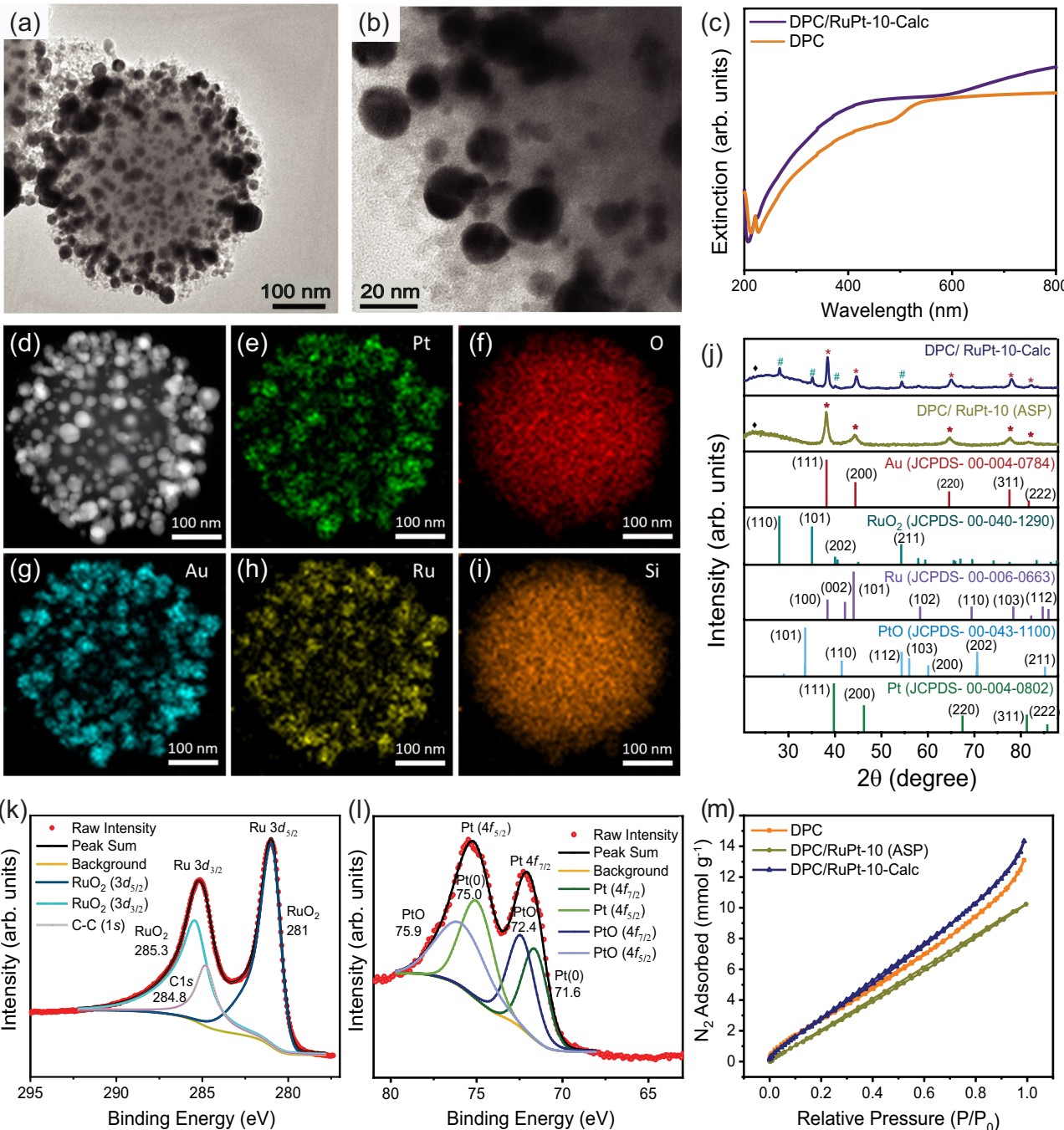

**Fig. 2 | Characterization and physicochemical properties of the DPC/RuPt-10-Calc catalyst. a**, **b** HRTEM images; **c** UV-vis extinction spectrum of DPC and DPC/RuPt-10- Calc as dispersion in ethanol; **d** HAADF-STEM image; **e–i** EDS elemental maps; **j** PXRD patterns for DPC/RuPt-10-ASP and DPC/RuPt-10-Calc; **k** XPS spectrum for Ru 3*d* and C 1*s* showing different oxides of Ru present in DPC/RuPt-10-Calc; **l** XPS spectrum for Pt 4*f* showing oxidation states of Pt in DPC/RuPt-10-Calc; **m** N₂ sorption isotherms for DPC, DPC/RuPt-10-ASP and DPC/RuPt-10-Calc. Error bars: Calculated from data of at least three repeated experiments (Source data are provided as a Source Data file).

makes it different from the Au nanoparticles having narrow band absorption (Fig. S26b). Hence, plasmonic excitation dominates over inter-band excitation in black gold because of its higher absorption cross-section, as indicated by the nearly constant reaction rates and catalyst bed temperatures across the visible region (Fig. 1e).

The PXRD patterns of DPC/RuPt-10-Calc showed peaks corresponding to $RuO_2$ and Au. In contrast, only Au peaks appeared for the as-prepared sample (DPC/RuPt-10-ASP) (Fig. 2j), confirming the oxidation of loaded Ru NPs during calcination. XPS also indicated the presence of $RuO_2$ (281, 285.3 eV)[55,56]. The oxidation of Pt to PtO was

shown by the XPS peaks corresponding to PtO (72.4, 75.9 eV) along with Pt (71.6, 75.0 eV)[57] (Fig. 2k–l). The shift of +0.5 eV in Pt (0) could be attributed to the interaction with oxygen-coordinated Ru. A similar positive shift was also observed in Au 4*f* peaks (Fig. S27)[58]. Ru and Pt oxidation states were also confirmed from XANES spectra and EXAFS oscillations at Ru K-edge, Pt L₃-edge, and Au L₃-edge were found to be in good agreement with XPS, as discussed in the later sections. The DPC/RuPt-10-ASP showed XPS peaks corresponding to Ru, four peaks of carbon from PVP, Pt(0), and Au(0) (Fig. S28). The nitrogen sorption isotherms of these catalysts showed typical type-II curves with weak

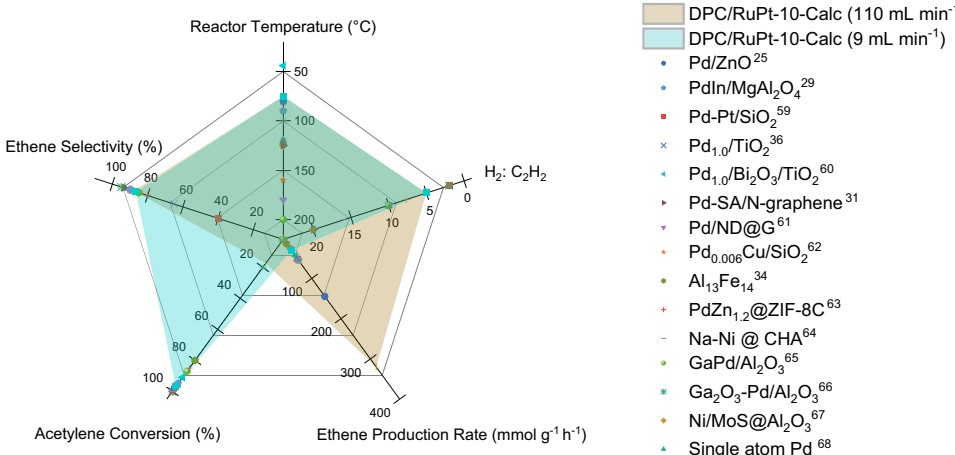

**Fig. 3 | Comparison of the best reported non-plasmonic catalytic systems.** employed for semi-hydrogenation of acetylene in excess ethene (A more comprehensive comparison, complete with experimental particulars, can be found in Table S4) (Source data are provided as a Source Data file).

hysteresis (Fig. 2m). The Brunauer–Emmett–Teller (BET) surface area (SA) of DPC/RuPt-10-ASP was similar to DPC (355 m² g⁻¹) with pore volume reduction from 0.41 to 0.31 cm³ g⁻¹, indicating the PVP-stabilized NPs occupying the pores. In contrast, the DPC/RuPt-10-Calc sample showed a higher SA of ~ 415 m² g⁻¹ with a pore volume of 0.45 cm³ g⁻¹. The higher SA and pore volume of DPC/RuPt-10-Calc than DPC could be due to the loss of 3-aminopropyltriethoxysilane, used in DPC synthesis, during calcination at 800 °C (Fig. S29, Table S3).

## Comparison with the best reported catalytic (plasmonic and non-plasmonic) systems

The catalytic performance of DPC/RuPt-10-Calc was compared with the best-reported catalysts (both plasmonic and non-plasmonic) for acetylene semi-hydrogenation in excess ethene in terms of ethene production rate, selectivity, acetylene conversion, H₂:C₂H₂ ratio, and reactor temperature (Fig. 3, Table 1 & S4). Initially, DPC/RuPt-10-Calc was exclusively compared with previously reported plasmonic catalysts, and it exhibited the highest ethene production rate among all the plasmonic catalysts (Table 1). Notably, it also maintains high selectivity at higher acetylene conversions (Fig. S30).

Additionally, aside from the plasmonic catalysts, DPC/RuPt-10-Calc was assessed in comparison to non-plasmonic catalysts (see Fig. 3 and Table S4)[25,29,31,34,36,59–68]. DPC/RuPt-10-Calc exhibited at par production rates and selectivity to the best-performing non-plasmonic catalysts. It showed the highest ethene production rate (320 mmol g⁻¹ h⁻¹) and high selectivity (~87%) at a comparatively lower H₂:C₂H₂ ratio (5:1) and low Ts of 132 °C and reactor temperature (T_R, Fig. S2) of 75 °C, achieved by solar energy (without external heating). It also showed long-term stability for at least 100 h in the presence of air. (Fig. 1h). The highest production rate was achieved with a total gas flow of 110 mL min⁻¹, but the acetylene conversion at such a high space velocity was moderate. At a lower flow rate of 9 mL min⁻¹ and higher illumination intensity of 6 W cm⁻², the conversion value of ~97% could be achieved while maintaining ~90% selectivity (Fig. S30). The large pentagon area spanned by DPC/RuPt-10-Calc (Fig. 3) indicated the efficiency of DPC/RuPt-10-Calc for acetylene semi-hydrogenation in excess ethene, with high ethene production rate, lower reactor temperature, and excellent stability for at least 100 h, using solar energy.

## Mechanism of plasmonic activation by light

Plasmonic activation involves many pathways, including concentrating the electric field around the nanoparticle which can facilitate bond polarization. To visualize the local electric field enhancement resulting from the LSPR, we conducted FDTD simulations. Our model included RuPt NPs (1.5 nm) deposited on Au nanospheres (10 nm) with a

separation of 3 nm. We investigated the impact of RuPt NPs on electric field enhancement (at light intensity: 2.7 W cm⁻², $E_0$ = 4509 V m⁻¹). Notably, without RuPt NPs (only DPC), a modest electric field enhancement of 4.9 was observed, with less concentration at the center (Fig. 4a). However, the presence of RuPt on DPC resulted in a significant electric field enhancement of 25.6 (five times that of DPC) (Fig. 4b, S31), owing to the near-field coupling between the RuPt NPs with DPC. The electric field was predominantly concentrated around the RuPt sites within the gaps of Au NPs (Fig. 4b, S31). We also investigated partially reduced RuPt sites (in the model DPC/RuPtOₓ), which exhibited similar behaviour to DPC/RuPt but with a lower enhancement factor of 19.2 (still four times higher than that of DPC) (Fig. 4c, S31). This small decrease in enhancement can be attributed to the reduced metallic character of RuPt sites affecting their near-field coupling to Au. The dependence of this enhancement on light intensity and wavelength (Fig. S32-S33) showed a negligible effect of the impinged light intensity, but enhancement was highest at around 550 nm. The heightened electric field was crucial in activating chemical bonds by inducing polarization.

The decay of plasmonic resonance generates hot carriers[2,4,7] (electrons and holes) within femtoseconds and localized heating[50,51,69] (electron-phonon coupling) within picoseconds, which can then activate the reactants in different ways[70–75]. To understand these plasmonic pathways in the current catalytic study, we investigated KIE, which depends on the nature of the activation pathway. The reactions driven by electrons (non-thermal pathway) have larger KIE in light as compared to dark whereas the KIE remains similar in case of reactions driven by phonons. (thermal pathway)[7]. The KIE was obtained as a ratio of ethene production rate using H₂ and D₂ in light with the light intensity of 3 W cm⁻² and in the dark at 137 °C. Notably, KIE measured in light (2.03) was found to be larger than that measured in the dark (1.34) (Fig. 4d). Same trend was observed at various light intensities and corresponding catalyst bed temperatures (Fig. S34), implying that there was a contribution from the non-thermal effects along with the photothermal activation of the reactants. The influence of plasmonic activation on the H₂ dissociation and rate-limiting step (RLS) of the reaction has been discussed in a later section.

As a function of Ts in light and dark, the Arrhenius plots showed a convex nature, indicating the temperature dependence of activation energy $Ea$ (Fig. 4e)[76]. This convex nature could be due to changes in adsorption and desorption kinetics of reactants at higher temperatures and light intensities, although this needs to be further studied for in-depth understanding. The $Ea$ values were plotted as a function of Ts in Fig. S35, which is a differential plot of Fig. 4e. The $Ea$ with light was smaller than that in the dark at all temperatures, which was evident

**Table 1 | Comparative analysis of plasmonic catalytic systems in acetylene semi-hydrogenation**

| Catalyst (metal loading and weight) | Light Intensity in W cm⁻² (λ) | Feed Composition in vol. % (total flow) | Reactor Type | Ethene Production Rate (mmol g⁻¹ h⁻¹) | Acetylene Conversion (%) | Ethene Selectivity (%) | Ref. |
|---|---|---|---|---|---|---|---|
| DPC/RuPt (10 %, Ru:Pt=9:1, 5 mg) | 2.7 (400–1100 nm) | $C_2H_2/C_2H_4/H_2/Ar/Air$ = 2.72/54.5/13.6/24.5/4.5 (110 mL min⁻¹) (With excess ethene) | Fixed bed flow reactor with quartz window (crucible i.d - 6 mm) | 320 | 18 | 88 | This Work |
| DPC/RuPt (10 %, Ru:Pt=9:1, 20 mg) | 6 (AM1.5) | $C_2H_2/C_2H_4/H_2/Ar/Air$ = 2.2/44.4/22.2/20/11.1 (9 mL min⁻¹) (With excess ethene) | Quartz flat cell flow reactor (internal gap is 0.5 mm in the flat section, length-50 mm, width-8mm) | 31 | 97 | 87 | This Work |
| Pd-Mg/GS (Pd:3 %, 20 mg) | 1.8 (785 nm) | $C_2H_2/H_2/N_2$ = 5/15/75 (Without ethene) $C_2H_2/H_2/N_2$ = 5/20/75 (Without ethene) | Horizontally-oriented packed bed reactor with CaF₂ window (6 mm) at the top of the reactor | NR 101 | 90 NR | 80 NR | 37 |
| DPC/Ni (Ni-10 %, 35 mg) | 0.58 (400–1100 nm) | $C_2H_2/H_2/Ar$ = 0.12/2/97.8 (14 mL min⁻¹) (Without ethene) | Quartz flat cell flow reactor (tube i.d-3.5 mm) | 2.4 | 30 | 86 | 16 |
| Au-Fe/C (1%, Au:Fe=1:1, 1g (in 2 g quartz sand)) | 0.45 (250–1100 nm) T ~130 °C, external heating) | $C_2H_2/C_2H_4/H_2/Ar$= 1/20/20/59 (Total flow- 20 mL min⁻¹) (With excess ethene) | Photothermal fixed-bed reactor | 0.5 | 98.4 | 97.5 | 35 |
| AlNC-Pd (catalyst weight unknown) | 14.3 (680–1080 nm) | $C_2H_2/H_2/He/N_2$ = 1.33/3.33/25.33/70 (15 mL min⁻¹) (Without ethene) | Stainless steel gas-phase high-temperature reaction chamber | 26.7 ×10⁻³ (mmol h⁻¹) | 5 | 97 | 20 |

NR-not reported, vol.-volume

from the differential of the Arrhenius plot. The reaction rate in the presence of light was higher than that in the dark in the low Ts region and became comparable at higher Ts (1000/Ts ~ 1.9).

To elucidate the contribution of non-thermal and photothermal effects, the acetylene semi-hydrogenation in excess ethene was carried out at various light intensities along with external heating. The rate of conversion of acetylene over DPC/RuPt-10-Calc was plotted as a function of Ts under visible light illumination (Fig. 4f) (400–1100 nm), and the reactant gas flow of 110 mL min⁻¹ (30 mL min⁻¹ $C_2H_2$ (10% in Ar), 60 mL min⁻¹ $C_2H_4$, 15 mL min⁻¹ $H_2$, 5 mL min⁻¹ air). The light intensities were varied from 0.2 to 3 W cm⁻² and $T_R$ was varied from 80 to 200 °C. The Ts value was measured at different light intensities for a constant $T_R$ (Fig. S36). Ts value was determined by the contributions from the reactor's heat supply and photothermal heating by light. It was observed that at similar Ts in the low-temperature range, the rate of acetylene conversion was higher when the light intensity was higher. For instance, the Ts reached 95 °C when $T_R$ was set to 80 °C, and the catalyst was illuminated with a visible light of the intensity of 1 W cm⁻². The same Ts (95 °C) was obtained by setting the $T_R$ to 110 °C and illuminating the catalyst with low-intensity (0.2 W cm⁻²) light. If the catalysis proceeds via only the photothermal pathway, similar conversion rates should be observed in both conditions. However, the reaction rate under a higher light intensity (1 W cm⁻²) was double that under a lower light intensity (0.2 W cm⁻²). This trend was not observed at higher temperatures (>125 °C) as the maximum acetylene conversion rate reached ~400 mmol g⁻¹ h⁻¹ at this high GHSV (1320000 mL g⁻¹ h⁻¹). The comprehensive evaluation of temperature and light intensity variation showed the importance of non-thermal effects, which predominated at lower light intensities. It is difficult to disentangle the contribution of thermal and non-thermal effects at higher catalyst bed temperatures (which can be reached at higher light intensities) where the reaction reaches saturation in terms of production rate due to limited acetylene adsorption.

**Role of air in enhancing catalyst stability**

The DPC/RuPt-10-Calc catalyst underwent continuous deactivation during the semi-hydrogenation of acetylene without flowing air (Figs. S12, S25, 1h). Surprisingly, the catalyst was stable when the air was flown with the reactant feed, and the initial activity could be maintained for at least 100 h (Fig. 1g). To get insights into the air-enhanced stability, STEM and EDS analyses of the spent catalysts were performed, and no change in the RuPt NPs dispersion was observed (Fig. S37–41). The reason for the deactivation in the absence of air could be then thought of as the formation of coke, which blocks the active sites. However, Raman spectroscopic analysis of this spent catalyst showed no peaks corresponding to graphitic carbon: 1370 cm⁻¹ (D-band) and 1590 cm⁻¹ (G-band)[77] (Fig. S42), indicating no graphitic coke formation. TGA of the spent catalyst (Fig. S43) also showed no weight loss, confirming no coke formation. The TGA analysis, however, showed a ~ 2% weight gain in the range of 400–700 °C, which could be due to the re-oxidation of in-situ reduced metal sites.

XRD analysis of fresh catalyst showed the peaks corresponding to $RuO_2$ for the calcined sample (Fig. 5a), which, however, were absent in both the spent catalysts (with or without flowing air), indicating the in-situ reduction of $RuO_2$, which was also indicated by the slight increase in the peak intensity corresponding to Ru-hcp (Fig. S44). To further investigate the oxide phase composition of the spent catalysts, the EXAFS spectra and XPS were recorded for fresh and spent catalysts (Fig. 5, S45–48, Table S5). Comparing Ru K-edge XANES spectra and Fourier-transformed EXAFS of the catalysts with those of reference metal and metal oxide, an in-situ reduction of $RuO_2$ during the reaction was indicated. The role of air was found to preserve a small percentage of Ru-O linkages (Table 2). The oxidation state of Ru in spent catalysts (in the absence of air) by XPS showed a complete reduction of $RuO_2$ to

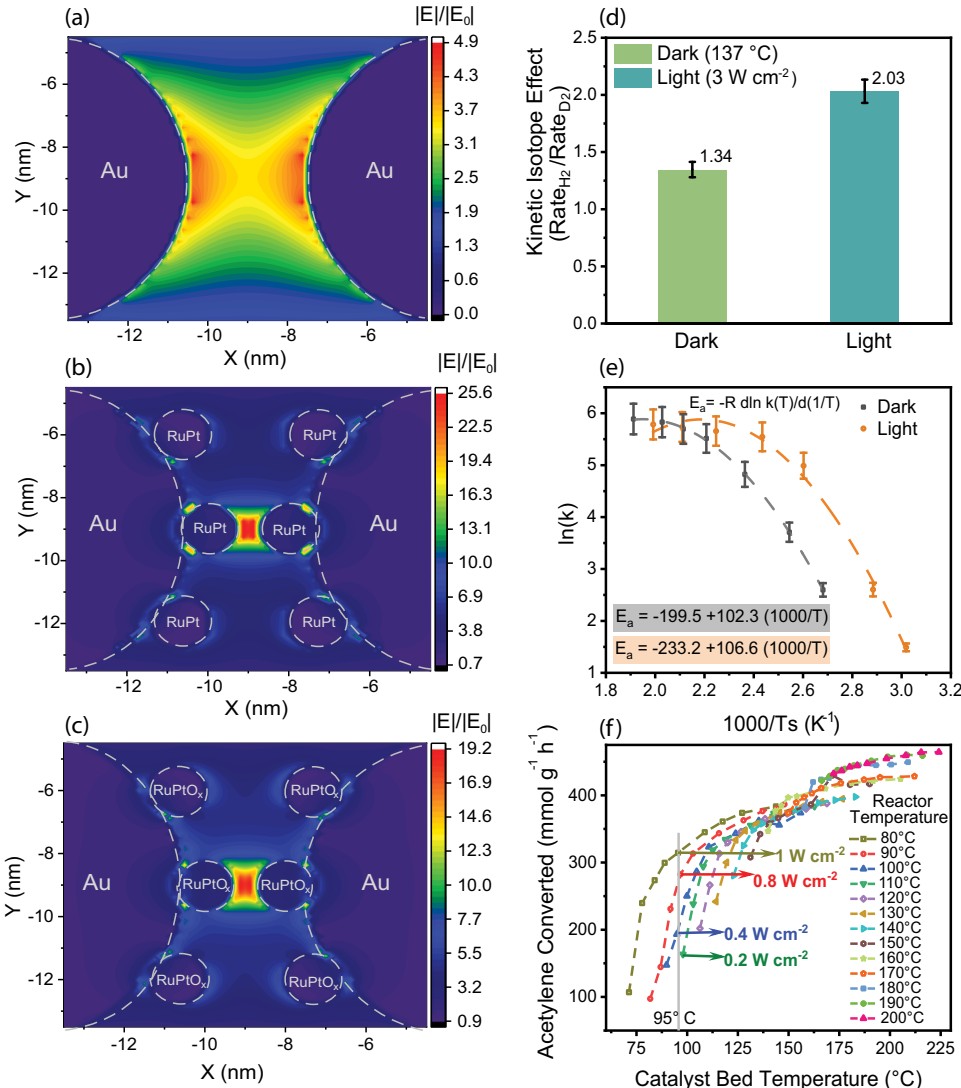

**Fig. 4 | Thermal and non-thermal activation mechanism.** Electric field enhancement in **a**) DPC, **b**) DPC/RuPt (fully reduced), and **c**) DPC/RuPtOx (partially reduced) (Source $E_0$ = 4509 V m⁻¹ at 2.7 W cm⁻²) using FDTD simulations; **d** KIE for acetylene semi-hydrogenation in light (3 W cm⁻²) and in the dark at corresponding catalyst bed temperature; **e** Arrhenius plots for activation energy calculation in light and dark at different Ts; **f** Acetylene semi-hydrogenation carried out at different light intensities along with external heating, using DPC/RuPt-10-Calc. Error bars: Calculated from data of at least three repeated experiments (Source data are provided as a Source Data file).

Ru in the deactivated catalyst, whereas only partial reduction was observed for the spent catalyst in the presence of air (Fig. 5c, d).

To know if the temperature reached by light illumination of the plasmonic catalyst was sufficient to reduce the oxide, we carried out the TPR of DPC/RuPt-10-Calc in the presence of $H_2$ (20 % in Ar). A reduction occurred at ~130 °C (Fig. S49), which was similar to the Ts achieved by illuminating light. This indicated the possibility of light-induced metal oxide reduction during photocatalysis. Light-induced reduction analysis of the metal oxide phase was also conducted using light-programmed reduction (LPR) to confirm that the loss of activity was associated with the in-situ catalyst reduction. The mass detector signal for $H_2$ decreased as soon as the light (2.7 W cm⁻²) was switched on, indicating a light-induced reduction of the oxide phase (Fig. 5e). This confirmed the possibility of metal oxide reduction to metal during plasmonic catalysis. $H_2$ oxidation took place, yielding water, one of the by-products of the reaction due to continuous regeneration (oxidation and reduction) of active sites (Ru to $RuO_2$) during the catalysis. No other oxidation products like CO and $CO_2$ were observed. It was evident from these studies that the Ru-O phase was crucial for the catalytic activity of the DPC/RuPt catalyst, and air in the reactant feed

preserved the Ru-O phase. Recently, Ramirez et al. [78] reported a similar deactivation in acetylene semi-hydrogenation over $In_2O_3$ at higher temperatures (>300 °C), which was attributed to the formation of O vacancies. They showed that $C_2H_2$ adsorbs more strongly than $H_2$ to the oxide phase to form the In-CH-CH-O complex and that only at higher temperatures $H_2$ interact to form hydroxy species, which later recombine to form water, leading to oxygen vacancy and, thus, deactivation. In the case of the DPC/RuPt catalyst, the Ru-O phase acted as an adsorption site for acetylene. However, it underwent light-induced complete reduction during the reaction in the absence of air, leading to the deactivation of the catalyst. Supplying air along with the reactant feed prevented this complete in-situ reduction of the Ru-O phase by re-oxidization. This simultaneous reduction and oxidation of the active site during the reaction were responsible for the high stability of the catalyst for at least 100 h.

A thorough analysis of the flammability of various components within the reactant feed was conducted to address the safety risk associated with introducing air into the reactant feed. Ideally, acetylene, ethene and hydrogen are highly flammable gases with lower flammable limit (LFL) and upper flammable limit (UFL) values of 2.5/

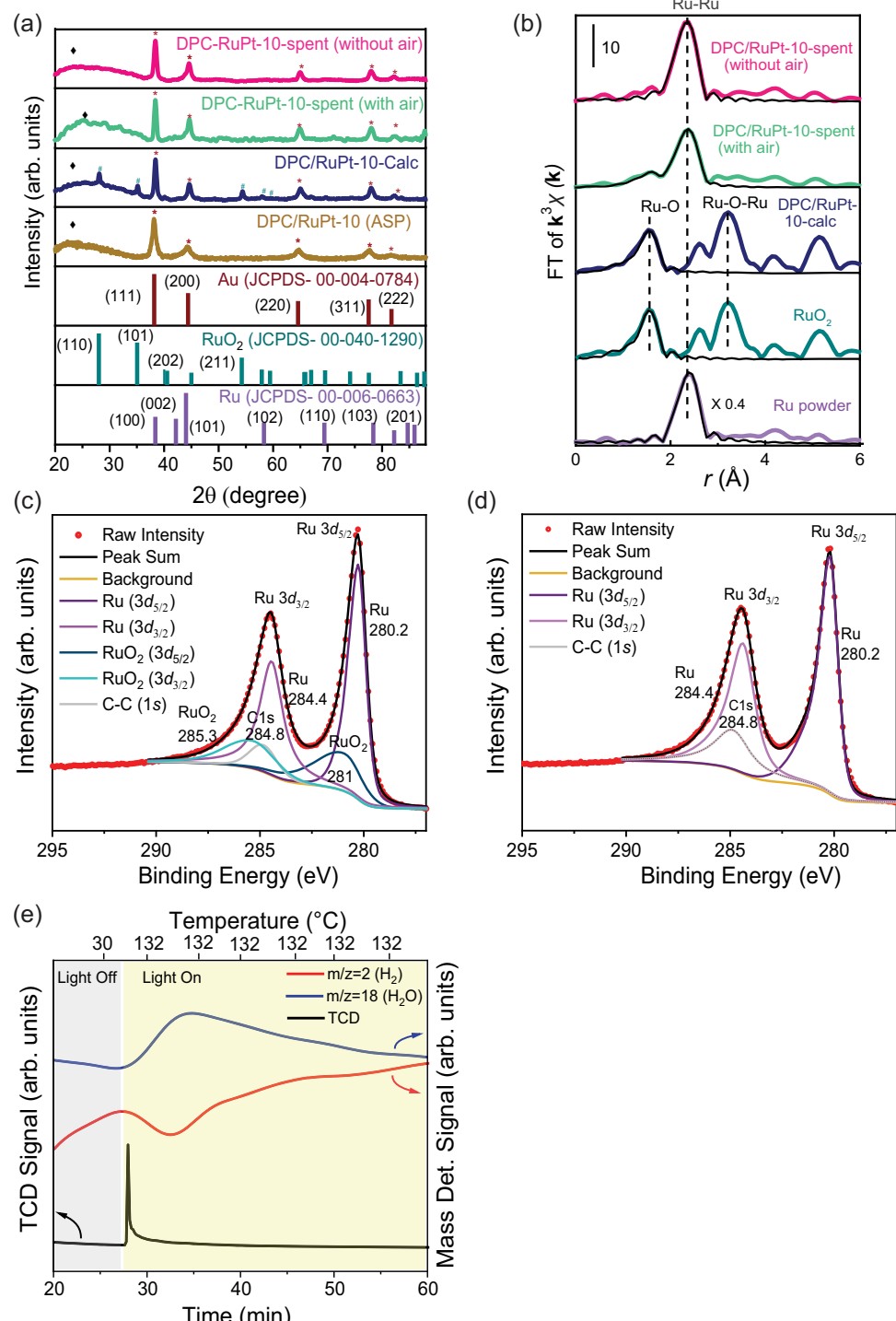

**Fig. 5 | Role of air in enhancing stability. a** XRD patterns of fresh and spent catalysts with or without flowing air; **b)** Fourier transform (FT) of Ru K-edge EXAFS of spent and fresh catalysts; XPS analysis for Ru 3$d$ of spent catalyst, **c)** with airflow, **d)** without airflow; **e)** Light-induced reduction of DPC/RuPt-10-Calc. (Source data are provided as a Source Data file).

100, 2.7/36 and 4/18.3 (vol% in the air) and can potentially explode in the presence of sufficient air. However, their ignition temperatures are 305, 490, and 500 °C respectively (Fig. S50)[79]. In our plasmonic catalysis, when operating under optimized conditions, we only reach a temperature of 132 °C, with only 1% $O_2$ present in the feed. As a result, there is no risk of an explosion. This highlights another advantage of using plasmonic catalysis, as it allows for the safe use of air even in the presence of highly flammable gases, which is not feasible with conventional thermal catalysis.

## Mechanism of acetylene semi-hydrogenation

In-situ diffuse reflectance infrared Fourier transform spectroscopy (DRIFTS) was carried out under light illumination with and without $H_2$ to unravel the mechanism of acetylene semi-hydrogenation over DPC/RuPt-10-Calc. As evident from in-situ DRIFTS (Fig. 6a), after saturation with $C_2H_2$, three peaks appeared at 1620, 1559, and 1476 cm$^{-1}$, assigned to C = C stretching, C−O stretching and =C−H bending modes of the di-σ-bonded acetylene respectively (Fig. 6a)[80]. Another peak at ~1731 cm$^{-1}$ was assigned to ethene π-bonded to surface oxygen, observed even

without supplying $H_2$. This observation highlights the significant role of surface hydroxyl in the first step of hydrogenation. The hydroxyl group was regenerated only after supplying $H_2$. The IR peaks due to gaseous ethene started to appear at ~3000 and 900 $cm^{-1}$, corresponding to the $=C–H$ stretching and $H–C–H$ out-of-plane wagging modes, respectively. Interestingly, the characteristic peak at 1255 $cm^{-1}$ corresponding to di-σ-bonded ethene[80,81] was absent even after supplying $H_2$, indicating that ethene was weakly bound to the catalyst surface via π orbitals. The weak interactions caused efficient deso-

### Table 2 | Structural parameters of fresh and spent catalysts obtained by Ru K-edge EXAFS curve fitting analysis

| Sample | Bond | Coordination Number | Bond length, $r$ (Å) | Debye-Waller factor, $\sigma$ (Å²) | R (%)[a] |
|---|---|---|---|---|---|
| Ru powder | Ru–Ru | 12.3(0.5) | 2.67(1) | 0.0048(0) | 14.7 |
| RuO$_2$ | Ru–O | 1.8(9) | 1.93(3) | 0.0014(4) | 11.1 |
| | Ru–O | 4.1(5) | 2.01(2) | 0.0022(0) | |
| DPC/RuPt-10-calc | Ru–O | 2.7(1.0) | 1.94(3) | 0.0044(1) | 12.0 |
| | Ru–O | 4.3(5) | 2.03(3) | 0.0053(0) | |
| DPC/RuPt-10-spent (with air) | Ru–O | 1.6(1.2) | 1.98(2) | 0.0076(60) | 5.0 |
| | Ru–Ru | 6.3(2) | 2.67(1) | 0.0067(5) | |
| | Ru–Pt | 1.9(1.0) | 2.65(3) | 0.0053(16) | |
| DPC/RuPt-10-spent (without air) | Ru–Ru | 6.6(1.7) | 2.67(1) | 0.0050(5) | 11.3 |
| | Ru–Pt | 1.6(1.0) | 2.64(3) | 0.0037(30) | |

[a]$R = (\sum(k^3\chi^{data}(k) - k^3\chi^{fit}(k))^2)^{\frac{1}{2}}/(\sum(k^3\chi^{data}(k))^2)^{1/2}$ Figures in parentheses show errors.

rption of ethene from the catalyst surface, resulting in high selectivity towards ethene by suppressing further hydrogenation. In-situ FTIR (Fig. 6b) showed a series of spectra captured every 30 s during $C_2H_2$ exposure (violet spectra) and subsequent reaction after $H_2$ was introduced (orange spectra) over the DPC/RuPt-10-Calc at 200 °C in transmission mode. The peak at ~1750 $cm^{-1}$ due to π- bonded ethene increased along with the peaks for gaseous acetylene in the initial adsorption periods (Fig. 6c). Once the surface was saturated with acetylene and $H_2$, the peaks for gaseous ethene started to appear, while those for gaseous acetylene peaks began to decrease owing to the conversion of acetylene selectively to ethene (Fig. 6d). Because of their continual creation during the reaction, the peaks corresponding to π- bonded ethene and di-σ-bonded acetylene remained steady throughout the hydrogenation.

Based on these observations, we hypothesized a reaction mechanism for acetylene semi-hydrogenation over DPC/RuPt-10 (Fig. 7a). A partially oxidized RuPt alloy surface with a terminal hydroxyl group (as evidenced by the IR spectra in Fig. S51)[82] interacted with acetylene and generated di-σ-bonded acetylene via the loss of 2H from the hydroxyl groups and the breaking of the triple bond to form two C–O bonds (step **i**). The hydrogen lost from hydroxyls was then utilized to form a $=C–H$ bond by breaking a C–O bond (step **ii**). After both the C–O bonds were broken, the ethene formed interacted with the O in a π-bonded fashion (step **iii**). After supplying external $H_2$, which was dissociated by the Pt sites of the RuPt catalyst, the hydroxyls were regenerated, and the ethene molecules were released, and the catalyst became available for the next cycle (step **iv**). RuPt NPs could thus facilitate the dissociation and migration of H radicals across the surface in a controlled way (step **v**), which could then add to the activated di-σ-bonded acetylene at the $RuO_2$ phase, lowering the

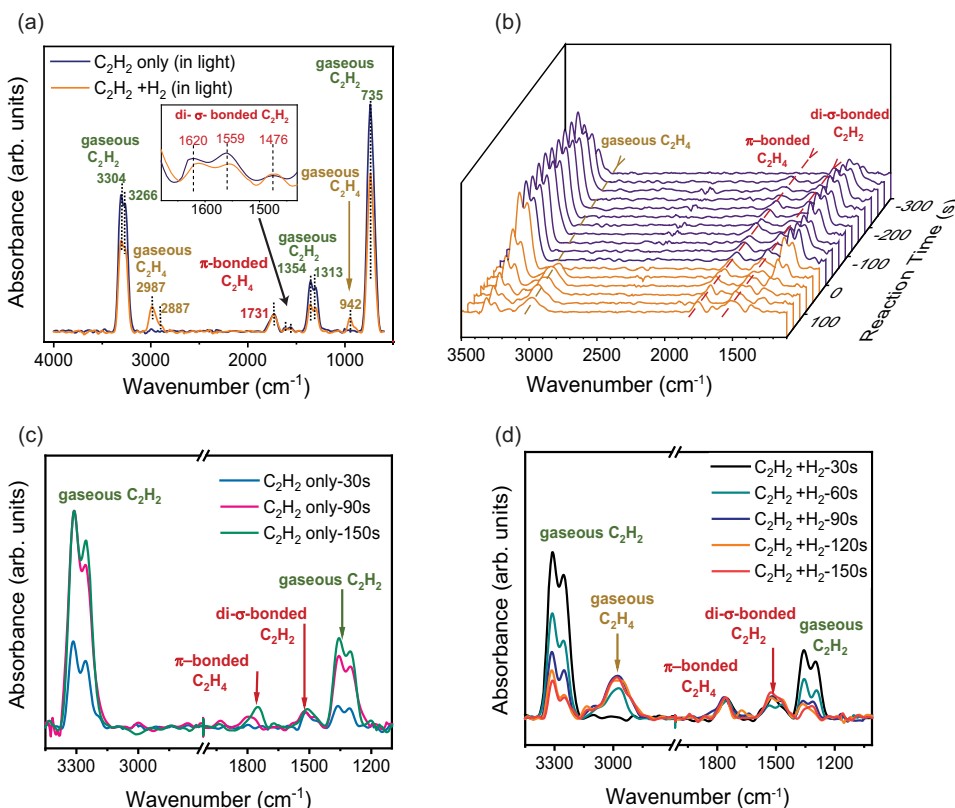

**Fig. 6 | In-situ FTIR study of acetylene semi-hydrogenation over DPC/RuPt-10-Calc. a** In-situ DRIFT spectra showing different intermediate formations during acetylene adsorption and hydrogenation driven by light; **b** Time-resolved in-situ FTIR spectra of the DPC/RuPt-10-Calc catalyst during $C_2H_2$ (violet) and subsequent $H_2$ treatment (orange) at 200 °C in transmission mode; enlarged view of the spectra while undergoing **c** $C_2H_2$ treatment, and **d** $H_2$ treatment. (Source data are provided as a Source Data file).

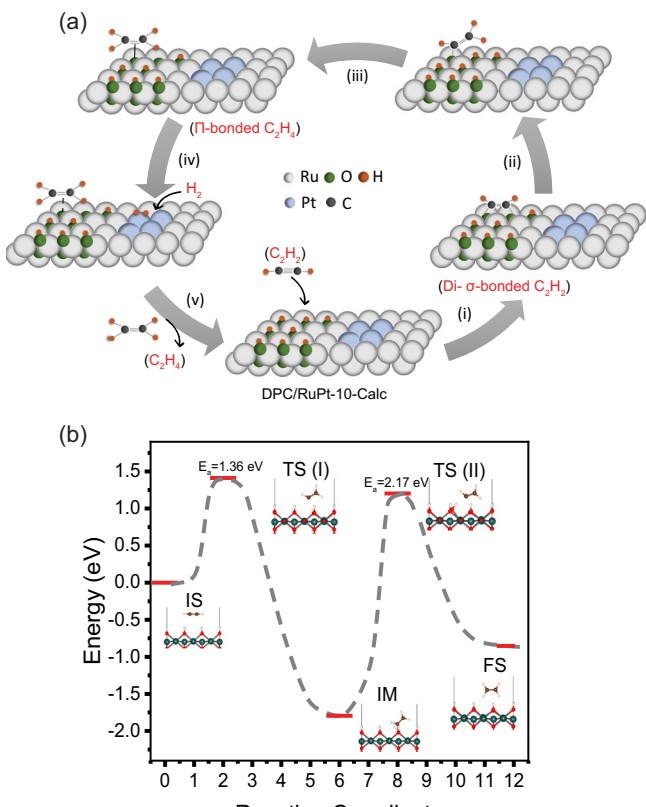

**Fig. 7 | Proposed reaction pathway for acetylene semi-hydrogenation on DPC/RuPt-10-Calc. a** Formation of various reaction intermediates showing the role of co-existing oxide and reduced species based on in-situ FTIR studies; **b** energies of initial (IS), transition (TS), intermediate (IM), and final (FS) states formed in the acetylene semi-hydrogenation by DFT calculation using nudged elastic band method. (Source data (7b) are provided as a Source Data file).

facilitating the C−O and O−H bond breaking to form TS(II) (Fig. 7), consequently leading to efficient desorption of hydrogenated product ethylene.

## Density functional theory calculations

The reaction pathway of acetylene semi-hydrogenation (Fig. 7a) was investigated using quantum chemical-based density functional theory calculations (Fig. 7b). First, the adsorption energy and stability of H atom at different adsorption sites were calculated (Fig. S52, Table S6). The HCP site at the Ru (110) surface was the most favourable site for H atom adsorption. With the RuPt surface, the HCP-bridge site had the highest adsorption energy, while the FCC-bridge and the central Pt-atom sites had equal energy. These results indicated that the Pt atoms of the RuPt catalyst had an important role in enhanced production rate and selectivity, consistent with the experiment. The dissociation of the $H_2$ molecule on both surfaces was energetically favorable process. The RuPt (110) surface showed lower energies than path (I) of the Ru (110) surface, indicating a vital role of Pt in the dissociation of $H_2$ molecules by providing extra charges (more $d$ electrons than the Ru atom) for dissociation (Fig. S53). Reaction energies corresponding to different migration paths for the H atom on Ru and RuPt surfaces were calculated (Fig. S54). The path $P_1$ on the Ru (110) surface involved the migration from one HCP site to another neighbouring HCP site through an intermediate FCC site. The three activation barriers along path $P_1$ were 0.13 eV, 0.02 eV, and 0.11 eV. Pt-loaded Ru catalyst showed a low activation barrier of 0.08 eV along path $P_2$ (from FCC-bridge site to HCP-bridge site), indicating the supporting role of Pt atoms in the catalytic reaction.

The interaction of $C_2H_2$ with $RuO_2$, followed by subsequent hydrogenation, was investigated (Fig. S55). There were three steps: first, the weak interaction between the surface and the molecule; second, the strong adsorption as the surface loses one hydrogen atom to form a C−O bond; and third, the weak adsorption between the product ethene and the surface (Fig. S56). Furthermore, the reaction path for the semi-hydrogenation process of σ-bonded acetylene to form ethene is shown in Fig. 7b. Two transition states were found to have an energy barrier of 1.36 eV and 2.17 eV. The first transition state, TS (I), involved the loss of a single hydrogen atom, forming a $C_2H_3$ intermediate. The dangling oxygen atom formed a strong C−O bond at the intermediate reaction state. The second transition state, TS (II), included the breaking of the C−O bond and the migration of a neighbouring H atom to form ethene, released from the surface and weakly adsorbed to the dangling oxygen atoms, consistent with experimental observations.

As per DFT calculations (Fig. 7b), ethene desorption seems to be the RLS in the dark. However, ethene desorption is dependent on the availability of dissociated hydrogen, which is enhanced by transient charge transfer to $H_2$ via Pt. It is evident from the DFT calculations that charge gets accumulated on Pt in the dark (Fig. S57), which will be even higher in light. The higher KIE observed for $H_2/D_2$ in light (as illustrated in Fig. 4d) also indicated enhancement in $H_2$ dissociation assisted by high-energy electrons. Thus, the polarizing electric field as well as dissociated hydrogen in light, accelerates the ethene desorption. Based on these observations, we propose that the reaction follows the Menzel−Gomer−Redhead (MGR)-type mechanism[84].

In conclusion, a hybrid plasmonic catalyst where active RuPt sites were loaded onto 'black gold' for photocatalytic semi-hydrogenation of acetylene in excess ethene was demonstrated. This catalyst showed excellent catalytic performance for competitive acetylene semi-hydrogenation in the presence of excess ethene with only visible light illumination and no external heating. With an ethene production rate of 320 mmol g$^{-1}$ h$^{-1}$, and ~90% selectivity, the DPC/RuPt-10-Calc catalyst outperformed all reported plasmonic catalysts. Better performance of DPC/RuPt-10-Calc than both of its monometallic counterparts indicated the synergy between Ru and Pt to maintain a high

overall activation energy barrier, also confirmed by the DFT calculations (discussed in the next section). Without the $RuO_2$ phase, there would not be such an activation of acetylene. Moreover, there is an expected decrease in the extent of acetylene adsorption on the DPC/RuPt due to the negative charge accumulation on the metal surface and the π electron cloud, which may be the reason for the decreased stability of the reduced catalyst.

In the dark case, larger KIE was observed at lower temperatures, suggesting the $H_2$ dissociation or acetylene hydrogenation as RLS at lower temperatures (Fig. S34). Whereas, the desorption of ethene seems RLS at higher temperatures since almost no KIE was observed at 200 °C. On the other hand, in the presence of light, KIE was seen at any light intensity (or temperature range) (Fig. S35), indicating that the desorption of ethene seems not the RLS under light irradiation. Based on these results, light irradiation promotes the desorption of ethene; thus, the $H_2$ dissociation or acetylene hydrogenation step is the RLS under light irradiation.

We believe that the ethene desorption step is accelerated by a polarizing electric field on the catalyst surface, indicating the role of non-thermal effects in enhancing the RLS. Hu et al.[83] made a similar observation, where they observed that an electric field pointing toward the plasmonic metal surface redistributed the electrons of $CO_2$ from the O lone-pair orbital to its antibonding π* orbital, weakening the C−O bond and increasing the bond length. When the electric field pointed away from the metal surface, the lone-pair electrons transferred to the σ orbital between O and the metal atom, making the desorption of the product facile. Such polarizing effect of the enhanced electric field around the active site might be happening in our catalytic system,

production rate and selectivity towards ethene. This plasmonic reduction catalyst was stable only when the air was flown together with the reactant feed, and the activity could be maintained for at least 100 h. Supplying air along with the reactant feed prevented the complete in-situ reduction of the metal oxide by simultaneously oxidizing some of the reduced active metal sites, which played a key role in acetylene chemisorption and, in turn, better catalytic performance. This plasmon-mediated simultaneous reduction and oxidation of the active site during the reaction were responsible for the good stability of the catalyst for at least 100 h. FDTD simulations showed a five-fold electric field enhancement in DPC/RuPt compared to pristine DPC, owing to the near-field coupling between the RuPt nanoparticles and DPC. This electric field, predominantly concentrated around the RuPt sites within the gaps of Au NPs was crucial in activating the chemical bonds of acetylene as well as enhancing the rate-limiting ethene desorption step. Both non-thermal and thermal effects were noted to be influential, with non-thermal effects prevailing at lower light intensities and thermal effects taking precedence at higher light intensities. KIE measured in light was larger than in the dark, also implying the contribution from the non-thermal effects along with the photothermal activation of the reactants.

The in-situ DRIFTS and DFT studies provided insight into the reaction mechanism over the oxide surface and highlighted the role of the intermediates in determining the selectivity. Hydroxyl groups of a partially oxidized RuPt catalyst surface generated di-σ-bonded acetylene, followed by breaking the triple bond to form two C−O, which then reacted with hydrogen to generate =C−H bonds. This intermediate formed interacts with the O in a π-bonded fashion and reacts with dissociated hydrogen to yield the final product ethene.

In conclusion, this work introduces an 'air-stabilized' and 'plasmonically activated' catalyst for the semi-hydrogenation of acetylene, showcasing its promising potential for various reduction reactions.

## Methods

### Materials
Cetyltrimethylammonium bromide (CTAB, ≥99%; Sigma-Aldrich), Urea (≥99%; Sigma-Aldrich), Tetraethyl orthosilicate (TEOS, 98%; Sigma-Aldrich), p-Xylene (p-xylene for synthesis, 99%; Loba Chemie), 1-Pentanol (≥99%; Sigma-Aldrich), (3-Aminopropyl)triethoxysilane (APTS, 99%; Sigma-Aldrich), Toluene (anhydrous, 99.8%; Sigma-Aldrich), Deionized water (H$_2$O; Milli-Q System; Millipore), Ethanol (99.9%, Changshu Hongsheng Fine Chemical Co. Ltd.), Gold (III) chloride trihydrate salt (HAuCl$_4$.3H$_2$O, CDH Fine Chemicals), Sodium borohydride (NaBH$_4$, 99%, Sigma Aldrich), Potassium carbonate (K$_2$CO$_3$, ≥99%, TCI Chemicals), Ammonium hydroxide solution (25%, Merck Life Sciences Pvt. Ltd), Formaldehyde (37 wt% in H$_2$O, contains 10−15% methanol as stabilizer, Sigma Aldrich), Ruthenium chloride (RuCl$_3$ • nH$_2$O, Tanaka Precious Metals), Hydrogen hexachloroplatinate (IV) hexahydrate (H$_2$PtCl$_6$ • 6H$_2$O). Polyvinylpyrrolidone (PVP) (K-30), ethylene glycol (EG) and sodium hydroxide (NaOH) were obtained from Fujifilm Wako Pure Chemical Industries.

### Synthesis of Dendritic Fibrous Nanosilica (DFNS)
Dendritic Fibrous Nanosilica (DFNS)[44] was synthesized by adding CTAB (10 g, 27.43 mmol) and urea (12 g, 199.8 mmol) to a 2-L four-neck round-bottom flask (RB flask). Distilled water (500 mL) was added to the flask, and the mixture was stirred at 700 rpm for 30 min at room temperature (RT). p-Xylene (500 mL) was mixed with TEOS (100 mL, 447.8 mmol) and added dropwise to the RB flask. The reaction mixture was stirred for another 30 min at 700 rpm and RT. Subsequently, 1-pentanol (30 mL, 277 mmol) was added dropwise to the reaction mixture while stirring at RT for 30 min. The mixture was refluxed for 12 h under continuous stirring at 130 °C. Ethanol (200 mL) was added to it after cooling. The solid product was

isolated by centrifugation at 12298 x g for 10 min and washed three times with distilled water (100 mL) and then three times with ethanol (100 mL). The solid was then dried in an oven at 80 °C for 12 h. Finally, the dried solid was calcined in air at 550 °C for 6 h with a ramp of 5 °C min$^{-1}$ in a muffle furnace to remove the surfactant (Yield: ~22 g).

### Synthesis of Dendritic Plasmonic Colloidosome (Black Gold)
Dendritic plasmonic colloidosomes (DPC)[43] were synthesized by first functionalizing DFNS (4 g) by refluxing with APTS (4 mL, 17 mmol) in 250 mL of toluene at 80 °C for 24 h. The resultant solid was washed three times with 50 mL of toluene and three times with 50 mL of ethanol and then dried in an oven for 10 h at 80 °C to yield DFNS-APTS. DFNS-APTS (500 mg) was then dispersed in water (50 mL) using a sonicator for 15 min and stirred for an additional 10 min at room temperature. Gold-stock solution containing 100 mg mL$^{-1}$ of gold (III) chloride trihydrate salt (430 μL 0.109 mmol of HAuCl$_4$) was added dropwise. The reaction mixture was then sonicated for 15 min, followed by stirring for 2 h at room temperature. Freshly prepared NaBH$_4$ solution (5 mL, 1 M in water) was then added to this dispersion, and it was then stirred for two hours at room temperature. The solid was isolated using centrifugation at 12298 x g for 10 min, followed by three washings with distilled water (30 mL) and ethanol (30 mL). The resultant solid was dried at 80 °C in an oven for 2 h and was named DPC-C0, 0$^{th}$ cycle.

The K-gold solution was prepared by dissolving HAuCl$_4$.3H$_2$O (300 mg, 0.761 mmol) and K$_2$CO$_3$ (2800 mg, 20.26 mmol) in 2 L of DI water. DPC-C0 (500 mg) was then dispersed in 1000 mL of K-gold solution for the following C1 growth cycle. The solution was sonicated for 10 s and then stirred at room temperature for 10 min (200 rpm), followed by the addition of ammonium hydroxide (5 mL, 25%) and subsequent stirring for 15 min. Formaldehyde solution (90 mL, 37 wt% in H$_2$O) was then added, and the solution was stirred for 24 h at room temperature. The solid product was isolated by centrifugation at 12298 x g for 10 min and washed three times with water (100 mL each time) and ethanol (100 mL each time). These growth steps were repeated, taking DPC-C1 as base material to obtain DPC-Cx with 2, 3, and 4 growth cycles. The resultant solid obtained after the 4$^{th}$ cycle, DPC-C4, was named as "black gold", and used as the plasmonic catalyst support over which RuPt nanoparticles (NPs) were loaded (Yield: ~75%).

### Synthesis of RuPt Clusters
Ru, Pt, and RuPt bimetallic clusters[42] were synthesized by using an ethylene glycol (EG) reduction process. For synthesizing Ru NPs, RuCl$_3$•nH$_2$O (1.0 mmol, 261 mg) and PVP (20 mmol, 2.22 g) were dissolved in EG (10 mL/mmol$_{PVP}$) and the mixture was stirred at 80 °C under Ar flow for 1 h. The mixture was heated to 180 °C and stirred for 1 h in an Ar environment. PVP-stabilized Pt NPs (denoted as Pt) were also synthesized by the EG reduction method. NaOH (50 mmol, 2.00 g) and PVP (20 mmol, 2.22 g) were dissolved in EG (10 mL/mmol$_{PVP}$) and the H$_2$PtCl$_6$•6H$_2$O (1.0 mmol, 518 mg) was added to the solution, which was then stirred at 80 °C under Ar flow for 1 h. The mixture was further stirred for 3 h at 140 °C in an Ar environment. The same approach of Ru NPs synthesis was used to synthesize RuPt bimetallic NPs using H$_2$PtCl$_6$ • 6H$_2$O (0.10 mmol, 51.8 mg), RuCl$_3$•nH$_2$O (0.90 mmol, 235 mg) and PVP (20 mmol, 2.22 g). The solution eventually turned dark brown while being stirred at 180 °C. In all cases, the suspension was cooled to room temperature under stirring, and water (same amount as EG) was added to reduce viscosity. The solution was added to the centrifuge tube equipped with a membrane filter with a cut-off molecular weight of 10 kDa (20 mL volume for each tube, Vivaspin 20) and reduced to less than 5 mL. The colloidal metal NPs were deionized three times with water (10 mL) before being collected as a powder by lyophilization (Yield: ~85 % for Ru:PVP and RuPt:PVP; ~80 % for Pt:PVP).

## Synthesis of DPC/RuPt

For a typical 10 wt% loading of RuPt over black gold, RuPt bimetallic NPs (75 mg) (effective metal content-2% (Table S1)) were dispersed in ethanol (15 mL) in 250 mL RB flask and sonicated for 90 min. Black gold (DPC-C4, 15 mg) was then added to this dispersion and again sonicated for 30 s. The mixture was then stirred at 60 °C under ambient conditions for 1 h and dried under vacuum at 80 °C for 2 h. Finally, the resultant powder (DPC/RuPt-10 (ASP)) was calcined in a muffle furnace at 800 °C (10 °C min⁻¹ ramp) for 2 h to oxidize the as-prepared sample to DPC/RuPt-10-Calc. (Yield: ~85%).

## Catalysts Characterizations

Scanning transmission electron microscopy (STEM) analysis was carried out using FEI-TITAN operated at an accelerating voltage of 300 kV. Elemental mapping was performed using energy-dispersive X-ray spectroscopy (EDS). A small amount of solid powder was dispersed in ethanol by sonicating for 30 s, and the dispersion was drop-casted onto a holey carbon-coated 200 mesh copper TEM grid. Aberration-corrected high-angle annular dark field scanning transmission electron microscope (HAADF-STEM) images were collected using a JEM-ARM200F microscope operated at an acceleration voltage of 200 kV. HAADF-STEM samples were prepared by drop casting the ethanol dispersion of the catalysts onto a thin carbon layer-coated copper grid (SHR-C075, Okenshoji). PXRD patterns were obtained using a PANalytical X'Pert Pro powder X-ray diffractometer with Cu-K radiation. A JASCO UV/vis/NIR spectrophotometer was used to conduct UV-Vis spectroscopic measurements. The UV-Vis extinction spectra of DPC and DPC/RuPt-10-Calc were recorded by dispersing them in ethanol using sonication for 10 s. The baseline subtraction process was carried out (to reduce potential scattering effects) by first recording the UV-Vis extinction spectra of DFNS/APTS dispersion in ethanol (which was the solid support used to prepare black gold) and then subtracting it from the UV-Vis data of black gold dispersion in ethanol. $N_2$ sorption measurements were performed using a Micromeritics 3-Flex surface analyzer (samples were degassed at 120 °C overnight under vacuum before analysis). The weight loss analysis, to study PVP removal (for fresh catalyst) as well as coke-formation (in after catalysis sample) was carried out by thermogravimetric analysis, using Mettler Toledo TGA-DSC2/LF/1100, from 30 to 1000 °C (ramp- 10 °C min⁻¹) in the airflow of 40 mL min⁻¹.

XPS analysis was carried out using a Thermo Kα+ spectrometer with micro-focused and monochromated Al-Kα radiation (1486.6 eV) as the X-ray source. The sample was prepared by sprinkling solid powder on carbon tape. The carbon signal at 284.8 eV was used as an internal reference. Raman measurements were performed at 633 nm using a Witec alpha300R confocal Raman microscope. The temperature-programmed reduction, light-programmed reduction and Kinetic Isotope Effect (KIE) measurements were conducted using a Catalyst Analyzer BELCAT II coupled with a Quadrupole mass spectrometer (Belmass). For $H_2$-TPR, DPC/RuPt-10-Calc (30 mg) sample was loaded into a quartz reactor and was exposed to a 20.0 vol% $H_2$/Ar mixture (25 mL min⁻¹) and heated to 700 °C at a rate of 5 °C/min. For KIE measurements, 30 mg of catalyst was loaded and gases ($C_2H_2$/$H_2$ or $D_2$/Ar=0.3/3/29.7) were introduced and the reaction was carried out at different temperatures and light intensities. The product was quantified by Belmass.

XAS was conducted in the transmission mode for Ru K-edge and Au $L_3$-edge using ion chambers for the $I_0$ and $I_I$, and the fluorescence mode for Pt $L_3$-edge using an ion chamber for the $I_0$ and 19 solid state detectors (SSDs) for $I_I$ detector in the BL14B2 beamline at SPring-8 of the Japan Synchrotron Radiation Research Institute (proposal number 2023B1608). The incident X-ray beam was monochromatized by a double-crystal monochromator of Si (111) for Au and Pt $L_3$-edge and Si (311) for Ru K-edge. X-ray energy was calibrated using Pd foil for Ru K-edge and Au foil for Au or Pt $L_3$-

edge. EXAFS data were analyzed by software REX2000 software (Rigaku Co.). The $\mathbf{k}^3$-weighted $\chi$ spectra in the $\mathbf{k}$ range of 3–14 Å⁻¹ were Fourier-transformed into the $r$ space. The curve-fitting analysis was conducted over the $r$ range of 1.0–3.1 Å for the Ru K-edge and 1.5–3.2 Å for the Au $L_3$-edge.

## Plasmonic Acetylene Semi-Hydrogenation using DPC/RuPt

Photocatalytic acetylene hydrogenation was carried out in a PIKE technologies flow reaction chamber with a quartz window equipped with a heater and a thermocouple to precisely measure the temperature of the catalyst bed and connected it to a temperature controller (Fig. S2). The inlet of the flow reactor chamber was connected to mass flow controllers (MFCs), and the outlet was connected to an Agilent 490 MicroGC equipped with CP-PoraPLOT U column and a thermal conductivity detector (TCD).

The catalyst (5 mg) was taken in a ceramic porous base crucible, which was placed inside the reactor chamber. Argon (Ar) gas (150 mL min⁻¹) flowed through the reactor for 10 min, and the reactant gases were then introduced into the reactor chamber through Alicat mass flow controllers; $C_2H_2$ (10 % in Ar, 30 mL min⁻¹), $H_2$ (15 mL min⁻¹), $C_2H_4$ (60 mL min⁻¹) for competitive and Ar (55 mL min⁻¹) for non-competitive along with air (5 mL min⁻¹) constituting a total flow of 110 mL min⁻¹ (for competitive) and 105 mL min⁻¹ (for non-competitive) at 1 bar pressure. The catalyst was then irradiated with light (300 W Xenon Lamp ~2.7 W cm⁻², 400–1100 nm), and the progress of the reaction was monitored by using online MicroGC every 4 min. Higher temperature studies in the dark were performed by providing external heating to the catalyst bed by the heater inside the reaction chamber. For tests under different light intensities, the light power was tuned by changing the light intensity of the Xenon lamp (Fig. S18). The wavelength-dependent studies were conducted using ThorLabs diode lasers of various wavelengths, keeping the light intensity constant (Fig. S22). For quantification, the GC was calibrated by injecting known concentrations of standard gases like $H_2$, $O_2$, $N_2$, $CO_2$, $CH_4$, $C_2H_4$, $C_2H_6$, $C_2H_2$, and higher hydrocarbons till $C_4$. The slope of peak area versus ppm plot gives the calibration constant (area/ppm), which was used to calculate the product formation rate and selectivity of the products formed. Production rate, selectivity, conversion, and apparent activation energy were calculated using the below formulae,

$$C_2H_2\ Conversion(\%) = \frac{(C_2H_{2in} - C_2H_{2out})(in\ ppm) \times 100}{(C_2H_{2in})(in\ ppm)} \quad (1)$$

$$C_2H_4\ Production\ Rate\left(mmol\ g^{-1}h^{-1}\right) = \frac{\begin{array}{c}(C_2H_{2in} - C_2H_{2out}) - (C_2H_{6out} - C_2H_{6in})(in\ ppm)\\ \times Total\ flow\left(\frac{mL}{min}\right) \times 60\end{array}}{Wt.of\ catalyst\ (in\ g) \times 22400 \times 1000} \quad (2)$$

$$C_2H_6\ Production\ Rate\left(mmol\ g^{-1}h^{-1}\right) = \frac{\begin{array}{c}(C_2H_{6out} - C_2H_{6in})(in\ ppm)\\ \times Total\ flow\left(\frac{mL}{min}\right) \times 60\end{array}}{Wt.of\ catalyst\ (in\ g) \times 22400\ X\ 1000} \quad (3)$$

$$C_2H_4\ Selectivity(\%) = \frac{C_2H_4\ Production\ Rate}{C_2H_4\ Production\ Rate + C_2H_6\ Production\ Rate} \quad (4)$$

## Activation Energy Calculations

ln($k$) versus 1000/$Ts$ was plotted (Fig. 4e) and apparent activation energy ($E_a$) was calculated by using the Arrhenius equation[76],

$$E_a = -\frac{R\ dln\left(k_{C_2H_4}\right)}{d\left(\frac{1}{Ts}\right)}(J\ mol^{-1}) \quad (5)$$

where $k_{C_2H_4}$ is ethene production rate.

The rate was first obtained as a function of temperature by fitting the data to a polynomial function (Fit shows ln$k$ is a quadratic polynomial in terms of $1/Ts$).

In Light: $-24.78 + 28.04 (1000/Ts) - 6.41(1000/Ts)^2$
In Dark: $-17.51 + 24 (1000/Ts) - 6.15 (1000/Ts)^2$

Differentiating the obtained expression, $Ea$ can be obtained to have a linear dependence on $1/Ts$ and the exact expressions are given below and plotted in Fig. S35.

In Light: $-233.2 + 106.6 (1000/Ts)$
In Dark: $-199.5 + 102.3 (1000/Ts)$

The plot of Fig. S35 is a differential plot of Fig. 4e.
The Supplementary Data 1 contains the detailed calculation.

## Quantum Efficiency Calculations

Quantum Efficiency is defined as:

$$Quantum\ Efficiency(\%) = \frac{C_2H_2\ molecules\ reacted\ per\ unit\ time \times 100}{Number\ of\ photons\ absorbed\ by\ DPC/RuPt\ per\ unit\ time} \quad (6)$$

The number of photons absorbed by DPC/RuPt per unit time was calculated by the absorption spectra of DPC/RuPt and emission spectra of the Asahi Xenon Lamp used.

$$N_{photon} = \sum_{\lambda=400nm}^{1100nm} \frac{Light\ Intensity \times I\% \times A\% \times Illumination\ area \times Time}{Average\ single\ photon\ energy\ (E_{photon}) \times N_A} \quad (7)$$

In photocatalytic acetylene semi-hydrogenation by DPC/RuPt, light intensity was varied over the area spanning the catalyst crucible of diameter ~0.47 cm. $I\%$ is the percentage of the light intensity of Xe-lamp at a specific wavelength and $A\%$ is the absorption percentage of the catalyst at that specific wavelength. Time was taken as 1 h (3600 s) and $N_A$ is the Avogadro constant.

The average single photon energy is given by:

$$E_{photon} = \frac{hc}{\lambda} \quad (8)$$

In this equation, $h$ is Planck's constant $6.626 \times 10^{-34}$ m$^2$ kg s$^{-1}$, $c$ is the speed of the light in vacuum ($3 \times 10^8$ m s$^{-1}$), and $\lambda$ is the wavelength of the photon.

The Supplementary Data 2 contains the detailed calculation.

## Finite Difference Time Domain (FDTD) Simulations

The electric field enhancement calculations were performed by the finite difference time domain method. For simulation, the Au NPs were modelled as spheres of 10 nm in diameter. We modelled 2 Au NPs at a distance of 3 nm from each other. The Au NPs were placed on a silica sheet of 5 nm thickness. This model was chosen to mimic DPC-C4, in which Au NPs are located close to each other. For the simulation of DPC/RuPtO$_x$, spherical RuPt NPs (1.5 nm diameter) were attached to Au NPs. The composition of the NPs was assumed to be in 0.1:0.6:0.3 (Pt:Ru:RuO$_2$) molar ratio. DPC/RuPt was also modeled with 0.1:0.9 (Pt:Ru) molar ratio. The Au NPs were randomly decorated with these NPs to mimic the surface of DPC/RuPt-Calc. An x-polarized total-field scattered-field (TFSF) source having a wavelength range of 400 nm to 1100 nm and E$_0$ of 4509 V m$^{-1}$ (for light intensity- 2.7 W cm$^{-2}$) was used as the excitation source to mimic the photocatalysis conditions. E$_0$ was varied according to different light intensities (0.5 W cm$^{-2}$, 1 W cm$^{-2}$, 2.7 W cm$^{-2}$). Frequency domain field profile monitors were used to calculate the electric field distribution in all the simulations. The

dielectric constants of SiO$_2$, Au, and Pt were taken from the Handbook of Optical Constants of Solids, Ed. E. D. Palik, (Academic Press, 1985) and the refractive indices of RuPt NPs were calculated by following the Effective Medium Approximation[85] based on the Drude model where the permittivity of the composite was assumed to be the sum of individuals (Ru[86], Pt, and RuO$_2$[87]) multiplied by their mole fraction.

## In-situ DRIFT Study

Diffuse reflectance infrared Fourier transform spectroscopy (DRIFTS) measurements were performed on a JASCO FT/IR-4700 instrument with a DiffusIR™-PIKE Technologies high-temperature reaction chamber with KBr windows. The catalyst (5 mg) was taken in a ceramic porous base crucible, which was placed inside the reactor chamber. 10 % C$_2$H$_2$ (in Ar) was filled in the reactor by flowing 50 mL min$^{-1}$ gas for 15 min, the light of intensity 2 W cm$^{-2}$ was shone and the spectra were recorded and averaged out using 2400 scans and with a resolution of 4 cm$^{-1}$. Then, the gas with a composition 10 % C$_2$H$_2$-50 mL min$^{-1}$ and H$_2$-25 mL min$^{-1}$ was filled in the reactor by flowing for 15 min and the spectra were recorded under similar conditions. The spectra were recorded against the baseline of the Ar filled reactor (50 mL min$^{-1}$) in light.

## Time-Resolved In-Situ Transmission FTIR Study for Acetylene Semi-Hydrogenation

Time-resolved operando Fourier transform infrared (FTIR) spectroscopy measurements were carried out to study reaction intermediates and products of acetylene hydrogenation over DPC/RuPt-10-Calc. In-situ FTIR experiments were carried out using a Specac high-temperature transmission IR reaction cell with ZnSe windows and JASCO FT/IR-4700 equipment. 20 mg of the catalyst was pressed into a pellet of about 13 mm in diameter. This self-supporting catalyst pellet was then inserted in Specac's high-temperature transmission IR reaction cell. Ar (50 mL min$^{-1}$) flowed through the reactor at 200 °C, and a baseline was taken with 36 scans and 4 cm$^{-1}$ resolution. To gain insights into the evolution of surface species over the DPC/RuPt catalyst, time-resolved experiments were performed by first saturating the catalyst with C$_2$H$_2$ flow (10 % in Ar) of 50 mL min$^{-1}$ at 200 °C and then flowing H$_2$ at a rate of 25 mL min$^{-1}$ for sufficiently long times to stabilize the transmission FTIR signal.

## Computational Methods

The dissociation and migration of hydrogen on the RuPt (110) surface and the conversion of acetylene into ethene on the RuO$_2$ (110) surface were investigated using the density functional theory calculations as employed in the Vienna ab initio Simulation Package (VASP)[88] package. For the exchange-correlation functional, the Perdew-Burke-Ernzerhof of generalized gradient approximation was used with a cutoff energy of 400 eV for the plane-wave expansion. A vacuum of 15 Å was added to avoid artificial interactions due to periodic boundary conditions. The calculations continued until the forces dropped below 0.002 eV/Å and energy below $10^{-4}$ eV. The van der Waals correction of Grimme (DFT-D3)[89] was considered for the weak interactions between the adsorbates and the surfaces. The reaction barriers were calculated using the climbing image nudged elastic band (CI-NEB) method[90].

The adsorption energy of H$_2$ molecule was calculated using,

$$E_{dH_2} = E_{total} - E_{surf} - E_{H_2} \quad (9)$$

where $E_{total}$, $E_{surf}$, and $E_{H_2}$ are the total ground state energy of the combined system, without the H$_2$ molecule and the isolated H$_2$ molecule, respectively. The adsorption energy of H atom is calculated using,

$$E_{dH} = E_{total} - E_{surf} - E_H \quad (10)$$

where $E_{total}$, $E_{surf}$ and $E_H$ are the total ground state of the combined system, without H atom, and the isolated H atom, respectively. The adsorption energy of acetylene and the reaction intermediate species is calculated using,

$$E_{ads} = E_{total} - E_{surf} - E_{mol} \qquad (11)$$

Where $E_{total}$, $E_{surf}$ and $E_{mol}$ are the total ground state energy of the combined system, without the molecule (or intermediate species), and of the isolated molecule (or intermediate species), respectively.

Four layers of Ru (110) surface were built using $3 \times 2$ supercell, where two bottom layers were kept fixed during structure optimization. After the optimization, 8 Ru atoms from the top layer were replaced with Pt atoms (Ru:Pt=90:10) to construct the RuPt (110) catalytic surface, as illustrated in Fig. S52. For semi-hydrogenation of acetylene molecule on the hydrogen-terminated $RuO_2$ (110) surface, four layers slab of tetragonal $RuO_2$ (110) surface, which was built using a $2 \times 3$ supercell, as shown in Fig. S55. Assuming that the $H_2$ molecule would dissociate at two neighbouring stable sites. Hence, two paths were considered for each surface and the reaction energies are shown in Fig. S53. For the Ru surface, these paths are: Path I: $H_2 \rightarrow H_{(HCP)} + H_{(FCC)}$, Path II: $H_2 \rightarrow H_{(HCP)} + H_{(HCP)}$ and for the RuPt surface: Path I: $H_2 \rightarrow H_{(HCP-Bridge)} + H_{(Pt)}$, Path II: $H_2 \rightarrow H_{(HCP-Bridge)} + H_{(FCC-Bridge)}$.

## Statistics and Reproducibility

The experiments were repeated minimum three times and the error bars were reported as the standard deviation of the measurements. No statistical method was used to predetermine sample size. No data were excluded from the analyses. The experiments were not randomized; The Investigators were not blinded to allocation during experiments and outcome assessment.

## Data availability

The data that support the findings of this work are available within the article, its supplementary information file (Figures S1 to S57, Tables S1 to S6), Supplementary Data 1 and 2, as well as source data file. The data are also available from the corresponding authors upon request. Source data are provided with this paper.

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

## Acknowledgements

We acknowledge the funding support of the Department of Atomic Energy project no. 12-R&D-TFR-RTI4003 and Mission Innovation India, Department of Science & Technology, Government of India (VP). We acknowledge the EM and XRD facility of TIFR, Mumbai. We also acknowledge the XPS facility of IIT, Mandi. We thank Prof. Achanta Venugopal and Mr. Gajendra Mulay (Dept. of Condensed Matter Physics & Materials Science) of TIFR, Mumbai, FDTD support. This work was supported by "Advanced Research Infrastructure for Materials and Nanotechnology in Japan (ARIM)" of the Ministry of Education, Culture, Sports, Science and Technology (MEXT), Grant Number JPMXP1223UT0082 (TT). The synchrotron radiation experiments were performed under the approval of the Japan Synchrotron Radiation Research Institute (JASRI) (Proposal No. 2023A1608). We acknowledge the financial support from Khalifa University of Science and Technology under the Research Innovation Grant (RIG)-2023-01 (NS).

## Author contributions

V.P. and T.T. proposed the research direction and designed the project. V.P. guided the project. G.S., S.M., and V.P. designed various experiments. G.S. performed the experiments (synthesis, characterizations, catalysis, mechanism, etc.). R.V. assisted G.S. in these experiments as well as in FDTD simulations. EM characterizations and X-ray studies were conducted by S.M. Data were analyzed by G.S., R.V., S.M., T.T., and V.P. Theoretical calculations were carried out by K.M.B. and N.S. G.S. and V.P. wrote the overall manuscript. Everyone commented on the manuscript.

## Competing interests

The authors declare no competing interests.
