## [Peer Review File · Nature Communications]

Pt-Doped Ru Nanoparticles Loaded on 'Black Gold' Plasmonic Nanoreactors as Air Stable Reduction CatalystsREVIEWER COMMENTS

Reviewer #1 (Remarks to the Author):

Manuscript Title: Pt-Doped Ru Nanoparticles Loaded on 'Black Gold' Plasmonic Nanoreactors as Air Stable Reduction Catalysts

Reviewer Comments:

The findings and results provided in this manuscript are novel and very interesting. The results provided in the manuscript can have very high impact in the field. In Reviewer's view, the manuscript in its current version has some minor issues. The manuscript may be considered for publication after these minor issues are addressed carefully. The issues are as follows:

1. The authors claim that the light enhancement effect reported in the manuscript is due to the plasmonic effect of Au. However, the wavelength-dependent results provided in Figure 1e raises some major questions about this claim. For example, in Figure 1e, Productivity is reported to be approximately constant in the 400 nm to 800 nm range. For Au, close to the 400 nm region, it is expected that inter-band transition dominates the optical excitation. Plasmonic excitation starts playing the role only in the later (longer) wavelength region. For example, please see the following three references for more details. So, the question here is – what dominates the enhancement effect reported in this manuscript? Inter-band excitation or plasmonic excitation or both? This issue needs to be addressed in the manuscript.

(i) Halas and coworkers, "Cu Nanoshells: Effects of Interband Transitions on the Nanoparticle Plasmon Resonance", *J. Phys. Chem. B* 2005, 109, 39, 18218–18222.

(ii) Jain and coworkers, "Plasmonic photosynthesis of C1–C3 hydrocarbons from carbon dioxide assisted by an ionic liquid", *Nature Communications*, Volume 10, Article number: 2022 (2019).

(iii) Kolwas and Derkachova, "Impact of the Interband Transitions in Gold and Silver on the Dynamics of Propagating and Localized Surface Plasmons", *Nanomaterials* 2020; 10(7): 1411.

2. The authors claim that (on page #2), "The catalytic activity showed superlinear dependence of ethene production on the light intensity with a power law exponent of 2.37 (rate $\propto I^n$), indicating a hot electron-mediated non-thermal pathway (Fig. 1c, Fig S16a)." The question that the Reviewer would like to raise here is - what is the rationale for this statement? For example, a similar trend can also be expected in purely thermal effects. For example, please see the following five references for more details. So, the authors need to provide clarity to this claim in the revised manuscript.

(i) Dubi, Y.; Un, I. W.; Sivan, Y. Thermal Effects – an Alternative Mechanism for Plasmon-Assisted Photocatalysis. *Chem. Sci.* 2020, 11 (19), 5017–5027.

(ii) Dubi, Y.; Sivan, Y. "Hot" Electrons in Metallic Nanostructures—Non-Thermal Carriers or Heating? *Light Sci Appl* 2019, 8 (1), 89.

(iii) Aizpurua, J. et al. Theory of Hot Electrons: General Discussion. *Faraday Discuss.* 2019, 214, 245–281.

(iv) Aizpurua, J. et al. New Materials for Hot Electron Generation: General Discussion. *Faraday Discuss.* 2019, 214, 365–386.

(v) Aizpurua, J. et al. Dynamics of Hot Electron Generation in Metallic Nanostructures: General Discussion. *Faraday Discuss.* 2019, 214, 123–146.

3. The authors also claim that (on page #2), "Notably, the quantum efficiency of this reaction first

increased with an increase in light intensity (up to 3 W cm⁻²) and then decreased with a further increase in light intensity (Fig. S16b). This indicated the transition from non-thermal to thermal pathways at higher intensities.” It is not clear to the Reviewer, how this change in the quantum efficiency is indication of a transition from the non-thermal pathway to the thermal pathway? The authors may need to provide references for this. Alternatively, the authors may revisit their analysis – both non-thermal and thermal effects may play in both regions and one effect may play the dominant role in comparison to the other effect in some region.

4. Does Figure 2c represent the UV-Vis extinction spectrum or absorption spectrum? Please see the reference below for the distinction. How was this spectrum measured in solid sample or particles dispersed in the dispersed liquid phase? The authors may consider including more details on this in the supporting information. Also, from this data (Figure 2c), what is the expected region of wavelengths to see the plasmonic effect? Does this Figure support any of the reaction data, such as productivity at different wavelengths?

Reference (i): U. Aslam, S. Chavez, S. Linic, *Nature Nanotech* 2017, 12, 1000.

5. The authors provided a plasmonic mechanism on page #6. But it is not mentioned whether this mechanism is expected to occur via the Photoredox mechanism or the MGR-type mechanism. This distinction is very important for the readers to visualize the mechanism. Please see the following review article for the distinction of these mechanisms. Also, what is the expected rate limiting step (RLS) for this reaction? Reactant activation (dissociation) or product desorption or surface reaction? How does the non-thermal effect enhance this RLS?

Reference (i): Ramakrishnan, S. B. et al. Photoinduced Electron and Energy Transfer Pathways and Photocatalytic Mechanisms in Hybrid Plasmonic Photocatalysis. *Advanced Optical Materials* 2021, 2101128.

6. The reaction data under dark conditions (pure thermal conditions) are provided in Figure 1d. For these pure thermal conditions, Arrhenius-type relation is expected between the rate and temperature. But the productivity/rate data at reaction temperatures of 200, 220 and 250 C show a relatively small increase. What are the possible reasons for this? Does the sintering of catalysts occur at these high temperatures?

7. Do authors see any evidence for H₂ oxidation or hydrocarbon oxidation while analyzing their product analysis? Or, on what basis these possible parallel oxidation reactions are ruled out?

8. Minor suggestion: In the catalysis and photocatalysis word “Rate” is used instead of the word “Productivity” as used by the authors in the manuscript. The authors may consider changing the word “Productivity” to in terms of “Rate of reaction” or “Product yield”.

Reviewer #2 (Remarks to the Author):

In this manuscript Sharma and coauthors reported the synthesis of PtRu alloy nanoparticles loaded on “black gold” as plasmonic nano-reactor for the semihydrogenation of acetylene in excess ethylene. In this work the authors performed abundant study including in terms of catalyst characterization, catalytic performance and mechanism study, including both experimental and computational studies. In summary they have three major findings, i.e., 1) the plasmonic hydrogenation of acetylene into ethylene with high

performance as they claimed while with which I cannot agree; 2) they found significant air flow dependent catalyst stability which I think quite interesting but with less convincing explanation in the current study; 3) significant synergistic effect between Pt and Ru without convincing explanation. In my humble opinion, these three parts are not closely related and authors might be able to report them in three independent papers. Putting them in one manuscript will make the manuscript complicated and overloaded thus losing the key point of a work. In addition, there were many critical issues for the whole work so that I cannot recommend the acceptance of this work at its current state.

The major issues are listed below for the authors' consideration.

1. For the first parts, i.e., the plasmonic reaction.

1) For acetylene semihydrogenation, an important point is that the selectivity is highly dependent on the acetylene conversion. For many or even most semihydrogenation catalysts, ethylene selectivity could be very high as long as the acetylene conversion is less than unity. However, once the conversion approaches 100%, the selectivity will decrease significantly to much negative value due to the hydrogenation of large amount of ethylene. Therefore, one of the major issues in this work is that for the plasmonic hydrogenation the acetylene conversion is rather low (often 20%), thus the high ethylene selectivity is unrepresentative thus is less meaningful. Authors may want to examine the selectivity in full conversion. In addition, for the practical application, a full conversion is necessary as it usually requires a very low concentration of acetylene (<5 ppm).

2) Another major issue is that authors claim their catalysis is highly active (the so-called first-of-its-kind). However, Table S1 suggested that quite a few catalysts are similar to or even more active than theirs. For example, their catalyst gives an ethylene productivity of 320 mmol/gcat/h with metal loading of 10 wt%. However, Pd/ZnO (entry 2) and Pd/TiO₂ (entry 5) give 131.5 and 48.2 mmol/gcat/h with only 1 wt% and 0.15 wt% metal loading, respectively. Therefore, if the activity was normalized by metals, the latter two will have much higher activity (10 and 66 times, i.e., 1215 and 3213). The same for Pd_{0.006}Cu/SiO₂ (entry 9) and GaPd/Al₂O₃ (entry 13) despite their activity at a slightly higher temperature. On the other hand, other catalysts such as entry 3, 4, 7 and 8 have similar activity. So their catalyst didn't have superior activity, not mention the high cost of the black gold support. Therefore, I think the high activity of their catalyst might be overstated.

3) The KIE data were not fully analyzed. Firstly, the KIE results of H₂-D₂ can only reflect the photo-thermal activation of H₂ rather than other reactants (last sentence on Page 6). In addition, the order of KIE can determine whether H₂ dissociation is involved into the rate-determining step. Authors may want to discuss this data more carefully.

4) Figure S28 suggested that the activation energy (E_a) is temperature dependent, so I am curious how did the authors obtain the different E_a values at certain temperatures?

5) On page 6 authors found for partially reduced sample the enhancement degree of local electronic field is lower than that on oxide samples, this is in contrast to the activity increase by adding air flow.

6) I have problem in understanding the activity became saturated with a conversion rate close to 400 mmol/g/h. What's the corresponding conversion of acetylene? Only reaching 100% one can claim the "saturated activity".

2. For the air flow enhancing catalyst stability

1) The adding of air into hydrogenation reaction is seriously dangerous. Authors should provide the concentration of various contents after adding air flow and the explosion limits of various contents.

2) The H₂-TPR results are hard to understand, why authors didn't provide the TCD signals and MS signals of H₂O? It seems the reduction occurred at room temperature as the H₂ signal increase with reduction temperature. Also the time should change to temperature in H₂-TPR figures.

3) According to Figure 1f, the true active metal for this reaction is Pt. Why did authors only analyze the change of Ru to determine the effect of air flow in catalysts stability? I think author should clarify the role of Pt and Ru in PtRu alloy for this reaction which is related to the 3rd question (see below).

3. For PtRu alloy

1) Figure 1f suggested that Pt and PtRu have almost same activity (productivity of ethylene+ ethane) but large different selectivity. Therefore, I think the deactivation without air flow and non-deactivation with the presence of air flow might be restudied based on revealing the separate role of Pt and Ru, and probably focusing on the role of Pt rather Ru.

2) It is well-accepted that Pd-based catalysts are best for semihydrogenation reaction. Why in this work authors use PtRu rather than Pd or Pd-M alloy?

Reviewer #3 (Remarks to the Author):

Sharma et al. have reported on the development of RuPtOx nanoparticles on SiO₂-supported Au nanoparticles for the chemoselective hydrogenation of acetylene. This study combines extensive catalyst characterization, in situ spectroscopy, and DFT theory to support their conclusions. Overall, the work is thorough. It appears that their claims are well supported by both experiment and theory. The authors presented a convincing argument for the cooperativity effects of RuOx and Pt sites. It was especially nice to see that this catalyst performs under the industrially relevant condition of excess ethene. The proposed mechanism derived from in situ FTIR is also logical, although in some regards some assumptions were made. This would warrant a follow-up study, in my opinion, which in an of itself could be highly impactful in the field of plasmonic photocatalysis.

At times, I found their claims were grandiose and that assertions of the efficacy of their materials were overblown or incomplete. This criticism is particularly true when comparing their work to previous reports (see Comparison of Best Reported Catalytic Systems & Table 1; Table S4). The authors cited references 29, 33-35 in their introduction, yet ignored all but Ref. 35 in their table & description in the main text. Several additional photocatalytic hydrogenation catalysts were reported in Table S4 and it was unclear why specific examples ended up in two tables reporting the same outcomes. I believe it's important for the authors to present a more leveled and unbiased comparison of the literature. This is always true, but especially so when comparing conversions and selectivity of reactions in series where reactor geometry, flow rates, reactant partial pressures, etc. are critical to the conclusions drawn. Many of the studies the authors cited use experimental conditions or different reactor designs that make a quantitative comparison between studies essentially meaningless. As reported, the text presents this work in a very positive light, which it deserves to be, but the data speaks for itself without potentially skewed colorful claims.

Overall, I commend the authors on their thorough work and recommend this for publication with minor revision (at the discretion of the editor).

We express our gratitude to the reviewers for their encouraging feedback, valuable comments and constructive suggestions, all of which have contributed to the enhancement of our manuscript. The detailed Point-by-point responses to each valuable comment and suggestion are given below.

Point-by-point Response to the Reviewer # 1

The findings and results provided in this manuscript are novel and very interesting. The results provided in the manuscript can have very high impact in the field. In Reviewer's view, the manuscript in its current version has some minor issues. The manuscript may be considered for publication after these minor issues are addressed carefully.

>> We thank the Reviewer for appreciating the novelty and impact of our findings. By addressing their constructive criticism, we were able to further strengthen the quality and impact of our study. Each point is addressed below with a description of the actions taken upon revision.

1. The authors claim that the light enhancement effect reported in the manuscript is due to the plasmonic effect of Au. However, the wavelength-dependent results provided in Figure 1e raises some major questions about this claim. For example, in Figure 1e, productivity is reported to be approximately constant in the 400 nm to 800 nm range. For Au, close to the 400 nm region, it is expected that inter-band transition dominates the optical excitation. Plasmonic excitation starts playing the role only in the later (longer) wavelength region. For example, please see the following three references for more details. So, the question here is – what dominates the enhancement effect reported in this manuscript? Inter-band excitation or plasmonic excitation or both? This issue needs to be addressed in the manuscript.

(i) Halas and co-workers, "Cu Nanoshells: Effects of Interband Transitions on the Nanoparticle Plasmon Resonance", J. Phys. Chem. B 2005, 109, 39, 18218–18222.

(ii) Jain and co-workers, "Plasmonic photosynthesis of C1–C3 hydrocarbons from carbon dioxide assisted by an ionic liquid", Nature Communications, Volume 10, Article number: 2022 (2019).

(iii) Kolwas and Derkachova, "Impact of the Interband Transitions in Gold and Silver on the Dynamics of Propagating and Localized Surface Plasmons", Nanomaterials 2020; 10(7): 1411.

>> We acknowledge the reviewer's valid concern regarding the need for clarification about the interband and plasmonic activation mechanism and are grateful for bringing this up and also kindly providing these valuable references.

We believe that interband transitions, as well as plasmonic excitations, both occur in the wavelength range ~ 400 nm for black gold, but it is the plasmonic activation that is dominant because of its higher absorption cross-section than the interband transitions. The wavelength-dependent FDTD simulations (Fig. S30) also indicate a significant electric field enhancement in that range. The plasmonic black gold employed in this work shows a broadband absorption profile in the visible range due to plasmonic coupling between different gold nanoparticles at varying inter-particle distances (Fig. S23a), which makes it different from the colloidal Au nanoparticles of certain sizes having narrow band absorption (Fig. S23b), used in Jain and co-workers, Nature Communications, Volume 10, 2022 (2019). Other mentioned references also possess a narrow band corresponding to plasmonic absorption

centred around 520 nm, with weak absorption in the 400 nm region, while black gold has strong absorption in the 400 nm region (Fig. S23a). Hence, plasmonic excitation dominates interband excitation in black gold, but with contribution from interband excitation also. We have now clarified this crucial point in the revised manuscript.

Figure S23. (a) Broadband UV-Vis extinction spectra of DPC and DPC/RuPt as a dispersion form in ethanol showing broadband absorption in the visible range; (b) Narrow UV-Vis extinction spectrum of a colloid of the Au NPs used for the preparation of the photocatalyst film. The spectrum exhibits a localized surface plasmon resonance (LSPR) band centered around 520 nm, as indicated by the dotted line (Adapted with permission from Jain and co-workers, *Nat. Commun.*, **10**, 2022 (2019), copyright 2019 Springer Nature)

2. The authors claim that (on page #2), “The catalytic activity showed superlinear dependence of ethene production on the light intensity with a power law exponent of 2.37 ($\text{rate} \propto I^n$), indicating a hot electron-mediated non-thermal pathway (Fig. 1c, Fig S16a).” The question that the Reviewer would like to raise here is - what is the rationale for this statement? For example, a similar trend can also be expected in purely thermal effects. For example, please see the following five references for more details. So, the authors need to provide clarity to this claim in the revised manuscript.

(i) Dubi, Y.; Un, I. W.; Sivan, Y. Thermal Effects – an Alternative Mechanism for Plasmon-Assisted Photocatalysis. *Chem. Sci.* 2020, 11 (19), 5017–5027.

(ii) Dubi, Y.; Sivan, Y. “Hot” Electrons in Metallic Nanostructures—Non-Thermal Carriers or Heating? *Light Sci Appl* 2019, 8 (1), 89.

(iii) Aizpurua, J. et al. Theory of Hot Electrons: General Discussion. *Faraday Discuss.* 2019, 214, 245–281.

(iv) Aizpurua, J. et al. New Materials for Hot Electron Generation: General Discussion. *Faraday Discuss.* 2019, 214, 365–386.

(v) Aizpurua, J. et al. Dynamics of Hot Electron Generation in Metallic Nanostructures: General Discussion. *Faraday Discuss.* 2019, 214, 123–146.

>> We extend our appreciation to the reviewer for their insightful comment. The superlinear dependence on light intensity, signifying a non-thermal pathway mediated by hot electrons, has been observed by numerous researchers and is a widely accepted phenomenon for comprehending these plasmonic effects. Nevertheless, as the referee correctly pointed out, there is still an ongoing debate among some authors regarding this matter, where a superlinear dependence trend was claimed to be expected in purely thermal effects. As suggested by the reviewer, we have now tried to clarify these critical aspects in the revised MS.

Sivan et al., in their report, *Thermal Effects – an Alternative Mechanism for Plasmon-Assisted Photocatalysis. Chem. Sci.* 2020, 11 (19), 5017–5027, explain the superlinear dependence of reaction rate on intensity observed by Linic and co-workers (*Christopher et al., Nat. Mater., 2012, 11, 1044–1050*) by assuming a purely thermal behaviour and the proposed temperature-shifted Arrhenius Law. They discussed heat distribution under light illumination in these systems and introduced the shifted Arrhenius equation as follows:

$$R \sim \exp\left(-\frac{\varepsilon_a}{k_B T(r) + a I_{inc}}\right) \quad (1)$$

This equation has been adjusted to account for heating induced by illumination.

They propose a linear temperature increase model-

$$T(I_{inc}) = T_M + b(I_{inc}) \quad (2)$$

where $T(I_{inc})$ is the actual temperature of the reactor in light and T_M , is the experimentally measured temperature.

To check the applicability of Sivan's model in the current study, we re-evaluated our intensity-dependent and temperature study conducted in the absence of light (Fig. S17a). If we assumed a purely thermal behaviour and correlate the measured temperature under light conditions (T_M) and the temperature at which similar activity was observed in the absence of light ($T(I_{inc})$), we observed that the difference between these two temperatures did not conform to the anticipated linear trend at higher intensities (Fig. S17b), as suggested by the Sivan's model presented in eq. (2). This observation clearly contradicts the assumption of a linear temperature increase model proposed by Dubi and Sivan et al.

Figure S17. (a) Acetylene conversion in light and dark versus the catalyst bed temperature, (b) Temperature difference between $T(I_{\text{inc}})$ and T_M versus light intensity.

Similar observations were also made by Halas et al. (*Response to Comment on “Quantifying hot carrier and thermal contributions in plasmonic photocatalysis”, Science 2019 9545*) that a linear model can only be applied for small temperature increases (<100 K). In the low-intensity regime, where the trend is indeed linear, the role of non-thermal effects can still not be neglected if the light dependence of activation energy is considered, as discussed in a comment on their study by Jain et al. (*Comment on “Thermal Effects – An Alternative Mechanism for Plasmon-Assisted Photocatalysis” by Y. Dubi, I. W. Un and Y. Sivan, Chem. Sci., 2020, 11, 5017. Chem. Sci. 11, 9022 (2020)*).

In their comprehensive analysis, Jain et al. delved into this matter in great detail, starting with the premise that a decrease in activation energy correlates linearly with light intensity in the field of photocatalysis. This assumption formed the foundation for a detailed examination of the complex relationship between light-induced effects and catalytic performance.

$$E_a = E_a^{\text{dark}} - BI \quad (3)$$

Here, B is a proportionality constant with units of eV cm² W⁻¹ if E_a is expressed in units of eV and I in units of W cm⁻². B is a wavelength-dependent quantity. On further solving the above and putting this in Arrhenius equation, we get:

$$E_a = E_a^{\text{dark}}(1 - bI) \quad (4)$$

b is $\frac{B}{E_a^{\text{dark}}}$ and has units of cm² W⁻¹.

$$R = A \exp\left[-\frac{E_a^{dark}}{k_B T_s}(1 - bI)\right] \quad (5)$$

After using a Taylor expansion around $I = 0$, which is a dark condition, we get,

$$\frac{1}{1-bI} = 1 + bI + (bI)^2 + (bI)^3 + \dots \quad (6)$$

Higher-order terms can be ignored in the light-intensity regime ($I < 1/b$). After putting this in equation (5), we get

$$R = R_0 \exp\left[-\frac{E_a^{dark}}{k_B T_s(1+bI)}\right] \quad (7)$$

When this equation is compared to the general Arrhenius equation, the reaction appears to take place at a theoretical temperature that is proportional to the light intensity (I) higher than the actual temperature T_s .

$$T_{dummy} = T_s(1 + bI) \quad (8)$$

This equation, which happens to be the same one used by Sivan and their colleagues in their research, reinforces the case for the thermal pathway over the non-thermal pathway. Within the mathematical context of this equation, the conclusion is that light excitation exclusively leads to an increase in temperature without any influence on activation energy. This mathematical treatment of the Arrhenius equation effectively curtains the potential non-thermal effects of light excitation, masking them under the guise of temperature increase, i.e. the thermal pathway.

Moreover, we also studied the non-thermal activation mechanism by the KIE measurement (Fig. 4d) as well as the photocatalytic studies with external heating (Fig. 4f). Therefore, we believe that the superlinear dependence of ethene production on the light intensity, characterized by a power law exponent of 2.37 ($\text{rate} \propto I^n$), suggests a hot electron-mediated non-thermal pathway, alluding to the possibility of multi-photon absorption as observed in Linic et al., *Singular characteristics and unique chemical bond activation mechanisms of photocatalytic reactions on plasmonic nanostructures*, *Nat. Mater.*, 2012, 11, 1044–1050.

A brief of the above discussion on the explanation of the superlinear dependence of reaction rates on light intensity with both points of view has been included in the revised manuscript.

3. The authors also claim that (on page #2), “Notably, the quantum efficiency of this reaction first increased with an increase in light intensity (up to 3 W cm⁻²) and then decreased with a further increase in light intensity (Fig. S16b). This indicated the transition from non-thermal to thermal pathways at higher intensities.” It is not clear to the Reviewer, how this change in the quantum efficiency is indication of a transition from the non-thermal pathway to the thermal pathway? The authors may need to provide references for this. Alternatively, the authors may revisit their analysis – both non-thermal and thermal effects may play in both regions and one effect may play the dominant role in comparison to the other effect in some region.

>> We fully agree with the reviewer that both non-thermal and thermal effects play key roles during plasmonic catalysis, with the non-thermal effect dominated at lower light intensity while the thermal effect dominated at higher light intensity.

Our results primarily indicate that there is a specific light intensity range in which the majority of incident photons actively engage in catalysing the reaction (Fig. S18). The increase in the quantum efficiency up to 3 W cm^{-2} (Fig. S18b), can be explained by the electron-assisted reaction over the plasmonic DPC/RuPt, as also explained in one of the reports by Linic et al, *Singular characteristics and unique chemical bond activation mechanisms of photocatalytic reactions on plasmonic nanostructures Nat. Mater., 2012, 11, 1044*, where the transition to the superlinear regime was found to be accompanied by an increase in quantum efficiency (Fig. S18a). Specifically, we observed that the disparity between the ethene production rate in light and dark diminishes at higher light intensities where the catalyst bed temperature is $\sim 200 \text{ }^\circ\text{C}$. At higher intensity, an increase in photon flux could lead to more photo-induced heating of the catalyst, increasing the local temperature. In this high-temperature regime, the reaction might be constrained due to the limited adsorption of acetylene on the catalyst surface (more desorption due to high local temperature), thus resulting in reduced values of quantum efficiency.

Figure S18. (a) Quantum efficiency (%) as a function of light intensity for various temperatures. The dotted lines show the quantum efficiency observed at source intensities $< \sim 300 \text{ mW cm}^{-2}$ for ethene epoxidation over Ag nanocubes (Adapted with permission from Linic et al. Singular characteristics and unique chemical bond activation

mechanisms of photocatalytic reactions on plasmonic nanostructures. *Nat. Mater.* **11**, 1044–1050 (2012), Copyright 2012 Springer Nature); (b) Quantum efficiency at different light intensities and (c) moles of acetylene converted (log scale) v/s light intensity (log scale) used at different reactor temperatures (T_R), for acetylene semi hydrogenation in excess ethene over DPC/RuPt-10-Calc

Thus, we agree that both non-thermal and thermal pathways contribute to catalysis. The predominance of the thermal pathway in the higher intensity regime was based on the difference in the catalytic activity in light and dark at similar catalyst bed temperatures, which is also highlighted in Fig. S17a and Fig. S18c.

A brief of the above discussion on the explanation of Quantum efficiency dependence and related references have been included in the revised manuscript.

4. Does Figure 2c represent the UV-Vis extinction spectrum or absorption spectrum? Please see the reference below for the distinction. How was this spectrum measured in solid sample or particles dispersed in the dispersed liquid phase? The authors may consider including more details on this in the supporting information. Also, from this data (Figure 2c), what is the expected region of wavelengths to see the plasmonic effect? Does this Figure support any of the reaction data, such as productivity at different wavelengths? Reference (i): U. Aslam, S. Chavez, S. Linic, *Nature Nanotech* 2017, 12, 1000.

>> Figure 2c is the UV-Vis extinction spectrum. The baseline subtraction process was carried out (to reduce potential scattering effects) by first recording the UV-Vis extinction spectrum of DFNS/APTES dispersion in ethanol (which is the solid support to prepare black gold) and then subtracting it from the UV-Vis extinction spectrum of DPC (black gold) and DPC/RuPt-10-Calc dispersion in ethanol. As per the reviewer's suggestion, these details have been included in the revised supporting information as well as in the revised Fig. 2.

Figure 2c. UV-Vis extinction spectra of DPC and DPC/RuPt as a dispersion form in ethanol showing broadband absorption in the visible range.

Figure 1e. Effect of the wavelength of light on acetylene semi-hydrogenation and corresponding catalyst bed temperatures (using optimized flow conditions at 2.7 W cm^{-2})

The UV-Vis extinction spectrum shows broadband plasmonic behaviour of the black gold catalyst in the entire visible region (400-800nm) (Fig. 2c). The wavelength-dependent ethene production data matches well with this extinction spectrum, which shows similar activity at different wavelengths spanning the visible region (Fig. 1e). This aspect is now discussed in the revised manuscript.

5. The authors provided a plasmonic mechanism on page #6. But it is not mentioned whether this mechanism is expected to occur via the Photoredox mechanism or the MGR-type mechanism. This distinction is very important for the readers to visualize the mechanism. Please see the following review article for the distinction of these mechanisms. Also, what is the expected rate limiting step (RLS) for this reaction? Reactant activation (dissociation) or product desorption or surface reaction? How does the non-thermal effect enhance this RLS?

Reference (i): Ramakrishnan, S. B. et al. Photo-induced Electron and Energy Transfer Pathways and Photocatalytic Mechanisms in Hybrid Plasmonic Photocatalysis. *Advanced Optical Materials* 2021, 2101128.

>> We appreciate the reviewer's insightful question about Photoredox Vs. MGR-type mechanism.

In the dark case, larger KIE was observed at lower temperatures, suggesting the H₂ dissociation or acetylene hydrogenation a rate-limiting step (RLS) at lower temperatures. Whereas, the desorption of ethene seems RLS at higher temperature since almost no KIE was observed at 200 °C. On the other hand, in the presence of light, KIE was seen at any light intensity (or temperature range), indicating that the desorption of ethene seems not the RLS under light irradiation. Based on these results, light irradiation promotes the desorption of ethene, thus, the H₂ dissociation or acetylene hydrogenation step is the RLS under light irradiation.

We believe that the ethene desorption step is accelerated by a polarizing electric field on the catalyst surface, indicating the role of non-thermal effects in enhancing rate limiting step (RLS). A similar observation was made by Hu, C. et al. *Near-infrared-featured broadband CO₂ reduction with water to hydrocarbons by surface plasmon. Nature Communications* 14, 221 (2023). They observed that the electric field pointing toward the plasmonic metal surface redistributed the electrons of CO₂ from the O lone-pair orbital to its antibonding π^* orbital, weakening the C–O bond and increasing the bond length. When the electric field pointed away from the metal surface, the lone-pair electrons transferred to the σ orbital between O and the metal atom making the desorption of the product, CO facile. Such polarising effect of the enhanced electric field around the active site might be happening in our catalytic system, facilitating the C-O and O-H bond breaking to form TS(II), consequently leading to the desorption of hydrogenated product ethylene.

As per DFT calculations (Figure 7b), ethene desorption seems to be the rate limiting step (RLS) in dark. However, ethene desorption is dependent on the availability of dissociated hydrogen, which is enhanced by transient charge transfer to H₂ via Pt. It is evident from the DFT calculations that charge gets accumulated on Pt in dark (Fig. S54),

which will be even higher in light. The higher kinetic isotope effect (KIE) observed for H₂/D₂ in light (as illustrated in Fig. 4d) also indicated enhancement in H₂ dissociation assisted by high-energy electrons. These observations indicated that the reaction follows the MGR-type mechanism as nicely discussed in the review by Ramakrishnan, S. B. et al. *Adv. Optical Mater.* 2021, 2101128.

These discussions and related data are now included in the revised manuscript.

Figure S54. Calculated charge density difference between Ru and RuPt surface showing larger charge accumulation on Pt. Pink and Green colors indicates charge accumulation and depletion, respectively. Blue-Ru and Orange-Pt.

Figure 7. Proposed reaction pathway for acetylene semi-hydrogenation on DPC/RuPt-10-Calc. a) Formation of various reaction intermediates showing the role of co-existing oxide and reduced species based on in-situ FTIR studies; b) energies of initial (IS), transition (TS), intermediate (IM), and final (FS) states formed in the acetylene semi-hydrogenation by DFT calculation using nudged elastic band method.

6. The reaction data under dark conditions (pure thermal conditions) are provided in Figure 1d. For these pure thermal conditions, Arrhenius-type relation is expected between the rate and temperature. But the productivity/rate data at reaction temperatures of 200, 220 and 250 C show a relatively small increase. What are the possible reasons for this? Does the sintering of catalysts occur at these high temperatures?

>> Inspired by the reviewer's thoughtful comment, we conducted the TEM imaging and EDS mapping of the catalyst after the reaction at a higher temperature (250 °C for 3 h) and we did not observe sintering of the active sites as shown in Fig. S19.

At higher temperatures above 200 °C, acetylene desorption is dominated over adsorption, which limits the overall rate of the reaction. We have now clarified this point in the revised manuscript and included Fig. S19 in the revised Supplementary Information.

Figure S19. HAADF STEM image and EDS elemental maps of DPC/RuPt-10-spent-250C-3h

7. Do authors see any evidence for H₂ oxidation or hydrocarbon oxidation while analyzing their product analysis? Or, on what basis these possible parallel oxidation reactions are ruled out?

>> We analysed the products via gas chromatography with TCD, FID, as well as mass detectors. No CO₂ was observed (other than 400 ppm which is present in air feed), indicating no hydrocarbon oxidation.

We believe that H₂ oxidation is taking place, yielding water due to continuous regeneration (oxidation and reduction) of active sites (Ru to RuO₂) during the catalysis (Fig. 7a). We carried out the light-induced reduction

of DPC/RuPt-Calc-10 (Figure 5e), and observed the water formation. We have highlighted the discussion about the by-products in the revised manuscript and revised Fig 5e.

Figure 5e. Light-induced reduction of DPC/RuPt-10-Calc in H₂ at light illumination $\sim 2.7 \text{ W cm}^{-2}$ (400-1100nm)

8. Minor suggestion: In the catalysis and photocatalysis word “Rate” is used instead of the word “Productivity” as used by the authors in the manuscript. The authors may consider changing the word “Productivity” to in terms of “Rate of reaction” or “Product yield

>> As per the reviewer’s kind suggestion, we have changed the word “productivity” to “production rate” in the revised manuscript.

Point-by-point Response to the Reviewer # 2

In this manuscript Sharma and coauthors reported the synthesis of PtRu alloy nanoparticles loaded on “black gold” as plasmonic nano-reactor for the semihydrogenation of acetylene in excess ethylene. In this work the authors performed abundant study including in terms of catalyst characterization, catalytic performance and mechanism study, including both experimental and computational studies. In summary they have three major findings, i.e., 1) the plasmonic hydrogenation of acetylene into ethylene with high performance as they claimed while with which I cannot agree; 2) they found significant air flow dependent catalyst stability which I think quite interesting but with less convincing explanation in the current study; 3) significant synergistic effect between Pt and Ru without convincing explanation. In my humble opinion, these three parts are not closely related and authors might be able to report them in three independent papers. Put them in one manuscript will make the manuscript complicated and overloaded thus losing the key point of a work. In addition, there were many critical issues for the whole work so that I cannot recommend the acceptance of this work at its current state.

>> We are grateful for the reviewer’s comment that he finds the work extensive and suitable for three independent publications. Our aim was to perform an in-depth study of this unique plasmonic catalytic system, which would be suitable for a prestigious journal like Nature Communications.

The major issues are listed below for the authors’ consideration.

1. For the first parts, i.e., the plasmonic reaction.

1) For acetylene semihydrogenation, an important point is that the selectivity is highly dependent on the acetylene conversion. For many or even most semihydrogenation catalysts, ethylene selectivity could be very high as long as the acetylene conversion is less than unity. However, once the conversion approaches to 100%, the selectivity will decrease significantly to a much negative value due to the hydrogenation of a large amount of ethylene. Therefore, one of the major issues in this work is that for the plasmonic hydrogenation the acetylene conversion is rather low (often 20%), thus the high ethylene selectivity is unrepresentative and thus less meaningful. Authors may want to examine the selectivity in full conversion. In addition, for the practical application, a full conversion is necessary as it usually requires a very low concentration of acetylene (<5 ppm).

>> We agree with the reviewer that acetylene conversion is an important criterion for determining the applicability of the catalyst in practical situations. However, our main focus was on understanding the role of plasmonic excitation for semihydrogenation reactions using solar energy at ambient conditions. Hence, we did not attempt to increase the conversion to 100%.

Inspired by the reviewer’s valid criticism, we now tried to increase the conversion. We realized that in our fixed bed flow chamber reactor, we could only achieve ~80% conversion with ~90% selectivity. This was due to the limitation of the reactor set-up used in terms of light-exposed catalyst area. We have now used a different reactor set-up, a thin flat cell made up of a quartz tube (Fig. 27a), in which we could achieve ~97% conversion, maintaining high selectivity (Fig. 27b).

We now hope that the reviewer will appreciate nearly a full conversion achieved and this is now added in the revised MS and SI.

Figure S27. (a) Quartz flat cell flow reactor set-up employed to increase the illuminated catalyst area (b) Acetylene conversion and ethene selectivity trend showing high selectivity being maintained at high conversion. The highest conversion achieved with total flow= 9 mL min⁻¹, C₂H₂/C₂H₄/H₂/Ar/Air=0.2/4/2/1.8/1 and AM1.5 light illumination at 1 bar pressure.

2) Another major issue is that authors claim their catalysis is highly active (the so-called first-of-its-kind). However, Table S1 suggested that quite a few catalysts are similar to or even more active than theirs. For example, their catalyst gives an ethylene productivity of 320 mmol/gcat/h with metal loading of 10 wt%. However, Pd/ZnO (entry 2) and Pd/TiO₂ (entry 5) give 131.5 and 48.2 mmol/gcat/h with only 1 wt% and 0.15 wt% metal loading, respectively. Therefore, if the activity was normalized by metals, the latter two will have much higher activity (10 and 66 times, i.e., 1215 and 3213). The same for Pd_{0.006}Cu/SiO₂ (entry 9) and GaPd/Al₂O₃ (entry 13) despite their activity at a slightly higher temperature. On the other hand, other catalysts such as entry 3, 4, 7 and 8 have similar activity. So their catalyst didn't have superior activity, not mention the high cost of the black gold support. Therefore, I think the high activity of their catalyst might be overstated.

>> The main focus of our work was to develop plasmonic acetylene semi-hydrogenation using solar light (without any external thermal energy). When we compared our catalysts with all reported plasmonic catalysts, Table 1 in the MS, it clearly indicated the significantly high activity as compared to the best-known plasmonic catalysts. Hence, we claimed the best catalysts reported.

The term “*first-of-its-kind*” was used not in terms of productivity but due to its stable behaviour in the air (even during hydrogenation reaction), since this was not observed before, indicating a unique possibility during plasmonic catalysis over conventional catalysis.

We apologise for the overstatements, which were unintentional. In the revised manuscript, we have removed all such text which sounded overstatements and toned down our claims and conclusions.

Regarding the calculations of productivity per gram of catalyst Vs. per gram of active sites, we fully understand the reviewer's viewpoints, which was actually our big dilemma while analysing our results. We contemplated extensively to decide how to compare our catalysts with reported ones and then decided to use productivity per gram of catalyst and not per gram of active sites. The reason is that in our black gold/RuPt catalysts, every single component (Gold, Ru, and Pt) plays a critical role and they can not work independently and catalyst is only active when all three components are together. Hence, we decided not to refer only to Ru or Pt as active sites. Therefore, we opted for the productivity comparison by taking the entire catalyst weight rather than one metal component.

Similar observations were also made by references cited by the reviewer. For example, Pd/ZnO catalyst was characterized as an intermetallic rather than supported metal catalyst, which implies that both Pd and ZnO play active roles. Therefore, focusing solely on Pd as the active site may not be accurate, a point similarly applicable to Pd_{0.006}Cu/SiO₂ and GaPd/Al₂O₃ catalysts. In our catalytic system, there are three metals: Au, Ru and Pt and only considering Pt as the active site would be inaccurate as without the other components, black gold and Ru, the desired acetylene conversion and selectivity cannot be achieved.

Also, several other factors need to be considered in the assessment of catalyst performance with reported catalysts, like the mode of activation and the energy input required. We were able to carry out the reaction purely using solar light (without any external heating) and it surpasses the other catalysts in terms of stability and selectivity, up to 100 hours, in contrast to the 20-hour testing duration for Pd/ZnO.

Regarding the cost of the black gold support, it is important to consider it as a one-time investment (like any other noble metal catalyst), particularly given the long-term stability of our catalyst.

Additionally, the photocatalytic nature of our catalyst, using solar light as an energy source (without any external thermal energy), offers the advantage of eliminating the need for additional expenses associated with the fuel required for conventional thermal reactions commonly used in industry.

3) The KIE data were not fully analyzed. Firstly, the KIE results of H₂-D₂ can only reflect the photo-thermal activation of H₂ rather than other reactants (last sentence on Page 6). In addition, the order of KIE can determine whether H₂ dissociation is involved into the rate-determining step. Authors may want to discuss this data more carefully.

>> We are grateful to the reviewer for bringing up this important point, as also pointed out in Comment-5 of Reviewer #1. The KIE results of H₂-D₂ indicate that there is a significant difference in the thermal and non-thermal mechanism of plasmonic activation for H₂ dissociation. Due to the unavailability of labelled acetylene, similar measurements could not be performed for acetylene activation.

In the dark case, larger KIE was observed at lower temperatures, suggesting the H₂ dissociation or acetylene hydrogenation a rate-limiting step (RLS) at lower temperatures. Whereas, the desorption of ethene seems RLS at higher temperature since almost no KIE was observed at 200 °C. On the other hand, in the presence of light, KIE was seen at any light intensity (or temperature range), indicating that the desorption of ethene seems not the RLS under light irradiation. Based on these results, light irradiation promotes the desorption of ethene, thus, the H₂ dissociation or acetylene hydrogenation step is the RLS under light irradiation.

We believe that the ethene desorption step is accelerated by a polarizing electric field on the catalyst surface, indicating the role of non-thermal effects in enhancing rate limiting step (RLS). A similar observation was made by Hu, C. et al. *Near-infrared-featured broadband CO₂ reduction with water to hydrocarbons by surface plasmon. Nature Communications 14, 221 (2023)*. They observed that the electric field pointing toward the plasmonic metal surface redistributed the electrons of CO₂ from the O lone-pair orbital to its antibonding π^* orbital, weakening the C–O bond and increasing the bond length. When the electric field pointed away from the metal surface, the lone-pair electrons transferred to the σ orbital between O and the metal atom making the desorption of the product, CO facile. Such polarising effect of the enhanced electric field around the active site might be happening in our catalytic system, facilitating the C-O and O-H bond breaking to form TS(II), consequently leading to the desorption of hydrogenated product ethylene.

As per DFT calculations (Figure 7b), ethene desorption seems to be the rate limiting step (RLS) in dark. However, ethene desorption is dependent on the availability of dissociated hydrogen, which is enhanced by transient charge transfer to H₂ via Pt. It is evident from the DFT calculations that charge gets accumulated on Pt in dark (Fig. S54), which will be even higher in light. The higher kinetic isotope effect (KIE) observed for H₂/D₂ in light (as illustrated in Fig. 4d) also indicated enhancement in H₂ dissociation assisted by high-energy electrons. These observations indicated that the reaction follows the MGR-type mechanism as nicely discussed in the review by Ramakrishnan, S. B. et al. *Adv. Optical Mater. 2021, 2101128*.

Figure S54. Calculated charge density difference between Ru and RuPt surface showing larger charge accumulation on Pt. Pink and Green colors indicates charge accumulation and depletion, respectively. Blue-Ru and Orange-Pt.

These discussions and related data are now included in the revised manuscript.

Figure 7. Proposed reaction pathway for acetylene semi-hydrogenation on DPC/RuPt-10-Calc. a) Formation of various reaction intermediates showing the role of co-existing oxide and reduced species based on in-situ FTIR studies; b) energies of initial (IS), transition (TS), intermediate (IM), and final (FS) states formed in the acetylene semi-hydrogenation by DFT calculation using nudged elastic band method.

These discussions have been included in the revised manuscript.

4) Figure S28 suggested that the activation energy (E_a) is temperature dependent, so I am curious how did the authors obtain the different E_a values at certain temperatures?

>> We are grateful for the reviewer's question. The Arrhenius plot in dark and light (Fig. 4e) indicates a non-linear behaviour in the given range of temperatures (*Ref. Truhlar, D. G., Kohen, A. Convex Arrhenius plots and their interpretation. Proc. Natl. Acad. Sci. U. S. A. 98, 848–851 (2001)*).

We plotted $\ln(k)$ versus $1000/T_s$ and apparent activation energy (E_a) was calculated by using the Arrhenius equation,

$$E_a = -\frac{R \, d\ln(k_{C_2H_4})}{d(1/T_s)} \quad (\text{J mol}^{-1}) \quad (9)$$

where $k_{C_2H_4}$ is ethene production rate.

The rate was first obtained as a function of temperature by fitting the data to a polynomial function (Fit shows $\ln K$ is a quadratic polynomial in terms of $1/T_s$).

$$\text{In Light: } -24.78 + 28.04 (1000/T_s) - 6.41(1000/T_s)^2 \quad (10)$$

$$\text{In Dark: } -17.51 + 24 (1000/T_s) - 6.15 (1000/T_s)^2 \quad (11)$$

Differentiating the obtained expression as per the equation given in (9), E_a can be obtained to have a linear dependence on $1/T_s$ and the exact expressions are given below and plotted in Fig. S32.

$$\text{In Light: } -233.2 + 106.6 (1000/T_s) \quad (12)$$

$$\text{In Dark: } -199.5 + 102.3 (1000/T_s) \quad (13)$$

The plot of Fig. S32 is a differential plot of Fig. 4e.

These points are now clarified in the revised manuscript and supporting information.

5) On page 6 authors found for partially reduced sample the enhancement degree of local electronic field is lower than that on oxide samples, this is in contrast to the activity increase by adding air flow.

>> We think that there is some confusion here in the nomenclature of samples. We compared fully reduced and partially reduced samples (and not oxide sample), as both metal and oxide phases play key role in catalysis.

The FDTD simulation data shows the electric field enhancement in the fully reduced catalyst (DPC/RuPt-25.6) is more than that in the partially reduced catalyst (DPC/RuPtOx-19.2) because of the greater extent of near field coupling with higher reduced metal content. It should be noted that the actual catalyst is a combination of the reduced and the oxidised form and both have their role to play. When only the reduced form is present, acetylene adsorption is limited since it is the oxidised form, which is crucial for the formation of the di- σ -bonded acetylene (Fig. 6, 7a), while the reduced form facilitates plasmon-assisted H_2 dissociation. Therefore, a partially reduced

catalyst performs better than a fully reduced one (the catalyst is deactivated without air due to complete reduction) without compromising significantly on the electric field enhancement (Fig. 4).

Figure 4. Electric field enhancement in a) DPC, b) DPC/RuPt (fully reduced), and c) DPC/RuPtOx (partially reduced) (Source $E_0 = 4509 \text{ V m}^{-1}$ at 2.7 W cm^{-2}) using FDTD simulations

The caption of Fig. 4 has been modified in the MS to avoid this confusion.

6) I have problem in understanding the activity became saturated with a conversion rate close to 400 mmol/g/h. What's the corresponding conversion of acetylene? Only reaching 100% one can claim the "saturated activity".

>> We evaluated the catalyst's activity based on i) the maximum rate of product formation it can achieve and ii) the highest conversion achievable. While they both may be interlinked, they are significantly different. The optimised gas hourly space velocity (GHSV) is different for full conversion and for the highest ethene production rate. At higher flow rates (higher GHSV), the interaction time is less, but the amount of the reactant molecules are in large quantities, which can react with a large number of active sites, leading to maximum productivity (but lower conversion). Whereas at low flow rates, the interaction time is greater, and a lesser number of reactant molecules are available to interact with the active sites; hence, the conversion is high (accompanied by low productivity).

The reason for choosing the high productivity regime for the study of thermal and non-thermal pathways of plasmonic activation was to understand the changes in the active site behaviour better when most of them are being utilised.

In our study, the maximum ethene production rate was achieved when acetylene conversion rate $\sim 400 \text{ mmol g}^{-1} \text{ h}^{-1}$ at $\sim 25\%$ conversion (Total flow rate = 110 mL/min). This lower value of conversion was due to the high GHSV used as well as the limitation of the reactor used in terms of light-exposed catalyst area. When we used different reactor set-up, a thin flat cell made up of quartz tube and low flow (Total flow rate = 9 mL/min) (Fig. S27b), we could achieve $\sim 97\%$ conversion maintaining high selectivity.

The maximum conversion data is now included in the revised MS and the statement: the activity became saturated with a conversion rate close to 400 mmol/g/h has been modified to avoid confusion.

2. For the air flow enhancing catalyst stability

1) The adding of air into hydrogenation reaction is seriously dangerous. Authors should provide the concentration of various contents after adding air flow and the explosion limits of various contents.

>> We thank the reviewer for raising this very serious concern. This was also our concern while doing the experiments with already highly flammable acetylene. But the concept of plasmonic catalysts seems to be helping here.

As the reviewer correctly pointed out, acetylene, ethene and hydrogen are highly flammable gases with lower flammable limit (LFL) and upper flammable limit (UFL) values of 2.5/100, 2.7/36 and 4/18.3 (vol% in the air) and should explode in the presence of sufficient air. However, their ignition temperatures are 305, 490, and 500 °C respectively. (Rowley, Jeffrey R., "Flammability Limits, Flash Points, and Their Consanguinity: Critical

In our plasmonic catalysis, when operating under optimized conditions, we only reach a temperature of 132°C, with only 1% O₂ present in the feed. As a result, there is no risk of an explosion. This highlights another advantage of using plasmonic catalysis, as it allows for the safe use of air even in the presence of highly flammable gases, which is not feasible with conventional thermal catalysis.

Figure S47. Dependence of ignition temperature of acetylene on its concentration in atmosphere (reproduced from Jones G. W. ; Miller, W. E. Ignition temperatures of acetylene-air and acetylene-oxygen mixtures. Report of Investigations, Bureau of Mines, May 1941)

This crucial point has now been addressed in the discussion for air stability in the revised MS.

2) The H₂-TPR results are hard to understand, why authors didn't provide the TCD signals and MS signals of H₂O? It seems the reduction occurred at room temperature as the H₂ signal increase with reduction temperature. Also the time should change to temperature in H₂-TPR figures.

>> Thanks for raising this important concern and sorry for not presenting this data accurately. Kindly note that this is not conventional temperature programmed reduction (TPR) using external thermal energy, but it is reduction using solar energy (without any external heating), so it can be named as a light programmed reduction (LPR).

When the light was off, the reactor was at room temperature (30 °C) and no reduction took place. As soon as light (400-1100 nm, 2.7 W cm⁻²) was switched on, the temperature increased to 132 °C (within 20 seconds) and the reduction took place immediately. Hence, the X-axis was time. We have now added a secondary X-axis for temperature, showing temperature corresponding to light on and off conditions.

On the reviewer's suggestion, we also added TCD and mass signals of H₂O in the plot, illustrating the reduction of the catalyst through a decrease in the H₂ mass signal and a simultaneous increase in the H₂O mass signal.

Figure 5e. Light induced reduction of DPC/RuPt-10-Calc in H₂ at light illumination $\sim 2.7 \text{ W cm}^{-2}$ (400-1100nm)

The figure has been modified accordingly in the revised MS.

3) According to Figure 1f, the true active metal for this reaction is Pt. Why did authors only analyze the change of Ru to determine the effect of air flow in catalysts stability? I think author should clarify the role of Pt and Ru in PtRu alloy for this reaction which is related to the 3rd question (see below).

For PtRu alloy, 1) Figure 1f suggested that Pt and PtRu have almost same activity (productivity of ethylene+ethane) but large different selectivity. Therefore, I think the deactivation without air flow and non-deactivation with the presence of air flow might be restudied based on revealing the separate role of Pt and Ru, and probably focusing on the role of Pt rather Ru.

>> Thank you for these important comments. Please note that we studied both Ru and Pt sites using XRD, XPS and XANES as well as in-situ FTIR, DRIFTS and TEM. However, we only observed changes in the oxidation state of Ru and not Pt in the presence or absence of airflow.

We agree with the reviewer that Pt alone can hydrogenate acetylene to ethylene, but it also hydrogenates ethylene to ethane, while our main goal was to achieve acetylene semi-hydrogenation and stop at the ethene stage, which was not possible by Pt sites. When we prepared Pt-doped Ru nanoparticles containing RuO₂ phases, acetylene was found to chemisorb on RuO₂ (and not on Pt) and only H₂ was dissociated at Pt (of RuPt). Hence, in our

catalytic system, RuO₂ and RuPt (activated by black gold) are active sites for acetylene activation and H₂ dissociation, respectively. This was also confirmed by DRIFTS, transmission FTIR, and DFT studies.

Based on these observations, we propose that during acetylene semi-hydrogenation (Figure 7), a partially oxidized RuPt alloy surface with a terminal hydroxyl group interacted with acetylene and generated di- σ -bonded acetylene via the loss of 2H from the hydroxyl groups and the breaking of the triple bond to form two C-O bonds (step i). The hydrogen lost from hydroxyls was then utilized to form a $=C-H$ bond by breaking a C-O bond (step ii). After both the C-O bonds were broken, the ethene formed interacted with the O in a π -bonded fashion (step iii). After supplying external H₂, which was dissociated by the Pt sites of the RuPt catalyst, the hydroxyls were regenerated, and the ethene molecules were released, and the catalyst became available for the next cycle (step iv). RuPt NPs could thus facilitate the dissociation and migration of H radicals across the surface in a controlled way (step v), which could then add to the activated di- σ -bonded acetylene at the RuO₂ phase, lowering the overall activation energy barrier, also confirmed by the DFT calculations. Without the RuO₂ phase, there would not be such an activation of acetylene. Moreover, there is an expected decrease in the extent of acetylene adsorption on the reduced DPC/RuPt due to the negative charge accumulation on the metal surface in the presence of light and the π electron cloud, which may be the reason for its decreased stability.

This has now been addressed in the discussion in the revised MS.

2) It is well-accepted that Pd-based catalysts are best for semihydrogenation reaction. Why in this work authors use PtRu rather than Pd or Pd-M alloy?

>> While Pd and Pd-based alloys are undoubtedly well-established catalysts for semi-hydrogenation, the choice of PtRu was motivated by the desire to investigate alternative catalytic systems made up of plasmonic black gold as antenna and RuPt as reactor based on our experience in such plasmonic systems for CO₂ reduction reactions, that could potentially offer advantages in terms of selectivity, stability, and catalytic performance.

Point-by-point Response to the Reviewer # 3

Sharma et al. have reported on the development of RuPtOx nanoparticles on SiO₂-supported Au nanoparticles for the chemoselective hydrogenation of acetylene. This study combines extensive catalyst characterization, in situ spectroscopy, and DFT theory to support their conclusions. Overall, the work is thorough. It appears that their claims are well supported by both experiment and theory. The authors presented a convincing argument for the cooperativity effects of RuOx and Pt sites. It was especially nice to see that this catalyst performs under the industrially relevant condition of excess ethene. The proposed mechanism derived from in situ FTIR is also

logical, although in some regards some assumptions were made. This would warrant a follow-up study, in my opinion, which in an of itself could be highly impactful in the field of plasmonic photocatalysis.

>> We thank the reviewer for their insightful and encouraging comments on our work. It provided a boost to all of us for further development of plasmonic catalysts. We appreciate their thorough evaluation and recognition of our efforts in catalyst development, characterization, and mechanistic exploration.

At times, I found their claims were grandiose and that assertions of the efficacy of their materials were overblown or incomplete. This criticism is particularly true when comparing their work to previous reports (see Comparison of Best Reported Catalytic Systems & Table 1; Table S4). The authors cited references 29, 33-35 in their introduction, yet ignored all but Ref. 35 in their table & description in the main text. Several additional photocatalytic hydrogenation catalysts were reported in Table S4 and it was unclear why specific examples ended up in two tables reporting the same outcomes. I believe it's important for the authors to present a more leveled and unbiased comparison of the literature. This is always true, but especially so when comparing conversions and selectivity of reactions in series where reactor geometry, flow rates, reactant partial pressures, etc. are critical to the conclusions drawn. Many of the studies the authors cited use experimental conditions or different reactor designs that make a quantitative comparison between studies essentially meaningless. As reported, the text presents this work in a very positive light, which it deserves to be, but the data speaks for itself without potentially skewed colorful claims.

>> We apologise for using grandiose statements to present our work, which was unintentional. In the revised submission, we toned down our claims and conclusions and revised the text accordingly across the MS and SI.

We have compared our catalysts, first with only plasmonic systems in Table 1 (which included reference 35). And then, we also compared it with the non-plasmonic catalysts, summarized in Figure 3 and Table S4 (which has more details of figure 3, like reactor geometry, flow rates, and feed composition, as also suggested by this reviewer).

Reference 35 (Pd-Mg/GS) reports a plasmonic catalyst, therefore, it is included in Table 1. Somehow, we missed comparing with ref.33 (Au-Fe/C), which is now added in revised Table 1. Please note changes in reference numbers in the revised manuscript due to the addition of more references, as suggested by reviewer 1.

References 29 (Pd-SA/ N-graphene), 34 (Pd/TiO₂), on the other hand, pertain to non-plasmonic catalysis, hence were included in Figure 3 and Table S4. We have modified the title of the tables to prevent this confusion.

We completely agree with the reviewer about challenges in quantitative comparison. We have now provided as much details as possible in revised Table 1 and Table S4, such as reactor geometry, flow rates, feed composition, etc, to have better comparison. The catalyst performance for all the mentioned catalysts has been evaluated at ambient pressure.

Table 1. Comparative analysis of plasmonic catalytic systems in acetylene semi-hydrogenation.

Catalyst (metal loading and weight)	Light Intensity (λ)	Feed Composition in vol. % (total flow)	Reactor Type	Ethene Production Rate ($\text{mmol g}^{-1} \text{h}^{-1}$)	Acetylene Conversion (%)	Ethene Selectivity (%)	Ref.
DPC/RuPt (10 %, Ru:Pt=9:1, 5 mg)	2.7 W cm^{-2} (400-1100 nm)	$\text{C}_2\text{H}_2/\text{C}_2\text{H}_4/\text{H}_2/\text{Ar} = 2.72/54.5/13.6/24.5/4.5$ (110 mL min^{-1})	Fixed bed flow reactor with quartz window (crucible i.d-6mm)	320	18	88	This Work
		With excess ethene					
DPC/RuPt (10 %, Ru:Pt=9:1, 20 mg)	6 W cm^{-2} (AM1.5)	$\text{C}_2\text{H}_2/\text{C}_2\text{H}_4/\text{H}_2/\text{Ar} = 2.2/44.4/22.2/20/11.1$ (9 mL min^{-1})	Quartz flat cell flow reactor (internal gap is 0.5 mm in the flat section, length-50 mm, width-8mm)	31	97	87	This Work
		With excess ethene					
Pd-Mg/GS (Pd:3 %, 20mg)	1.8 W cm^{-2} (785 nm)	$\text{C}_2\text{H}_2/\text{H}_2/\text{N}_2 = 5/15/75$	Horizontally oriented packed bed reactor with CaF_2 window (6 mm) at the top of the reactor	NR	90	80	37
		$\text{C}_2\text{H}_2/\text{H}_2/\text{N}_2 = 5/20/75$ (Total flow: 5-200 mL min^{-1})		101	NR	NR	
DPC/Ni (Ni-10 %, 35 mg)	0.58 W cm^{-2} (400-1100 nm)	$\text{C}_2\text{H}_2/\text{H}_2/\text{Ar} = 0.12/2/97.8$ (Total flow- 14 mL min^{-1})	Quartz flat cell flow reactor (tube i.d-3.5 mm)	2.4	30	86	16
		Without ethene					
Au-Fe /C (1%, Au:Fe=1:1, 1g (in 2g quartz sand))	0.45 W cm^{-2} (250-1100 nm) T~130 °C, external heating)	$\text{C}_2\text{H}_2/\text{C}_2\text{H}_4/\text{H}_2/\text{Ar} = 1/20/20/59$ (Total flow- 20 mL min^{-1})	Photothermal fixed-bed reactor	0.5	98.4	97.5	35
		With excess ethene					
AINC-Pd (catalyst weight unknown)	14.3 W cm^{-2} (680-1080 nm)	$\text{C}_2\text{H}_2/\text{H}_2/\text{He}/\text{N}_2 = 1.33/3.33/25.33/70$ (Total Flow= 15 mL min^{-1})	Stainless steel gas-phase high temperature reaction chamber	26.7 $\times 10^{-3}$ (mmol h^{-1})	5	97	20
		Without ethene					

NR-not reported, vol.- volume

Table S4. Comparison of the best reported non-plasmonic catalytic systems employed for semi-hydrogenation of acetylene.

Entry	Catalyst (loading, weight)	Temperature (°C)	Feed Composition in vol. % (total flow)	Reactor Type	Ethene Production Rate (mmol g ⁻¹ h ⁻¹)	Acetylene Conversion (%)	Ethene Selectivity (%)	Ref.
1	DPC/RuPt-10-Calc (10 wt%, 5 mg)	T _R -75 Ts-132 with light 2.7 W cm ⁻² 400-1100 nm	C ₂ H ₂ /C ₂ H ₄ /H ₂ /Ar/Air = 2.72/54.5/13.6/24.5/4.5 (110 mL min ⁻¹)	Fixed bed flow reactor with quartz window (crucible i.d-6mm)	320	18	88	This Work
2	DPC/RuPt-10-Calc (10 wt%, 20 mg)	Ts- 262 light 6 W cm ⁻² AM 1.5	C ₂ H ₂ /C ₂ H ₄ /H ₂ /Ar/Air = 2.2/44.4/22.2/20/11.1 (9 mL min ⁻¹)	Quartz flat cell flow reactor (internal gap is 0.5 mm in the flat section, length-50 mm, width-8mm)	31	97	87	This Work
3	Pd/ZnO (1 wt%, 10 mg)	80	C ₂ H ₂ /C ₂ H ₄ /H ₂ /He = 2/40/20/38 (30 mL min ⁻¹)	Fixed bed quartz microreactor (i.d. 4 mm)	131.5	92	89	15
4	PdIn/MgAl ₂ O ₄ (2 wt%, 25 mg)	90	C ₂ H ₂ /C ₂ H ₄ /H ₂ /He = 0.5/50/5/44.5 (120 mL min ⁻¹)	Quartz bed flow reactor	54.9	95	90	16
5	Pd-Pt/SiO ₂ (1.5 wt%, 20 mg)	80	C ₂ H ₂ /H ₂ /He = 1.67/3.33/95 (60 mL min ⁻¹)	Fixed bed flow quartz reactor (1/4 inch)	52.0	97	40	17
6	Pd1/TiO ₂ (SAC) (0.15 wt%, 15 mg)	120 (dark) 60 (light-167 mW cm ⁻² , UV-Vis)	C ₂ H ₂ /C ₂ H ₄ /H ₂ /He = 1/20/10/69 (45 mL min ⁻¹)	Quartz fixed-bed flow reaction chamber equipped with a quartz window (d = 35 mm)	48.2 12.0	100 25	65-50 65	18
7	Pd _{1.0} /Bi ₂ O ₃ /TiO ₂ (Pd 2.3 wt%, Bi 4.9 wt%, 30mg)	44	C ₂ H ₂ /C ₂ H ₄ /H ₂ /He = 1/20/20/59 (60 mL min ⁻¹)	Fixed bed vertical quartz reactor	43.8	91	90	19
8	Pd Single atom/N-graphene (1.04 wt%, 50 mg)	125 (Photothermal: 5.1 W cm ⁻² , UV- Vis)	C ₂ H ₂ /C ₂ H ₄ /H ₂ /Ar = 1/20/20/59 (60 mL min ⁻¹)	Flow reactor with a quartz window at the top for light irradiation. (Volume- 50 cc)	29.7	99	93	20
9	Pd/ND@G (0.11 wt%, 30 mg)	180	C ₂ H ₂ /C ₂ H ₄ /H ₂ /He = 1/20/10/69 (30 mL min ⁻¹)	Quartz bed flow reactor	24.1	100	90	21
10	Pd _{0.006} Cu/SiO ₂ (Pd 0.05wt%, Cu 4.96wt%, 30 mg)	160	C ₂ H ₂ /C ₂ H ₄ /H ₂ /He = 1/20/20/59 (30 mL min ⁻¹)	Quartz reactor	22.7	100	85	22

11	Al ₁₃ Fe ₄ (20 mg)	200	C ₂ H ₂ /C ₂ H ₄ /H ₂ /He = 0.5/50/5/44.5 (30 mL min ⁻¹)	Quartz glass plug-flow reactor (i.d.- 7 mm) catalyst bed supported by a quartz glass frit.	13.6	80	85	23
12	PdZn-1.2@ZIF-8C (0.7 wt%, 50 mg)	120	C ₂ H ₂ /C ₂ H ₄ /H ₂ /Ar = 0.65/50/5/45 (40 mL min ⁻¹)	Fixed-bed quartz tubular reactor	9.4	85	80	24
13	Na-Ni@CHA (Na 6.3 wt%, Ni 3.5 wt%, 200 mg)	180	C ₂ H ₂ /H ₂ /He = 1/16/83 (50 mL min ⁻¹)	Quartz fixed bed flow micro reactor	6.0	100	90	25
14	GaPd/Al ₂ O ₃ (Pd 0.005 wt%, 75 mg)	200	C ₂ H ₂ /C ₂ H ₄ /H ₂ /He = 0.5/50/5/44.5 (30 mL min ⁻¹)	Plug flow reactor consisting of a quartz glass tube with a length of 300mm, i.d.- 7mm and a sintered glass frit to support the catalyst.	3.9	87	85	26
15	Ga ₂ O ₃ -Pd/Al ₂ O ₃ (0.23 wt%, 50 mg)	100	C ₂ H ₂ /C ₂ H ₄ /H ₂ /N ₂ = 0.3/33.1/0.6/66 (50 mL min ⁻¹)	Fixed-bed flow quartz tube reactor	1.5	20	95	27
16	Ni MoS/Al ₂ O ₃ (0.5 wt%, 0.5 g)	125	C ₂ H ₂ /C ₂ H ₄ /H ₂ /He = 0.15/15/3/82 (165 mL min ⁻¹)	Fixed-bed microreactor	1.19	100	90	28
17	Single-atom Pd (0.16 wt%, 1 g)	120	C ₂ H ₂ /C ₂ H ₄ /H ₂ /He = 0.5/50/5/44.5 (20 mL min ⁻¹)	Fixed-bed quartz-glass flow microreactor (i.d. = 6 mm).	0.2	96	93	29

i.d.- internal diameter of the reactor, vol.- volume

Overall, I commend the authors on their thorough work and recommend this for publication with minor revision (at the discretion of the editor).

>> We thank the reviewer for recommending our work for publication.

REVIEWERS' COMMENTS

Reviewer #1 (Remarks to the Author):

The authors have carefully addressed the suggested comments. The manuscript may be considered for publication.

Reviewer #2 (Remarks to the Author):

I believe the authors have tried to address all my concerns and think their revision is publishable now.

Reviewer #3 (Remarks to the Author):

The authors have rigorously addressed my comments and those of the other reviewers. I believe the work is of satisfactory quality for publication in Nature Communications.

2nd REVIEW

Reviewer #1 (Remarks to the Author):

The authors have carefully addressed the suggested comments. The manuscript may be considered for publication.

>> We thank the Reviewer for appreciating the revised version and supporting the manuscript for publication.

Reviewer #2 (Remarks to the Author):

I believe the authors have tried to address all my concerns and think their revision is publishable now.

>> We appreciate the Reviewer's positive feedback on the revised version and their support for the manuscript's publication.

Reviewer #3 (Remarks to the Author):

The authors have rigorously addressed my comments and those of the other reviewers. I believe the work is of satisfactory quality for publication in Nature Communications.

>> We are grateful for the Reviewer's favourable response to the revised manuscript and their endorsement for its publication.

1st REVIEW

We express our gratitude to the reviewers for their encouraging feedback, valuable comments and constructive suggestions, all of which have contributed to the enhancement of our manuscript. The detailed Point-by-point responses to each valuable comment and suggestion are given below.

Point-by-point Response to the Reviewer # 1

The findings and results provided in this manuscript are novel and very interesting. The results provided in the manuscript can have very high impact in the field. In Reviewer's view, the manuscript in its current version has some minor issues. The manuscript may be considered for publication after these minor issues are addressed carefully.

>> We thank the Reviewer for appreciating the novelty and impact of our findings. By addressing their constructive criticism, we were able to further strengthen the quality and impact of our study. Each point is addressed below with a description of the actions taken upon revision.

1. The authors claim that the light enhancement effect reported in the manuscript is due to the plasmonic effect of Au. However, the wavelength-dependent results provided in Figure 1e raises some major questions about this claim. For example, in Figure 1e, productivity is reported to be approximately constant in the 400 nm to 800 nm range. For Au, close to the 400 nm region, it is expected that inter-band transition dominates the optical excitation. Plasmonic excitation starts playing the role only in the later (longer) wavelength region. For example, please see

the following three references for more details. So, the question here is – what dominates the enhancement effect reported in this manuscript? Inter-band excitation or plasmonic excitation or both? This issue needs to be addressed in the manuscript.

(i) Halas and co-workers, “Cu Nanoshells: Effects of Interband Transitions on the Nanoparticle Plasmon Resonance”, *J. Phys. Chem. B* 2005, 109, 39, 18218–18222.

(ii) Jain and co-workers, “Plasmonic photosynthesis of C1–C3 hydrocarbons from carbon dioxide assisted by an ionic liquid”, *Nature Communications*, Volume 10, Article number: 2022 (2019).

(iii) Kolwas and Derkachova, “Impact of the Interband Transitions in Gold and Silver on the Dynamics of Propagating and Localized Surface Plasmons”, *Nanomaterials* 2020; 10(7): 1411.

>> We acknowledge the reviewer's valid concern regarding the need for clarification about the interband and plasmonic activation mechanism and are grateful for bringing this up and also kindly providing these valuable references.

We believe that interband transitions, as well as plasmonic excitations, both occur in the wavelength range ~ 400 nm for black gold, but it is the plasmonic activation that is dominant because of its higher absorption cross-section than the interband transitions. The wavelength-dependent FDTD simulations (Fig. S30) also indicate a significant electric field enhancement in that range. The plasmonic black gold employed in this work shows a broadband absorption profile in the visible range due to plasmonic coupling between different gold nanoparticles at varying inter-particle distances (Fig. S23a), which makes it different from the colloidal Au nanoparticles of certain sizes having narrow band absorption (Fig. S23b), used in Jain and co-workers, *Nature Communications*, Volume 10, 2022 (2019). Other mentioned references also possess a narrow band corresponding to plasmonic absorption centred around 520 nm, with weak absorption in the 400 nm region, while black gold has strong absorption in the 400 nm region (Fig. S23a). Hence, plasmonic excitation dominates interband excitation in black gold, but with contribution from interband excitation also. We have now clarified this crucial point in the revised manuscript.

Figure S23. (a) Broadband UV-Vis extinction spectra of DPC and DPC/RuPt as a dispersion form in ethanol showing broadband absorption in the visible range; (b) Narrow UV-Vis extinction spectrum of a colloid of the Au NPs used for the preparation of the photocatalyst film. The spectrum exhibits a localized surface plasmon resonance (LSPR) band centered around 520 nm, as indicated by the dotted line (Adapted with permission from Jain and co-workers, *Nat. Commun.*, **10**, 2022 (2019), copyright 2019 Springer Nature)

2. The authors claim that (on page #2), “The catalytic activity showed superlinear dependence of ethene production on the light intensity with a power law exponent of 2.37 ($\text{rate} \propto I^n$), indicating a hot electron-mediated non-thermal pathway (Fig. 1c, Fig S16a).” The question that the Reviewer would like to raise here is - what is the rationale for this statement? For example, a similar trend can also be expected in purely thermal effects. For example, please see the following five references for more details. So, the authors need to provide clarity to this claim in the revised manuscript.

(i) Dubi, Y.; Un, I. W.; Sivan, Y. Thermal Effects – an Alternative Mechanism for Plasmon-Assisted Photocatalysis. *Chem. Sci.* 2020, 11 (19), 5017–5027.

(ii) Dubi, Y.; Sivan, Y. “Hot” Electrons in Metallic Nanostructures—Non-Thermal Carriers or Heating? *Light Sci Appl* 2019, 8 (1), 89.

(iii) Aizpurua, J. et al. Theory of Hot Electrons: General Discussion. *Faraday Discuss.* 2019, 214, 245–281.

(iv) Aizpurua, J. et al. New Materials for Hot Electron Generation: General Discussion. *Faraday Discuss.* 2019, 214, 365–386.

(v) Aizpurua, J. et al. Dynamics of Hot Electron Generation in Metallic Nanostructures: General Discussion. *Faraday Discuss.* 2019, 214, 123–146.

>> We extend our appreciation to the reviewer for their insightful comment. The superlinear dependence on light intensity, signifying a non-thermal pathway mediated by hot electrons, has been observed by numerous researchers and is a widely accepted phenomenon for comprehending these plasmonic effects. Nevertheless, as the referee correctly pointed out, there is still an ongoing debate among some authors regarding this matter, where a superlinear dependence trend was claimed to be expected in purely thermal effects. As suggested by the reviewer, we have now tried to clarify these critical aspects in the revised MS.

Sivan et al., in their report, *Thermal Effects – an Alternative Mechanism for Plasmon-Assisted Photocatalysis*. *Chem. Sci.* 2020, 11 (19), 5017–5027, explain the superlinear dependence of reaction rate on intensity observed by Linic and co-workers (*Christopher et al., Nat. Mater.*, 2012, 11, 1044–1050) by assuming a purely thermal behaviour and the proposed temperature-shifted Arrhenius Law. They discussed heat distribution under light illumination in these systems and introduced the shifted Arrhenius equation as follows:

$$R \sim \exp\left(-\frac{\varepsilon_a}{k_B T(r) + a I_{inc}}\right) \quad (1)$$

This equation has been adjusted to account for heating induced by illumination.

They propose a linear temperature increase model-

$$T(I_{inc}) = T_M + b(I_{inc}) \quad (2)$$

where $T(I_{inc})$ is the actual temperature of the reactor in light and T_M , is the experimentally measured temperature.

To check the applicability of Sivan's model in the current study, we re-evaluated our intensity-dependent and temperature study conducted in the absence of light (Fig. S17a). If we assumed a purely thermal behaviour and correlate the measured temperature under light conditions (T_M) and the temperature at which similar activity was observed in the absence of light ($T(I_{inc})$), we observed that the difference between these two temperatures did not conform to the anticipated linear trend at higher intensities (Fig. S17b), as suggested by the Sivan's model presented in eq. (2). This observation clearly contradicts the assumption of a linear temperature increase model proposed by Dubi and Sivan et al.

Figure S17. (a) Acetylene conversion in light and dark versus the catalyst bed temperature, (b) Temperature difference between $T(I_{inc})$ and T_M versus light intensity.

Similar observations were also made by Halas et al. (*Response to Comment on “Quantifying hot carrier and thermal contributions in plasmonic photocatalysis”, Science 2019 9545*) that a linear model can only be applied for small temperature increases (<100 K). In the low-intensity regime, where the trend is indeed linear, the role of non-thermal effects can still not be neglected if the light dependence of activation energy is considered, as discussed in a comment on their study by Jain et al. (*Comment on “Thermal Effects – An Alternative Mechanism for Plasmon-Assisted Photocatalysis” by Y. Dubi, I. W. Un and Y. Sivan, Chem. Sci., 2020, 11, 5017. Chem. Sci. 11, 9022 (2020)*).

In their comprehensive analysis, Jain et al. delved into this matter in great detail, starting with the premise that a decrease in activation energy correlates linearly with light intensity in the field of photocatalysis. This assumption formed the foundation for a detailed examination of the complex relationship between light-induced effects and catalytic performance.

$$E_a = E_a^{dark} - BI \quad (3)$$

Here, B is a proportionality constant with units of $\text{eV cm}^2 \text{W}^{-1}$ if E_a is expressed in units of eV and I in units of W cm^{-2} . B is a wavelength-dependent quantity. On further solving the above and putting this in Arrhenius equation, we get:

$$E_a = E_a^{dark}(1 - bI) \quad (4)$$

b is $\frac{B}{E_a^{dark}}$ and has units of $\text{cm}^2 \text{W}^{-1}$.

$$R = A \exp\left[-\frac{E_a^{dark}}{k_B T_s}(1 - bI)\right] \quad (5)$$

After using a Taylor expansion around $I = 0$, which is a dark condition, we get,

$$\frac{1}{1-bI} = 1 + bI + (bI)^2 + (bI)^3 + \dots \quad (6)$$

Higher-order terms can be ignored in the light-intensity regime ($I < 1/b$). After putting this in equation (5), we get

$$R = R_0 \exp\left[-\frac{E_a^{dark}}{k_B T_s(1+bI)}\right] \quad (7)$$

When this equation is compared to the general Arrhenius equation, the reaction appears to take place at a theoretical temperature that is proportional to the light intensity (I) higher than the actual temperature T_s .

$$T_{dummy} = T_s(1 + bI) \quad (8)$$

This equation, which happens to be the same one used by Sivan and their colleagues in their research, reinforces the case for the thermal pathway over the non-thermal pathway. Within the mathematical context of this equation, the conclusion is that light excitation exclusively leads to an increase in temperature without any influence on activation energy. This mathematical treatment of the Arrhenius equation effectively curtains the potential non-thermal effects of light excitation, masking them under the guise of temperature increase, i.e. the thermal pathway.

Moreover, we also studied the non-thermal activation mechanism by the KIE measurement (Fig. 4d) as well as the photocatalytic studies with external heating (Fig. 4f). Therefore, we believe that the superlinear dependence of ethene production on the light intensity, characterized by a power law exponent of 2.37 ($\text{rate} \propto I^n$), suggests a

hot electron-mediated non-thermal pathway, alluding to the possibility of multi-photon absorption as observed in Linic et al., *Singular characteristics and unique chemical bond activation mechanisms of photocatalytic reactions on plasmonic nanostructures*, *Nat. Mater.*, 2012, 11, 1044–1050.

A brief of the above discussion on the explanation of the superlinear dependence of reaction rates on light intensity with both points of view has been included in the revised manuscript.

3. The authors also claim that (on page #2), “Notably, the quantum efficiency of this reaction first increased with an increase in light intensity (up to 3 W cm⁻²) and then decreased with a further increase in light intensity (Fig. S16b). This indicated the transition from non-thermal to thermal pathways at higher intensities.” It is not clear to the Reviewer, how this change in the quantum efficiency is indication of a transition from the non-thermal pathway to the thermal pathway? The authors may need to provide references for this. Alternatively, the authors may revisit their analysis – both non-thermal and thermal effects may play in both regions and one effect may play the dominant role in comparison to the other effect in some region.

>> We fully agree with the reviewer that both non-thermal and thermal effects play key roles during plasmonic catalysis, with the non-thermal effect dominated at lower light intensity while the thermal effect dominated at higher light intensity.

Our results primarily indicate that there is a specific light intensity range in which the majority of incident photons actively engage in catalysing the reaction (Fig. S18). The increase in the quantum efficiency up to 3W cm⁻² (Fig. S18b), can be explained by the electron-assisted reaction over the plasmonic DPC/RuPt, as also explained in one of the reports by Linic et al, *Singular characteristics and unique chemical bond activation mechanisms of photocatalytic reactions on plasmonic nanostructures* *Nat. Mater.*, 2012, 11, 1044, where the transition to the superlinear regime was found to be accompanied by an increase in quantum efficiency (Fig. S18a). Specifically, we observed that the disparity between the ethene production rate in light and dark diminishes at higher light intensities where the catalyst bed temperature is ~ 200 °C. At higher intensity, an increase in photon flux could lead to more photo-induced heating of the catalyst, increasing the local temperature. In this high-temperature regime, the reaction might be constrained due to the limited adsorption of acetylene on the catalyst surface (more desorption due to high local temperature), thus resulting in reduced values of quantum efficiency.

Figure S18. (a) Quantum efficiency (%) as a function of light intensity for various temperatures. The dotted lines show the quantum efficiency observed at source intensities $<\sim 300$ mW cm⁻² for ethene epoxidation over Ag nanocubes (Adapted with permission from Linic et al. Singular characteristics and unique chemical bond activation mechanisms of photocatalytic reactions on plasmonic nanostructures. *Nat. Mater.* **11**, 1044–1050 (2012), Copyright 2012 Springer Nature); (b) Quantum efficiency at different light intensities and (c) moles of acetylene converted (log scale) v/s light intensity (log scale) used at different reactor temperatures (T_R), for acetylene semi hydrogenation in excess ethene over DPC/RuPt-10-Calc

Thus, we agree that both non-thermal and thermal pathways contribute to catalysis. The predominance of the thermal pathway in the higher intensity regime was based on the difference in the catalytic activity in light and dark at similar catalyst bed temperatures, which is also highlighted in Fig. S17a and Fig. S18c.

A brief of the above discussion on the explanation of Quantum efficiency dependence and related references have been included in the revised manuscript.

4. Does Figure 2c represent the UV-Vis extinction spectrum or absorption spectrum? Please see the reference below for the distinction. How was this spectrum measured in solid sample or particles dispersed in the dispersed liquid phase? The authors may consider including more details on this in the supporting information. Also, from this data (Figure 2c), what is the expected region of wavelengths to see the plasmonic effect? Does this Figure

support any of the reaction data, such as productivity at different wavelengths? Reference (i): U. Aslam, S. Chavez, S. Linic, Nature Nanotech 2017, 12, 1000.

>> Figure 2c is the UV-Vis extinction spectrum. The baseline subtraction process was carried out (to reduce potential scattering effects) by first recording the UV-Vis extinction spectrum of DFNS/APTES dispersion in ethanol (which is the solid support to prepare black gold) and then subtracting it from the UV-Vis extinction spectrum of DPC (black gold) and DPC/RuPt-10-Calc dispersion in ethanol. As per the reviewer's suggestion, these details have been included in the revised supporting information as well as in the revised Fig. 2.

Figure 2c. UV-Vis extinction spectra of DPC and DPC/RuPt as a dispersion form in ethanol showing broadband absorption in the visible range.

Figure 1e. Effect of the wavelength of light on acetylene semi-hydrogenation and corresponding catalyst bed temperatures (using optimized flow conditions at 2.7 W cm^{-2})

The UV-Vis extinction spectrum shows broadband plasmonic behaviour of the black gold catalyst in the entire visible region (400-800nm) (Fig. 2c). The wavelength-dependent ethene production data matches well with this extinction spectrum, which shows similar activity at different wavelengths spanning the visible region (Fig. 1e). This aspect is now discussed in the revised manuscript.

5. The authors provided a plasmonic mechanism on page #6. But it is not mentioned whether this mechanism is expected to occur via the Photoredox mechanism or the MGR-type mechanism. This distinction is very important for the readers to visualize the mechanism. Please see the following review article for the distinction of these mechanisms. Also, what is the expected rate limiting step (RLS) for this reaction? Reactant activation (dissociation) or product desorption or surface reaction? How does the non-thermal effect enhance this RLS? Reference (i): Ramakrishnan, S. B. et al. Photo-induced Electron and Energy Transfer Pathways and Photocatalytic Mechanisms in Hybrid Plasmonic Photocatalysis. Advanced Optical Materials 2021, 2101128.

>> We appreciate the reviewer's insightful question about Photoredox Vs. MGR-type mechanism.

In the dark case, larger KIE was observed at lower temperatures, suggesting the H₂ dissociation or acetylene hydrogenation a rate-limiting step (RLS) at lower temperatures. Whereas, the desorption of ethene seems RLS at higher temperature since almost no KIE was observed at 200 °C. On the other hand, in the presence of light, KIE was seen at any light intensity (or temperature range), indicating that the desorption of ethene seems not the RLS under light irradiation. Based on these results, light irradiation promotes the desorption of ethene, thus, the H₂ dissociation or acetylene hydrogenation step is the RLS under light irradiation.

We believe that the ethene desorption step is accelerated by a polarizing electric field on the catalyst surface, indicating the role of non-thermal effects in enhancing rate limiting step (RLS). A similar observation was made by Hu, C. et al. *Near-infrared-featured broadband CO₂ reduction with water to hydrocarbons by surface plasmon. Nature Communications 14, 221 (2023)*. They observed that the electric field pointing toward the plasmonic metal surface redistributed the electrons of CO₂ from the O lone-pair orbital to its antibonding π^* orbital, weakening the C–O bond and increasing the bond length. When the electric field pointed away from the metal surface, the lone-pair electrons transferred to the σ orbital between O and the metal atom making the desorption of the product, CO facile. Such polarising effect of the enhanced electric field around the active site might be happening in our catalytic system, facilitating the C-O and O-H bond breaking to form TS(II), consequently leading to the desorption of hydrogenated product ethylene.

As per DFT calculations (Figure 7b), ethene desorption seems to be the rate limiting step (RLS) in dark. However, ethene desorption is dependent on the availability of dissociated hydrogen, which is enhanced by transient charge transfer to H₂ via Pt. It is evident from the DFT calculations that charge gets accumulated on Pt in dark (Fig. S54), which will be even higher in light. The higher kinetic isotope effect (KIE) observed for H₂/D₂ in light (as illustrated in Fig. 4d) also indicated enhancement in H₂ dissociation assisted by high-energy electrons. These observations indicated that the reaction follows the MGR-type mechanism as nicely discussed in the review by Ramakrishnan, S. B. et al. *Adv. Optical Mater. 2021, 2101128*.

These discussions and related data are now included in the revised manuscript.

Figure S54. Calculated charge density difference between Ru and RuPt surface showing larger charge accumulation on Pt. Pink and Green colors indicates charge accumulation and depletion, respectively. Blue-Ru and Orange-Pt.

Figure 7. Proposed reaction pathway for acetylene semi-hydrogenation on DPC/RuPt-10-Calc. a) Formation of various reaction intermediates showing the role of co-existing oxide and reduced species based on in-situ FTIR studies; b) energies of initial (IS), transition (TS), intermediate (IM), and final (FS) states formed in the acetylene semi-hydrogenation by DFT calculation using nudged elastic band method.

6. The reaction data under dark conditions (pure thermal conditions) are provided in Figure 1d. For these pure thermal conditions, Arrhenius-type relation is expected between the rate and temperature. But the productivity/rate data at reaction temperatures of 200, 220 and 250 C show a relatively small increase. What are the possible reasons for this? Does the sintering of catalysts occur at these high temperatures?

>> Inspired by the reviewer's thoughtful comment, we conducted the TEM imaging and EDS mapping of the catalyst after the reaction at a higher temperature (250 °C for 3 h) and we did not observe sintering of the active sites as shown in Fig. S19.

At higher temperatures above 200 °C, acetylene desorption is dominated over adsorption, which limits the overall rate of the reaction. We have now clarified this point in the revised manuscript and included Fig. S19 in the revised Supplementary Information.

Figure S19. HAADF STEM image and EDS elemental maps of DPC/RuPt-10-spent-250C-3h

7. Do authors see any evidence for H₂ oxidation or hydrocarbon oxidation while analyzing their product analysis? Or, on what basis these possible parallel oxidation reactions are ruled out?

>> We analysed the products via gas chromatography with TCD, FID, as well as mass detectors. No CO₂ was observed (other than 400 ppm which is present in air feed), indicating no hydrocarbon oxidation.

We believe that H₂ oxidation is taking place, yielding water due to continuous regeneration (oxidation and reduction) of active sites (Ru to RuO₂) during the catalysis (Fig. 7a). We carried out the light-induced reduction

of DPC/RuPt-Calc-10 (Figure 5e), and observed the water formation. We have highlighted the discussion about the by-products in the revised manuscript and revised Fig 5e.

Figure 5e. Light-induced reduction of DPC/RuPt-10-Calc in H₂ at light illumination $\sim 2.7 \text{ W cm}^{-2}$ (400-1100nm)

8. Minor suggestion: In the catalysis and photocatalysis word “Rate” is used instead of the word “Productivity” as used by the authors in the manuscript. The authors may consider changing the word “Productivity” to in terms of “Rate of reaction” or “Product yield

>> As per the reviewer’s kind suggestion, we have changed the word “productivity” to “production rate” in the revised manuscript.

Point-by-point Response to the Reviewer # 2

In this manuscript Sharma and coauthors reported the synthesis of PtRu alloy nanoparticles loaded on “black gold” as plasmonic nano-reactor for the semihydrogenation of acetylene in excess ethylene. In this work the authors performed abundant study including in terms of catalyst characterization, catalytic performance and mechanism study, including both experimental and computational studies. In summary they have three major findings, i.e., 1) the plasmonic hydrogenation of acetylene into ethylene with high performance as they claimed while with which I cannot agree; 2) they found significant air flow dependent catalyst stability which I think quite interesting but with less convincing explanation in the current study; 3) significant synergistic effect between Pt and Ru without convincing explanation. In my humble opinion, these three parts are not closely related and authors might be able to report them in three independent papers. Put them in one manuscript will make the manuscript complicated and overloaded thus losing the key point of a work. In addition, there were many critical issues for the whole work so that I cannot recommend the acceptance of this work at its current state.

>> We are grateful for the reviewer’s comment that he finds the work extensive and suitable for three independent publications. Our aim was to perform an in-depth study of this unique plasmonic catalytic system, which would be suitable for a prestigious journal like Nature Communications.

The major issues are listed below for the authors’ consideration.

1. For the first parts, i.e., the plasmonic reaction.

1) For acetylene semihydrogenation, an important point is that the selectivity is highly dependent on the acetylene conversion. For many or even most semihydrogenation catalysts, ethylene selectivity could be very high as long as the acetylene conversion is less than unity. However, once the conversion approaches to 100%, the selectivity will decrease significantly to a much negative value due to the hydrogenation of a large amount of ethylene. Therefore, one of the major issues in this work is that for the plasmonic hydrogenation the acetylene conversion is rather low (often 20%), thus the high ethylene selectivity is unrepresentative and thus less meaningful. Authors may want to examine the selectivity in full conversion. In addition, for the practical application, a full conversion is necessary as it usually requires a very low concentration of acetylene (<5 ppm).

>> We agree with the reviewer that acetylene conversion is an important criterion for determining the applicability of the catalyst in practical situations. However, our main focus was on understanding the role of plasmonic excitation for semihydrogenation reactions using solar energy at ambient conditions. Hence, we did not attempt to increase the conversion to 100%.

Inspired by the reviewer’s valid criticism, we now tried to increase the conversion. We realized that in our fixed bed flow chamber reactor, we could only achieve ~80% conversion with ~90% selectivity. This was due to the limitation of the reactor set-up used in terms of light-exposed catalyst area. We have now used a different reactor set-up, a thin flat cell made up of a quartz tube (Fig. 27a), in which we could achieve ~97% conversion, maintaining high selectivity (Fig. 27b).

We now hope that the reviewer will appreciate nearly a full conversion achieved and this is now added in the revised MS and SI.

Figure S27. (a) Quartz flat cell flow reactor set-up employed to increase the illuminated catalyst area (b) Acetylene conversion and ethene selectivity trend showing high selectivity being maintained at high conversion. The highest conversion achieved with total flow= 9 mL min⁻¹, C₂H₂/C₂H₄/H₂/Ar/Air=0.2/4/2/1.8/1 and AM1.5 light illumination at 1 bar pressure.

2) Another major issue is that authors claim their catalysis is highly active (the so-called first-of-its-kind). However, Table S1 suggested that quite a few catalysts are similar to or even more active than theirs. For example, their catalyst gives an ethylene productivity of 320 mmol/gcat/h with metal loading of 10 wt%. However, Pd/ZnO (entry 2) and Pd/TiO₂ (entry 5) give 131.5 and 48.2 mmol/gcat/h with only 1 wt% and 0.15 wt% metal loading, respectively. Therefore, if the activity was normalized by metals, the latter two will have much higher activity (10 and 66 times, i.e., 1215 and 3213). The same for Pd_{0.006}Cu/SiO₂ (entry 9) and GaPd/Al₂O₃ (entry 13) despite their activity at a slightly higher temperature. On the other hand, other catalysts such as entry 3, 4, 7 and 8 have similar activity. So their catalyst didn't have superior activity, not mention the high cost of the black gold support. Therefore, I think the high activity of their catalyst might be overstated.

>> The main focus of our work was to develop plasmonic acetylene semi-hydrogenation using solar light (without any external thermal energy). When we compared our catalysts with all reported plasmonic catalysts, Table 1 in the MS, it clearly indicated the significantly high activity as compared to the best-known plasmonic catalysts. Hence, we claimed the best catalysts reported.

The term “*first-of-its-kind*” was used not in terms of productivity but due to its stable behaviour in the air (even during hydrogenation reaction), since this was not observed before, indicating a unique possibility during plasmonic catalysis over conventional catalysis.

We apologise for the overstatements, which were unintentional. In the revised manuscript, we have removed all such text which sounded overstatements and toned down our claims and conclusions.

Regarding the calculations of productivity per gram of catalyst Vs. per gram of active sites, we fully understand the reviewer's viewpoints, which was actually our big dilemma while analysing our results. We contemplated extensively to decide how to compare our catalysts with reported ones and then decided to use productivity per gram of catalyst and not per gram of active sites. The reason is that in our black gold/RuPt catalysts, every single component (Gold, Ru, and Pt) plays a critical role and they can not work independently and catalyst is only active when all three components are together. Hence, we decided not to refer only to Ru or Pt as active sites. Therefore, we opted for the productivity comparison by taking the entire catalyst weight rather than one metal component.

Similar observations were also made by references cited by the reviewer. For example, Pd/ZnO catalyst was characterized as an intermetallic rather than supported metal catalyst, which implies that both Pd and ZnO play active roles. Therefore, focusing solely on Pd as the active site may not be accurate, a point similarly applicable to Pd_{0.006}Cu/SiO₂ and GaPd/Al₂O₃ catalysts. In our catalytic system, there are three metals: Au, Ru and Pt and only considering Pt as the active site would be inaccurate as without the other components, black gold and Ru, the desired acetylene conversion and selectivity cannot be achieved.

Also, several other factors need to be considered in the assessment of catalyst performance with reported catalysts, like the mode of activation and the energy input required. We were able to carry out the reaction purely using solar light (without any external heating) and it surpasses the other catalysts in terms of stability and selectivity, up to 100 hours, in contrast to the 20-hour testing duration for Pd/ZnO.

Regarding the cost of the black gold support, it is important to consider it as a one-time investment (like any other noble metal catalyst), particularly given the long-term stability of our catalyst.

Additionally, the photocatalytic nature of our catalyst, using solar light as an energy source (without any external thermal energy), offers the advantage of eliminating the need for additional expenses associated with the fuel required for conventional thermal reactions commonly used in industry.

3) The KIE data were not fully analyzed. Firstly, the KIE results of H₂-D₂ can only reflect the photo-thermal activation of H₂ rather than other reactants (last sentence on Page 6). In addition, the order of KIE can determine whether H₂ dissociation is involved into the rate-determining step. Authors may want to discuss this data more carefully.

>> We are grateful to the reviewer for bringing up this important point, as also pointed out in Comment-5 of Reviewer #1. The KIE results of H₂-D₂ indicate that there is a significant difference in the thermal and non-thermal mechanism of plasmonic activation for H₂ dissociation. Due to the unavailability of labelled acetylene, similar measurements could not be performed for acetylene activation.

In the dark case, larger KIE was observed at lower temperatures, suggesting the H₂ dissociation or acetylene hydrogenation a rate-limiting step (RLS) at lower temperatures. Whereas, the desorption of ethene seems RLS at higher temperature since almost no KIE was observed at 200 °C. On the other hand, in the presence of light, KIE was seen at any light intensity (or temperature range), indicating that the desorption of ethene seems not the RLS under light irradiation. Based on these results, light irradiation promotes the desorption of ethene, thus, the H₂ dissociation or acetylene hydrogenation step is the RLS under light irradiation.

We believe that the ethene desorption step is accelerated by a polarizing electric field on the catalyst surface, indicating the role of non-thermal effects in enhancing rate limiting step (RLS). A similar observation was made by Hu, C. et al. *Near-infrared-featured broadband CO₂ reduction with water to hydrocarbons by surface plasmon. Nature Communications 14, 221 (2023)*. They observed that the electric field pointing toward the plasmonic metal surface redistributed the electrons of CO₂ from the O lone-pair orbital to its antibonding π^* orbital, weakening the C–O bond and increasing the bond length. When the electric field pointed away from the metal surface, the lone-pair electrons transferred to the σ orbital between O and the metal atom making the desorption of the product, CO facile. Such polarising effect of the enhanced electric field around the active site might be happening in our catalytic system, facilitating the C-O and O-H bond breaking to form TS(II), consequently leading to the desorption of hydrogenated product ethylene.

As per DFT calculations (Figure 7b), ethene desorption seems to be the rate limiting step (RLS) in dark. However, ethene desorption is dependent on the availability of dissociated hydrogen, which is enhanced by transient charge transfer to H₂ via Pt. It is evident from the DFT calculations that charge gets accumulated on Pt in dark (Fig. S54), which will be even higher in light. The higher kinetic isotope effect (KIE) observed for H₂/D₂ in light (as illustrated in Fig. 4d) also indicated enhancement in H₂ dissociation assisted by high-energy electrons. These observations indicated that the reaction follows the MGR-type mechanism as nicely discussed in the review by Ramakrishnan, S. B. et al. *Adv. Optical Mater. 2021, 2101128*.

Figure S54. Calculated charge density difference between Ru and RuPt surface showing larger charge accumulation on Pt. Pink and Green colors indicates charge accumulation and depletion, respectively. Blue-Ru and Orange-Pt.

These discussions and related data are now included in the revised manuscript.

Figure 7. Proposed reaction pathway for acetylene semi-hydrogenation on DPC/RuPt-10-Calc. a) Formation of various reaction intermediates showing the role of co-existing oxide and reduced species based on in-situ FTIR studies; b) energies of initial (IS), transition (TS), intermediate (IM), and final (FS) states formed in the acetylene semi-hydrogenation by DFT calculation using nudged elastic band method.

These discussions have been included in the revised manuscript.

4) Figure S28 suggested that the activation energy (E_a) is temperature dependent, so I am curious how did the authors obtain the different E_a values at certain temperatures?

>> We are grateful for the reviewer's question. The Arrhenius plot in dark and light (Fig. 4e) indicates a non-linear behaviour in the given range of temperatures (*Ref. Truhlar, D. G., Kohen, A. Convex Arrhenius plots and their interpretation. Proc. Natl. Acad. Sci. U. S. A. 98, 848–851 (2001)*).

We plotted $\ln(k)$ versus $1000/T_s$ and apparent activation energy (E_a) was calculated by using the Arrhenius equation,

$$E_a = -\frac{R \, d\ln(k_{C_2H_4})}{d(1/T_s)} \quad (\text{J mol}^{-1}) \quad (9)$$

where $k_{C_2H_4}$ is ethene production rate.

The rate was first obtained as a function of temperature by fitting the data to a polynomial function (Fit shows $\ln K$ is a quadratic polynomial in terms of $1/T_s$).

$$\text{In Light: } -24.78 + 28.04 (1000/T_s) - 6.41(1000/T_s)^2 \quad (10)$$

$$\text{In Dark: } -17.51 + 24 (1000/T_s) - 6.15 (1000/T_s)^2 \quad (11)$$

Differentiating the obtained expression as per the equation given in (9), E_a can be obtained to have a linear dependence on $1/T_s$ and the exact expressions are given below and plotted in Fig. S32.

$$\text{In Light: } -233.2 + 106.6 (1000/T_s) \quad (12)$$

$$\text{In Dark: } -199.5 + 102.3 (1000/T_s) \quad (13)$$

The plot of Fig. S32 is a differential plot of Fig. 4e.

These points are now clarified in the revised manuscript and supporting information.

5) On page 6 authors found for partially reduced sample the enhancement degree of local electronic field is lower than that on oxide samples, this is in contrast to the activity increase by adding air flow.

>> We think that there is some confusion here in the nomenclature of samples. We compared fully reduced and partially reduced samples (and not oxide sample), as both metal and oxide phases play key role in catalysis.

The FDTD simulation data shows the electric field enhancement in the fully reduced catalyst (DPC/RuPt-25.6) is more than that in the partially reduced catalyst (DPC/RuPtOx-19.2) because of the greater extent of near field coupling with higher reduced metal content. It should be noted that the actual catalyst is a combination of the reduced and the oxidised form and both have their role to play. When only the reduced form is present, acetylene adsorption is limited since it is the oxidised form, which is crucial for the formation of the di- σ -bonded acetylene (Fig. 6, 7a), while the reduced form facilitates plasmon-assisted H_2 dissociation. Therefore, a partially reduced

catalyst performs better than a fully reduced one (the catalyst is deactivated without air due to complete reduction) without compromising significantly on the electric field enhancement (Fig. 4).

Figure 4. Electric field enhancement in a) DPC, b) DPC/RuPt (fully reduced), and c) DPC/RuPtOx (partially reduced) (Source $E_0 = 4509 \text{ V m}^{-1}$ at 2.7 W cm^{-2}) using FDTD simulations

The caption of Fig. 4 has been modified in the MS to avoid this confusion.

6) I have problem in understanding the activity became saturated with a conversion rate close to 400 mmol/g/h. What's the corresponding conversion of acetylene? Only reaching 100% one can claim the "saturated activity".

>> We evaluated the catalyst's activity based on i) the maximum rate of product formation it can achieve and ii) the highest conversion achievable. While they both may be interlinked, they are significantly different. The optimised gas hourly space velocity (GHSV) is different for full conversion and for the highest ethene production rate. At higher flow rates (higher GHSV), the interaction time is less, but the amount of the reactant molecules are in large quantities, which can react with a large number of active sites, leading to maximum productivity (but lower conversion). Whereas at low flow rates, the interaction time is greater, and a lesser number of reactant molecules are available to interact with the active sites; hence, the conversion is high (accompanied by low productivity).

The reason for choosing the high productivity regime for the study of thermal and non-thermal pathways of plasmonic activation was to understand the changes in the active site behaviour better when most of them are being utilised.

In our study, the maximum ethene production rate was achieved when acetylene conversion rate $\sim 400 \text{ mmol g}^{-1} \text{ h}^{-1}$ at $\sim 25\%$ conversion (Total flow rate = 110 mL/min). This lower value of conversion was due to the high GHSV used as well as the limitation of the reactor used in terms of light-exposed catalyst area. When we used different reactor set-up, a thin flat cell made up of quartz tube and low flow (Total flow rate = 9 mL/min) (Fig. S27b), we could achieve $\sim 97\%$ conversion maintaining high selectivity.

The maximum conversion data is now included in the revised MS and the statement: the activity became saturated with a conversion rate close to 400 mmol/g/h has been modified to avoid confusion.

2. For the air flow enhancing catalyst stability

1) The adding of air into hydrogenation reaction is seriously dangerous. Authors should provide the concentration of various contents after adding air flow and the explosion limits of various contents.

>> We thank the reviewer for raising this very serious concern. This was also our concern while doing the experiments with already highly flammable acetylene. But the concept of plasmonic catalysts seems to be helping here.

As the reviewer correctly pointed out, acetylene, ethene and hydrogen are highly flammable gases with lower flammable limit (LFL) and upper flammable limit (UFL) values of 2.5/100, 2.7/36 and 4/18.3 (vol% in the air) and should explode in the presence of sufficient air. However, their ignition temperatures are 305, 490, and 500 °C respectively. (Rowley, Jeffrey R., "Flammability Limits, Flash Points, and Their Consanguinity: Critical

In our plasmonic catalysis, when operating under optimized conditions, we only reach a temperature of 132°C, with only 1% O₂ present in the feed. As a result, there is no risk of an explosion. This highlights another advantage of using plasmonic catalysis, as it allows for the safe use of air even in the presence of highly flammable gases, which is not feasible with conventional thermal catalysis.

Figure S47. Dependence of ignition temperature of acetylene on its concentration in atmosphere (reproduced from Jones G. W. ; Miller, W. E. Ignition temperatures of acetylene-air and acetylene-oxygen mixtures. Report of Investigations, Bureau of Mines, May 1941)

This crucial point has now been addressed in the discussion for air stability in the revised MS.

2) The H₂-TPR results are hard to understand, why authors didn't provide the TCD signals and MS signals of H₂O? It seems the reduction occurred at room temperature as the H₂ signal increase with reduction temperature. Also the time should change to temperature in H₂-TPR figures.

>> Thanks for raising this important concern and sorry for not presenting this data accurately. Kindly note that this is not conventional temperature programmed reduction (TPR) using external thermal energy, but it is reduction using solar energy (without any external heating), so it can be named as a light programmed reduction (LPR).

When the light was off, the reactor was at room temperature (30 °C) and no reduction took place. As soon as light (400-1100 nm, 2.7 W cm⁻²) was switched on, the temperature increased to 132 °C (within 20 seconds) and the reduction took place immediately. Hence, the X-axis was time. We have now added a secondary X-axis for temperature, showing temperature corresponding to light on and off conditions.

On the reviewer's suggestion, we also added TCD and mass signals of H₂O in the plot, illustrating the reduction of the catalyst through a decrease in the H₂ mass signal and a simultaneous increase in the H₂O mass signal.

Figure 5e. Light induced reduction of DPC/RuPt-10-Calc in H₂ at light illumination $\sim 2.7 \text{ W cm}^{-2}$ (400-1100nm)

The figure has been modified accordingly in the revised MS.

3) According to Figure 1f, the true active metal for this reaction is Pt. Why did authors only analyze the change of Ru to determine the effect of air flow in catalysts stability? I think author should clarify the role of Pt and Ru in PtRu alloy for this reaction which is related to the 3rd question (see below).

For PtRu alloy, 1) Figure 1f suggested that Pt and PtRu have almost same activity (productivity of ethylene+ ethane) but large different selectivity. Therefore, I think the deactivation without air flow and non-deactivation with the presence of air flow might be restudied based on revealing the separate role of Pt and Ru, and probably focusing on the role of Pt rather Ru.

>> Thank you for these important comments. Please note that we studied both Ru and Pt sites using XRD, XPS and XANES as well as in-situ FTIR, DRIFTS and TEM. However, we only observed changes in the oxidation state of Ru and not Pt in the presence or absence of airflow.

We agree with the reviewer that Pt alone can hydrogenate acetylene to ethylene, but it also hydrogenates ethylene to ethane, while our main goal was to achieve acetylene semi-hydrogenation and stop at the ethene stage, which was not possible by Pt sites. When we prepared Pt-doped Ru nanoparticles containing RuO₂ phases, acetylene was found to chemisorb on RuO₂ (and not on Pt) and only H₂ was dissociated at Pt (of RuPt). Hence, in our

catalytic system, RuO₂ and RuPt (activated by black gold) are active sites for acetylene activation and H₂ dissociation, respectively. This was also confirmed by DRIFTS, transmission FTIR, and DFT studies.

Based on these observations, we propose that during acetylene semi-hydrogenation (Figure 7), a partially oxidized RuPt alloy surface with a terminal hydroxyl group interacted with acetylene and generated di- σ -bonded acetylene via the loss of 2H from the hydroxyl groups and the breaking of the triple bond to form two C-O bonds (step i). The hydrogen lost from hydroxyls was then utilized to form a $=C-H$ bond by breaking a C-O bond (step ii). After both the C-O bonds were broken, the ethene formed interacted with the O in a π -bonded fashion (step iii). After supplying external H₂, which was dissociated by the Pt sites of the RuPt catalyst, the hydroxyls were regenerated, and the ethene molecules were released, and the catalyst became available for the next cycle (step iv). RuPt NPs could thus facilitate the dissociation and migration of H radicals across the surface in a controlled way (step v), which could then add to the activated di- σ -bonded acetylene at the RuO₂ phase, lowering the overall activation energy barrier, also confirmed by the DFT calculations. Without the RuO₂ phase, there would not be such an activation of acetylene. Moreover, there is an expected decrease in the extent of acetylene adsorption on the reduced DPC/RuPt due to the negative charge accumulation on the metal surface in the presence of light and the π electron cloud, which may be the reason for its decreased stability.

This has now been addressed in the discussion in the revised MS.

2) It is well-accepted that Pd-based catalysts are best for semihydrogenation reaction. Why in this work authors use PtRu rather than Pd or Pd-M alloy?

>> While Pd and Pd-based alloys are undoubtedly well-established catalysts for semi-hydrogenation, the choice of PtRu was motivated by the desire to investigate alternative catalytic systems made up of plasmonic black gold as antenna and RuPt as reactor based on our experience in such plasmonic systems for CO₂ reduction reactions, that could potentially offer advantages in terms of selectivity, stability, and catalytic performance.

Point-by-point Response to the Reviewer # 3

Sharma et al. have reported on the development of RuPtOx nanoparticles on SiO₂-supported Au nanoparticles for the chemoselective hydrogenation of acetylene. This study combines extensive catalyst characterization, in situ spectroscopy, and DFT theory to support their conclusions. Overall, the work is thorough. It appears that their claims are well supported by both experiment and theory. The authors presented a convincing argument for the cooperativity effects of RuOx and Pt sites. It was especially nice to see that this catalyst performs under the industrially relevant condition of excess ethene. The proposed mechanism derived from in situ FTIR is also

logical, although in some regards some assumptions were made. This would warrant a follow-up study, in my opinion, which in an of itself could be highly impactful in the field of plasmonic photocatalysis.

>> We thank the reviewer for their insightful and encouraging comments on our work. It provided a boost to all of us for further development of plasmonic catalysts. We appreciate their thorough evaluation and recognition of our efforts in catalyst development, characterization, and mechanistic exploration.

At times, I found their claims were grandiose and that assertions of the efficacy of their materials were overblown or incomplete. This criticism is particularly true when comparing their work to previous reports (see Comparison of Best Reported Catalytic Systems & Table 1; Table S4). The authors cited references 29, 33-35 in their introduction, yet ignored all but Ref. 35 in their table & description in the main text. Several additional photocatalytic hydrogenation catalysts were reported in Table S4 and it was unclear why specific examples ended up in two tables reporting the same outcomes. I believe it's important for the authors to present a more leveled and unbiased comparison of the literature. This is always true, but especially so when comparing conversions and selectivity of reactions in series where reactor geometry, flow rates, reactant partial pressures, etc. are critical to the conclusions drawn. Many of the studies the authors cited use experimental conditions or different reactor designs that make a quantitative comparison between studies essentially meaningless. As reported, the text presents this work in a very positive light, which it deserves to be, but the data speaks for itself without potentially skewed colorful claims.

>> We apologise for using grandiose statements to present our work, which was unintentional. In the revised submission, we toned down our claims and conclusions and revised the text accordingly across the MS and SI.

We have compared our catalysts, first with only plasmonic systems in Table 1 (which included reference 35). And then, we also compared it with the non-plasmonic catalysts, summarized in Figure 3 and Table S4 (which has more details of figure 3, like reactor geometry, flow rates, and feed composition, as also suggested by this reviewer).

Reference 35 (Pd-Mg/GS) reports a plasmonic catalyst, therefore, it is included in Table 1. Somehow, we missed comparing with ref.33 (Au-Fe/C), which is now added in revised Table 1. Please note changes in reference numbers in the revised manuscript due to the addition of more references, as suggested by reviewer 1.

References 29 (Pd-SA/ N-graphene), 34 (Pd/TiO₂), on the other hand, pertain to non-plasmonic catalysis, hence were included in Figure 3 and Table S4. We have modified the title of the tables to prevent this confusion.

We completely agree with the reviewer about challenges in quantitative comparison. We have now provided as much details as possible in revised Table 1 and Table S4, such as reactor geometry, flow rates, feed composition, etc, to have better comparison. The catalyst performance for all the mentioned catalysts has been evaluated at ambient pressure.

Table 1. Comparative analysis of plasmonic catalytic systems in acetylene semi-hydrogenation.

Catalyst (metal loading and weight)	Light Intensity (λ)	Feed Composition in vol. % (total flow)	Reactor Type	Ethene Production Rate ($\text{mmol g}^{-1} \text{h}^{-1}$)	Acetylene Conversion (%)	Ethene Selectivity (%)	Ref.
DPC/RuPt (10 %, Ru:Pt=9:1, 5 mg)	2.7 W cm^{-2} (400-1100 nm)	$\text{C}_2\text{H}_2/\text{C}_2\text{H}_4/\text{H}_2/\text{Ar} = 2.72/54.5/13.6/24.5/4.5$ (110 mL min^{-1})	Fixed bed flow reactor with quartz window (crucible i.d-6mm)	320	18	88	This Work
		With excess ethene					
DPC/RuPt (10 %, Ru:Pt=9:1, 20 mg)	6 W cm^{-2} (AM1.5)	$\text{C}_2\text{H}_2/\text{C}_2\text{H}_4/\text{H}_2/\text{Ar} = 2.2/44.4/22.2/20/11.1$ (9 mL min^{-1})	Quartz flat cell flow reactor (internal gap is 0.5 mm in the flat section, length-50 mm, width-8mm)	31	97	87	This Work
		With excess ethene					
Pd-Mg/GS (Pd:3 %, 20mg)	1.8 W cm^{-2} (785 nm)	$\text{C}_2\text{H}_2/\text{H}_2/\text{N}_2 = 5/15/75$	Horizontally oriented packed bed reactor with CaF_2 window (6 mm) at the top of the reactor	NR	90	80	37
		$\text{C}_2\text{H}_2/\text{H}_2/\text{N}_2 = 5/20/75$ (Total flow: 5-200 mL min^{-1})		101	NR	NR	
DPC/Ni (Ni-10 %, 35 mg)	0.58 W cm^{-2} (400-1100 nm)	$\text{C}_2\text{H}_2/\text{H}_2/\text{Ar} = 0.12/2/97.8$ (Total flow- 14 mL min^{-1})	Quartz flat cell flow reactor (tube i.d-3.5 mm)	2.4	30	86	16
		Without ethene					
Au-Fe /C (1%, Au:Fe=1:1, 1g (in 2g quartz sand))	0.45 W cm^{-2} (250-1100 nm) T~130 °C, external heating)	$\text{C}_2\text{H}_2/\text{C}_2\text{H}_4/\text{H}_2/\text{Ar} = 1/20/20/59$ (Total flow- 20 mL min^{-1})	Photothermal fixed-bed reactor	0.5	98.4	97.5	35
		With excess ethene					
AINC-Pd (catalyst weight unknown)	14.3 W cm^{-2} (680-1080 nm)	$\text{C}_2\text{H}_2/\text{H}_2/\text{He}/\text{N}_2 = 1.33/3.33/25.33/70$ (Total Flow= 15 mL min^{-1})	Stainless steel gas-phase high temperature reaction chamber	26.7 $\times 10^{-3}$ (mmol h^{-1})	5	97	20
		Without ethene					

NR-not reported, vol.- volume

Table S4. Comparison of the best reported non-plasmonic catalytic systems employed for semi-hydrogenation of acetylene.

Entry	Catalyst (loading, weight)	Temperature (°C)	Feed Composition in vol. % (total flow)	Reactor Type	Ethene Production Rate (mmol g ⁻¹ h ⁻¹)	Acetylene Conversion (%)	Ethene Selectivity (%)	Ref.
1	DPC/RuPt-10-Calc (10 wt%, 5 mg)	T _R -75 Ts-132 with light 2.7 W cm ⁻² 400-1100 nm	C ₂ H ₂ /C ₂ H ₄ /H ₂ /Ar/Air = 2.72/54.5/13.6/24.5/4.5 (110 mL min ⁻¹)	Fixed bed flow reactor with quartz window (crucible i.d-6mm)	320	18	88	This Work
2	DPC/RuPt-10-Calc (10 wt%, 20 mg)	Ts- 262 light 6 W cm ⁻² AM 1.5	C ₂ H ₂ /C ₂ H ₄ /H ₂ /Ar/Air = 2.2/44.4/22.2/20/11.1 (9 mL min ⁻¹)	Quartz flat cell flow reactor (internal gap is 0.5 mm in the flat section, length-50 mm, width-8mm)	31	97	87	This Work
3	Pd/ZnO (1 wt%, 10 mg)	80	C ₂ H ₂ /C ₂ H ₄ /H ₂ /He = 2/40/20/38 (30 mL min ⁻¹)	Fixed bed quartz microreactor (i.d. 4 mm)	131.5	92	89	15
4	PdIn/MgAl ₂ O ₄ (2 wt%, 25 mg)	90	C ₂ H ₂ /C ₂ H ₄ /H ₂ /He = 0.5/50/5/44.5 (120 mL min ⁻¹)	Quartz bed flow reactor	54.9	95	90	16
5	Pd-Pt/SiO ₂ (1.5 wt%, 20 mg)	80	C ₂ H ₂ /H ₂ /He = 1.67/3.33/95 (60 mL min ⁻¹)	Fixed bed flow quartz reactor (1/4 inch)	52.0	97	40	17
6	Pd1/TiO ₂ (SAC) (0.15 wt%, 15 mg)	120 (dark) 60 (light-167 mW cm ⁻² , UV-Vis)	C ₂ H ₂ /C ₂ H ₄ /H ₂ /He = 1/20/10/69 (45 mL min ⁻¹)	Quartz fixed-bed flow reaction chamber equipped with a quartz window (d = 35 mm)	48.2 12.0	100 25	65-50 65	18
7	Pd _{1.0} /Bi ₂ O ₃ /TiO ₂ (Pd 2.3 wt%, Bi 4.9 wt%, 30mg)	44	C ₂ H ₂ /C ₂ H ₄ /H ₂ /He = 1/20/20/59 (60 mL min ⁻¹)	Fixed bed vertical quartz reactor	43.8	91	90	19
8	Pd Single atom/N-graphene (1.04 wt%, 50 mg)	125 (Photothermal: 5.1 W cm ⁻² , UV- Vis)	C ₂ H ₂ /C ₂ H ₄ /H ₂ /Ar = 1/20/20/59 (60 mL min ⁻¹)	Flow reactor with a quartz window at the top for light irradiation. (Volume- 50 cc)	29.7	99	93	20
9	Pd/ND@G (0.11 wt%, 30 mg)	180	C ₂ H ₂ /C ₂ H ₄ /H ₂ /He = 1/20/10/69 (30 mL min ⁻¹)	Quartz bed flow reactor	24.1	100	90	21
10	Pd _{0.006} Cu/SiO ₂ (Pd 0.05wt%, Cu 4.96wt%, 30 mg)	160	C ₂ H ₂ /C ₂ H ₄ /H ₂ /He = 1/20/20/59 (30 mL min ⁻¹)	Quartz reactor	22.7	100	85	22

11	Al ₁₃ Fe ₄ (20 mg)	200	C ₂ H ₂ /C ₂ H ₄ /H ₂ /He = 0.5/50/5/44.5 (30 mL min ⁻¹)	Quartz glass plug-flow reactor (i.d.- 7 mm) catalyst bed supported by a quartz glass frit.	13.6	80	85	23
12	PdZn-1.2@ZIF-8C (0.7 wt%, 50 mg)	120	C ₂ H ₂ /C ₂ H ₄ /H ₂ /Ar = 0.65/50/5/45 (40 mL min ⁻¹)	Fixed-bed quartz tubular reactor	9.4	85	80	24
13	Na-Ni@CHA (Na 6.3 wt%, Ni 3.5 wt%, 200 mg)	180	C ₂ H ₂ /H ₂ /He = 1/16/83 (50 mL min ⁻¹)	Quartz fixed bed flow micro reactor	6.0	100	90	25
14	GaPd/Al ₂ O ₃ (Pd 0.005 wt%, 75 mg)	200	C ₂ H ₂ /C ₂ H ₄ /H ₂ /He = 0.5/50/5/44.5 (30 mL min ⁻¹)	Plug flow reactor consisting of a quartz glass tube with a length of 300mm, i.d.- 7mm and a sintered glass frit to support the catalyst.	3.9	87	85	26
15	Ga ₂ O ₃ -Pd/Al ₂ O ₃ (0.23 wt%, 50 mg)	100	C ₂ H ₂ /C ₂ H ₄ /H ₂ /N ₂ = 0.3/33.1/0.6/66 (50 mL min ⁻¹)	Fixed-bed flow quartz tube reactor	1.5	20	95	27
16	Ni MoS/Al ₂ O ₃ (0.5 wt%, 0.5 g)	125	C ₂ H ₂ /C ₂ H ₄ /H ₂ /He = 0.15/15/3/82 (165 mL min ⁻¹)	Fixed-bed microreactor	1.19	100	90	28
17	Single-atom Pd (0.16 wt%, 1 g)	120	C ₂ H ₂ /C ₂ H ₄ /H ₂ /He = 0.5/50/5/44.5 (20 mL min ⁻¹)	Fixed-bed quartz-glass flow microreactor (i.d. = 6 mm).	0.2	96	93	29

i.d.- internal diameter of the reactor, vol.- volume

Overall, I commend the authors on their thorough work and recommend this for publication with minor revision (at the discretion of the editor).

>> We thank the reviewer for recommending our work for publication.